# Tight Generalization Bounds for Large-Margin Halfspaces

**Kasper Green Larsen**
Department of Computer Science
Aarhus University
Aarhus, Denmark
larsen@cs.au.dk

**Natascha Schalburg**
Department of Computer Science
Aarhus University
Aarhus, Denmark
n.schalburg@cs.au.dk

## Abstract

We prove the first generalization bound for large-margin halfspaces that is asymptotically tight in the tradeoff between the margin, the fraction of training points with the given margin, the failure probability and the number of training points.

## 1 Introduction

Halfspaces are arguably among the simplest and most fundamental classic learning models. Given a normal vector $w \in \mathbb{R}^d$ and a bias $b \in \mathbb{R}$ defining a hyperplane, the corresponding halfspace classifier predicts the label of a data point $x \in \mathbb{R}^d$ by returning $\text{sign}(\langle w, x \rangle + b)$, corresponding to a $+1$ label on points inside the halfspace above the hyperplane, and $-1$ on points below.

Classic examples of learning algorithms for obtaining a halfspace classifier from a training set of points $S = \{(x_i, y_i)\}_{i=1}^n$ with $(x_i, y_i) \in \mathbb{R}^d \times \{-1, 1\}$, includes the Perceptron Learning Algorithm (PLA) (Mcculloch and Pitts [1943]) and Support Vector Machines (SVM) (Cortes and Vapnik [1995]). A key intuition underlying SVM, is the empirical observation that halfspaces with a large *margin* to the training data tend to generalize well. Ignoring the bias variable $b$ (which we later handle by adding a special feature) and assuming $w \in \mathcal{S}^{d-1}$ (i.e. $w$ has unit length), the margin of the halfspace with normal vector $w$ on a labeled point $(x, y)$ is $y \langle w, x \rangle$. Observe that $\langle w, x \rangle$ gives the signed distance of $x$ from the hyperplane, and the margin is positive when $\text{sign}(\langle w, x \rangle)$ correctly predicts the label $y$. With this definition, *hard-margin* SVM computes the normal vector $w$ of the hyperplane with the largest minimum margin. There are also margin variants of the Perceptron (Freund and Schapire [1999]) that computes a halfspace with minimum margin approaching the optimal, as in hard-margin SVM.

To handle data that is not linearly separable, and to add robustness to outliers, the *soft-margin* SVM relaxes the optimization problem to the following

$$\min_{w, \xi} \|w\|_2^2 + \lambda \sum_i \xi_i, \qquad \text{s.t. } y_i \langle w, x_i \rangle \geq 1 - \xi_i, \qquad \xi_i \geq 0.$$

Here $\lambda > 0$ is a regularization parameter. The soft-margin SVM thus allows for smaller margins on some training points at the cost of a penalty $\lambda \xi_i$. For fast implementations of SVM, see Gu et al. [2025] and S. Shalev-Shwartz and Cotter [2011].

To theoretically justify and explain the empirical success of focusing on large margins, Bartlett and Shawe-Taylor [1999] proved the first generalization bounds upper bounding the probability $\mathcal{L}_{\mathcal{D}}(w) := \mathbb{P}_{(\mathbf{x}, \mathbf{y}) \sim \mathcal{D}}[\text{sign}(\langle w, \mathbf{x} \rangle) \neq \mathbf{y}]$ of misclassifying the label of a new data point. Concretely, Bartlett and Shawe-Taylor first studied the hard-margin case and proved that for any distribution $\mathcal{D}$ over $\mathbb{B}_2^d \times \{-1, 1\}$ and any $0 < \delta < 1$, it holds with probability at least $1 - \delta$ over a training set $\mathbf{S} \sim \mathcal{D}^n$ that for every $w \in \mathcal{S}^{d-1}$ and every margin $0 < \gamma < 1$, if $y \langle w, x \rangle \geq \gamma$ for all $(x, y) \in \mathbf{S}$

then

$$\mathcal{L}_{\mathcal{D}}(w) \leq c \cdot \left( \frac{\ln^2(n)}{\gamma^2 n} + \frac{\ln(e/\delta)}{n} \right), \tag{1}$$

for a constant $c > 0$. Here $\mathbb{B}_2^d$ is the $d$-dimensional unit ball and $\mathcal{S}^{d-1}$ is the $d$-dimensional unit sphere, both with respect to the $l_2$-norm. The restriction to $x \in \mathbb{B}_2^d$ can be relaxed by multiplying the first term by $R^2$ for $x \in R \cdot \mathbb{B}_2^d$. A dependency on the scaling of input points is inevitable as margins scale with $\|x\|_2$. Throughout the paper, we state bounds for $R = 1$ and remark that all bounds generalize to arbitrary $R$ by replacing $\gamma$ by $\gamma/R$.

Defining $\mathcal{L}_S^\gamma(w)$ as the fraction of data points in a training set $S$ where $w$ has margin at most $\gamma$, Bartlett and Shawe-Taylor [1999] also prove a more general result, saying that with probability $1 - \delta$ over $\mathbf{S} \sim \mathcal{D}^n$, it holds for every $w \in \mathcal{S}^{d-1}$ that

$$\mathcal{L}_{\mathcal{D}}(w) \leq \mathcal{L}_{\mathbf{S}}^\gamma(w) + c \cdot \sqrt{\frac{\ln^2(n)}{\gamma^2 n} + \frac{\ln(e/\delta)}{n}}. \tag{2}$$

This was later improved by Bartlett and Mendelson [2002] using Rademacher complexity arguments, replacing the $\ln^2(n)$ term in (2) by 1. Here, and throughout the paper, we refer to $\mathcal{L}_{\mathbf{S}}^\gamma(w)$ as the (empirical) *margin loss*.

**First-Order Bounds.** The first work to interpolate between the hard-margin and soft-margin bounds was due to McAllester [2003], who gave a general tradeoff of

$$\mathcal{L}_{\mathcal{D}}(w) \leq \mathcal{L}_{\mathbf{S}}^\gamma(w) + c \cdot \left( \sqrt{\mathcal{L}_{\mathbf{S}}^\gamma(w) \cdot \frac{\ln n}{\gamma^2 n}} + \frac{\ln n}{\gamma^2 n} + \sqrt{\frac{\ln n + \ln(e/\delta)}{n}} \right). \tag{3}$$

Notice how the $\mathcal{L}_{\mathbf{S}}^\gamma(w)$ term is multiplied onto $\ln n/(\gamma^2 n)$ inside the first square-root. Since the hard-margin case corresponds to this term being 0, this gives a way of interpolating between the cases. Such bounds are often referred to as *first-order bounds*. Unfortunately, (3) still has the seemingly superfluous $\sqrt{(\ln n + \ln(e/\delta))/n}$ term even when $\mathcal{L}_{\mathbf{S}}^\gamma(w) = 0$ and thus falls short of even matching (1) in the hard-margin case.

The current state-of-the-art generalization bound is due to Grønlund et al. [2020a] and states that with probability $1 - \delta$ over $\mathbf{S} \sim \mathcal{D}^n$, it holds for every $w \in \mathcal{S}^{d-1}$ that

$$\mathcal{L}_{\mathcal{D}}(w) \leq \mathcal{L}_{\mathbf{S}}^\gamma(w) + c \cdot \left( \sqrt{\mathcal{L}_{\mathbf{S}}^\gamma(w) \cdot \left( \frac{\ln n}{\gamma^2 n} + \frac{\ln(e/\delta)}{n} \right)} + \frac{\ln n}{\gamma^2 n} + \frac{\ln(e/\delta)}{n} \right). \tag{4}$$

This improves previous hard-margin bounds by a logarithmic factor and gives a cleaner interpolation between the hard- and soft-margin cases. Furthermore, the bound is close to optimal. Concretely, the dependency on $\delta$ is optimal by tweaking standard results for agnostic PAC learning, see e.g. Devroye et al. [1996] [Chapter 11]. Moreover, Grønlund et al. [2020a] complemented their upper bound by the following lower bound

**Theorem 1** (Grønlund et al. [2020a]). *There is a constant $c > 0$ such that for any $cn^{-1/2} < \gamma < c^{-1}$, any parameter $0 \leq \tau \leq 1$, and any $n \geq c$, there is a distribution $\mathcal{D}$ such that it holds with constant probability over $\mathbf{S} \sim \mathcal{D}^n$ that there is a $w \in \mathcal{S}^{d-1}$ such that $\mathcal{L}_{\mathbf{S}}^\gamma(w) \leq \tau$ and*

$$\mathcal{L}_{\mathcal{D}}(w) \geq \mathcal{L}_{\mathbf{S}}^\gamma(w) + c \cdot \left( \sqrt{\tau \cdot \frac{\ln(e/\tau)}{\gamma^2 n}} + \frac{\ln(\gamma^2 n)}{\gamma^2 n} \right)$$

$$\geq \mathcal{L}_{\mathbf{S}}^\gamma(w) + c \cdot \left( \sqrt{\mathcal{L}_{\mathbf{S}}^\gamma(w) \cdot \frac{\ln(e/\mathcal{L}_{\mathbf{S}}^\gamma(w))}{\gamma^2 n}} + \frac{\ln(\gamma^2 n)}{\gamma^2 n} \right).$$

Notice how the parameter $\tau$ allows for showing that the upper bound (4) is nearly tight across the range of $\mathcal{L}_{\mathbf{S}}^\gamma(w)$. Let us also remark that Grønlund et al. [2020a] states their lower bound with a $\ln n$ rather than $\ln(e\gamma^2 n)$, but require that $\gamma > n^{-0.499}$. A careful examination of their proof however reveals the more general lower bound stated here.

Unfortunately, there still remains a discrepancy between the lower bound and (4). Concretely, there is a gap of $\sqrt{\ln n / \ln(e/\mathcal{L}_{\mathbf{S}}^{\gamma}(w))}$. Moreover, for constant $\mathcal{L}_{\mathbf{S}}^{\gamma}(w)$, the Rademacher complexity based bound in (2) improves over both of the first-order bounds (3) and (4), and matches the lower bound in Theorem 1. This seems to suggest that a better upper bound might be possible.

**Our Contribution.** In this work, we settle the generalization performance of large-margin half-spaces by proving a new upper bound matching the lower bound in Theorem 1 across the entire tradeoff between $\gamma$, $\mathcal{L}_{\mathbf{S}}^{\gamma}(w)$ and $n$ (and is also tight in terms of $\delta$). Our result is stated in the following theorem

**Theorem 2.** *There is a constant $c > 0$ such that for any distribution $\mathcal{D}$ over $\mathbb{B}_2^d \times \{-1, 1\}$, it holds with probability at least $1 - \delta$ over $\mathbf{S} \sim \mathcal{D}^n$ that for every $w \in \mathcal{S}^{d-1}$ and every margin $n^{-1/2} \leq \gamma \leq 1$, we have*

$$\mathcal{L}_{\mathcal{D}}(w) \leq \mathcal{L}_{\mathbf{S}}^{\gamma}(w) + c \left( \sqrt{\mathcal{L}_{\mathbf{S}}^{\gamma}(w) \cdot \left( \frac{\ln(e/\mathcal{L}_{\mathbf{S}}^{\gamma}(w))}{\gamma^2 n} + \frac{\ln(e/\delta)}{n} \right)} + \frac{\ln(e\gamma^2 n)}{\gamma^2 n} + \frac{\ln(e/\delta)}{n} \right).$$

Using a simple reduction, our results generalize to all non-homogeneous halfspaces (i.e. including a bias term) and data in a ball of radius $R$, yielding Theorem 2 with $\gamma/R$ in place of $\gamma$. See Appendix A.

While one might argue that our improvement is small in magnitude, this finally pins down the exact generalization performance of a classic learning model. Furthermore, our proof of Theorem 2 brings several novel ideas that we hope may find further applications in generalization bounds.

We next proceed to give an overview of our proof and new ideas in Section 2, before giving the full details of the proof in Section 3.

## 2 Proof Overview

In this section, we present the main ideas in our proof of Theorem 2. As our proof builds on, and greatly extends, the work of Grønlund et al. [2020a] establishing the previous state-of-the-art in (4), we first present their overall proof strategy and the barriers we need to overcome to obtain our tight generalization bound. Throughout this proof overview, we use the notation $x \lesssim y$ to denote that there is an absolute constant $c > 0$ so that $x \leq cy$.

### 2.1 Previous Proof

The proof of Grønlund et al. [2020a] follows a framework proposed by Schapire et al. [1998] for proving generalization of large-margin *voting classifiers* (i.e. boosting). The main idea is to randomly discretize the infinite hypothesis set $\mathcal{S}^{d-1}$ to obtain a finite set $\mathcal{G} \subseteq \mathbb{R}^d \to \{-1, 1\}$. If $\mathcal{G}$ is small enough, then a standard union bound over all $h \in \mathcal{G}$ suffices to bound the difference between the empirical error and the true error $\mathcal{L}_{\mathcal{D}}(h)$ for every $h \in \mathcal{G}$. The key trick is to exploit large margins to allow for a discretization to a smaller $\mathcal{G}$.

To elaborate on the above, let us first generalize our notation $\mathcal{L}_{\mathcal{D}}(w)$ and $\mathcal{L}_S^{\gamma}(w)$ a bit. For a distribution $\mathcal{D}$ over $\mathbb{B}_2^d \times \{-1, 1\}$, let $\mathcal{L}_{\mathcal{D}}^{\gamma}(w) := \mathbb{P}_{(\mathbf{x}, \mathbf{y}) \sim \mathcal{D}}[\mathbf{y}\langle w, \mathbf{x} \rangle \leq \gamma]$, that is, $\mathcal{L}_{\mathcal{D}}^{\gamma}(w)$ is the probability over a fresh sample $(\mathbf{x}, \mathbf{y})$ from $\mathcal{D}$, of $w$ having margin no more than $\gamma$ on $(\mathbf{x}, \mathbf{y})$. For a training set $S$, we slightly abuse notation and write $(\mathbf{x}, \mathbf{y}) \sim S$ to denote a uniform random sample from $S$. We thus have

$$\mathcal{L}_S^{\gamma}(w) := \mathbb{P}_{(\mathbf{x}, \mathbf{y}) \sim S}[\mathbf{y}\langle w, \mathbf{x} \rangle \leq \gamma] = \frac{|\{(x, y) \in S : y\langle w, x \rangle \leq \gamma\}|}{|S|}.$$

When writing $\mathcal{L}_{\mathcal{D}}(w)$ we implicitly mean $\mathcal{L}_{\mathcal{D}}^0(w)$ and note that this coincides with our previous definition of $\mathcal{L}_{\mathcal{D}}(w) = \mathbb{P}_{(\mathbf{x}, \mathbf{y}) \sim \mathcal{D}}[\text{sign}(\langle w, \mathbf{x} \rangle) \neq \mathbf{y}]$ (defining $\text{sign}(0) = 0$).

**Random Discretization.** With this notation, the main idea in the proof of Grønlund et al. [2020a], is to apply a Johnson-Lindenstrauss transform (Johnson and Lindenstrauss [1984]), followed by a random snapping to a grid, in order to map each $w \in \mathcal{S}^{d-1}$ to a point on a grid $\mathcal{G}$ of size $\exp(ck)$ in $\mathbb{R}^k$, with $c > 0$ a sufficiently large constant. In more detail, let $\mathbf{A}$ be a $k \times d$ matrix

with i.i.d. $\mathcal{N}(0, 1/k)$ normal distributed entries. Such a matrix is a classic implementation of the Johnson-Lindenstrauss transform and has the property that $|\langle \mathbf{A}w, \mathbf{A}x \rangle - \langle w, x \rangle|$ is greater than $\varepsilon$ with probability at most $\exp(-\varepsilon^2 k/c)$ when $\|w\|_2, \|x\|_2 \leq 1$ (Dasgupta and Gupta [2003]). Note that this also preserves the norm of a vector $w$ by considering $x = w$ and noting $\langle w, w \rangle = \|w\|_2^2$. Secondly, following an idea of Alon and Klartag [2017] in a lower bound proof for the Johnson-Lindenstrauss transform, Grønlund et al. [2020a] randomly round $\mathbf{A}w$ to a point $h_{\mathbf{A},\mathbf{t}}(w)$ with coordinates integer multiples of $k^{-1/2}$ while guaranteeing that $|\langle h_{\mathbf{A},\mathbf{t}}(w), \mathbf{A}x \rangle - \langle \mathbf{A}w, \mathbf{A}x \rangle|$ is less than $\varepsilon$, except with probability $\exp(-\varepsilon^2 k/c)$. Here we use $\mathbf{t}$ to denote the randomness involved in the rounding.

Now choosing $\varepsilon = \gamma/4$ gives, by the triangle inequality, that $|\langle h_{\mathbf{A},\mathbf{t}}(w), \mathbf{A}x \rangle - \langle w, x \rangle| \leq \gamma/2$, except with probability $2\exp(-\gamma^2 k/(16c))$. Furthermore, by plugging in $x = w$ and setting $\varepsilon = 1$, we can also deduce that $\|h_{\mathbf{A},\mathbf{t}}(w)\|_2 \leq 2$ except with probability $\exp(-k/c)$. Simple counting arguments show that there are only $\exp(ck)$ many vectors of norm at most 2 with all coordinates integer multiples of $k^{-1/2}$. That is, except with probability $\exp(-k/c)$, $h_{\mathbf{A},\mathbf{t}}(w)$ belongs to a finite set $\mathcal{G}$ of $\exp(ck)$ many points.

**Framework.** With the above random discretization, the proof of Grønlund et al. [2020a] now follows the framework of Schapire et al. [1998] by relating $\mathcal{L}_{\mathcal{D}}(w)$ to $\mathcal{L}_{\mathbf{A}\mathcal{D}}^{\gamma/2}(h_{\mathbf{A},\mathbf{t}}(w))$ and $\mathcal{L}_{\mathbf{S}}^{\gamma}(w)$ to $\mathcal{L}_{\mathbf{A}\mathbf{S}}^{\gamma/2}(h_{\mathbf{A},\mathbf{t}}(w))$. Here $\mathbf{A}\mathcal{D}$ is the distribution obtained by sampling $(\mathbf{x}, \mathbf{y}) \sim \mathcal{D}$ and returning $(\mathbf{A}\mathbf{x}, \mathbf{y})$. Similarly, $\mathbf{A}S$ is the training set obtained by replacing each $(x, y) \in S$ by $(\mathbf{A}x, y)$. The intuition is that the random discretization changes margins by no more than $\gamma/2$ for most data points and hence points with margin at most 0 under $\mathcal{D}$ often have margin at most $\gamma/2$ under $\mathbf{A}\mathcal{D}$ and similarly for $S$ and $\mathbf{A}S$. Let us make this more formal. We have for any $A, t$ in the support of $\mathbf{A}, \mathbf{t}$ that

$$\mathcal{L}_{\mathcal{D}}(w) \leq \mathcal{L}_{A\mathcal{D}}^{\gamma/2}(h_{A,t}(w)) + \mathbb{P}_{(\mathbf{x},\mathbf{y})\sim\mathcal{D}}[\mathbf{y}\langle w, \mathbf{x} \rangle \leq 0 \wedge \mathbf{y}\langle h_{A,t}(w), A\mathbf{x} \rangle > \gamma/2]. \quad (5)$$

Similarly, we have

$$\mathcal{L}_S^{\gamma}(w) \geq \mathcal{L}_{AS}^{\gamma/2}(h_{A,t}(w)) - \mathbb{P}_{(\mathbf{x},\mathbf{y})\sim S}[\mathbf{y}\langle w, \mathbf{x} \rangle > \gamma \wedge \mathbf{y}\langle h_{A,t}(w), A\mathbf{x} \rangle \leq \gamma/2]. \quad (6)$$

Taking expectation we see that

$$\mathcal{L}_{\mathcal{D}}(w) - \mathcal{L}_{\mathbf{S}}^{\gamma}(w) = \mathbb{E}_{\mathbf{A},\mathbf{t}}[\mathcal{L}_{\mathcal{D}}(w) - \mathcal{L}_{\mathbf{S}}^{\gamma}(w)]$$

$$\leq \mathbb{E}_{\mathbf{A},\mathbf{t}}[\mathcal{L}_{\mathbf{A}\mathcal{D}}^{\gamma/2}(h_{\mathbf{A},\mathbf{t}}(w)) - \mathcal{L}_{\mathbf{A}S}^{\gamma/2}(h_{\mathbf{A},\mathbf{t}}(w))] \quad (7)$$

$$+ \mathbb{E}_{\mathbf{A},\mathbf{t}}[\mathbb{P}_{(\mathbf{x},\mathbf{y})\sim\mathcal{D}}[\mathbf{y}\langle w, \mathbf{x} \rangle \leq 0 \wedge \mathbf{y}\langle h_{\mathbf{A},\mathbf{t}}(w), \mathbf{A}\mathbf{x} \rangle > \gamma/2]] \quad (8)$$

$$+ \mathbb{E}_{\mathbf{A},\mathbf{t}}[\mathbb{P}_{(\mathbf{x},\mathbf{y})\sim S}[\mathbf{y}\langle w, \mathbf{x} \rangle > \gamma \wedge \mathbf{y}\langle h_{\mathbf{A},\mathbf{t}}(w), \mathbf{A}\mathbf{x} \rangle \leq \gamma/2]]. \quad (9)$$

To bound (7), we exploit that $h_{\mathbf{A},\mathbf{t}}(w)$ belongs to the grid $\mathcal{G}$, except with probability $\exp(-k/c)$. Using Bernstein's inequality (and a careful partitioning of hypotheses $w$ depending on $\mathcal{L}_{\mathcal{D}}^{\gamma}(w)$), it is possible to union bound over the entire grid and conclude

$$\mathbb{E}_{\mathbf{A},\mathbf{t}}[\mathcal{L}_{\mathbf{A}\mathcal{D}}^{\gamma/2}(h_{\mathbf{A},\mathbf{t}}(w)) - \mathcal{L}_{\mathbf{A}S}^{\gamma/2}(h_{\mathbf{A},\mathbf{t}}(w))] \leq$$

$$\mathbb{E}_{\mathbf{A},\mathbf{t}}[\sup_{h \in \mathcal{G}} \mathcal{L}_{\mathbf{A}\mathcal{D}}^{\gamma/2}(h) - \mathcal{L}_{\mathbf{A}S}^{\gamma/2}(h)] + \mathbb{P}_{\mathbf{A},\mathbf{t}}[h_{\mathbf{A},\mathbf{t}}(w) \notin \mathcal{G}] \lesssim$$

$$\sqrt{\mathcal{L}_{\mathbf{S}}^{\gamma}(w) \cdot \frac{\ln(|\mathcal{G}|/\delta)}{n}} + \frac{\ln(|\mathcal{G}|/\delta)}{n} + \exp(-k/c) \lesssim$$

$$\sqrt{\mathcal{L}_{\mathbf{S}}^{\gamma}(w) \cdot \frac{k + \ln(e/\delta)}{n}} + \frac{k + \ln(e/\delta)}{n} + \exp(-k/c). \quad (10)$$

To bound (9), we use the guarantees of the random discretization to conclude that

$$\mathbb{E}_{\mathbf{A},\mathbf{t}}[\mathbb{P}_{(\mathbf{x},\mathbf{y})\sim S}[\mathbf{y}\langle w, \mathbf{x} \rangle > \gamma \wedge \mathbf{y}\langle h_{\mathbf{A},\mathbf{t}}(w), \mathbf{A}\mathbf{x} \rangle \leq \gamma/2]] =$$

$$\mathbb{E}_{(\mathbf{x},\mathbf{y})\sim S}[\mathbb{P}_{\mathbf{A},\mathbf{t}}[\mathbf{y}\langle w, \mathbf{x} \rangle > \gamma \wedge \mathbf{y}\langle h_{\mathbf{A},\mathbf{t}}(w), \mathbf{A}\mathbf{x} \rangle \leq \gamma/2]] \leq$$

$$\mathbb{E}_{(\mathbf{x},\mathbf{y})\sim S}[\mathbb{P}_{\mathbf{A},\mathbf{t}}[\mathbf{y}\langle h_{\mathbf{A},\mathbf{t}}(w), \mathbf{A}\mathbf{x} \rangle \leq \gamma/2 \mid \mathbf{y}\langle w, \mathbf{x} \rangle > \gamma]] \leq 2\exp(-\gamma^2 k/(16c)).$$

We can bound (8) in a similar fashion (even with slightly better guarantees scaled by $\mathcal{L}_{\mathcal{D}}(w)$, but this does not help for (9)). The final generalization error thus becomes

$$\mathcal{L}_{\mathcal{D}}(w) \leq \mathcal{L}_{\mathbf{S}}^{\gamma}(w) + c' \cdot \left( \sqrt{\mathcal{L}_{\mathbf{S}}^{\gamma}(w) \cdot \frac{k + \ln(e/\delta)}{n}} + \frac{k + \ln(e/\delta)}{n} + \exp(-\gamma^2 k/c') \right), \quad (11)$$

where $c' > 0$ is a sufficiently large constant. Comparing this expression with the desired bound from Theorem 2, we see that we have to choose $k$ large enough that $c' \exp(-\gamma^2 k/c')$ is no larger than

$$\sqrt{\mathcal{L}_{\mathbf{S}}^{\gamma}(w) \cdot \left( \frac{\ln(e/\mathcal{L}_{\mathbf{S}}^{\gamma}(w))}{\gamma^2 n} + \frac{\ln(e/\delta)}{n} \right)} + \frac{\ln(e\gamma^2 n)}{\gamma^2 n} + \frac{\ln(e/\delta)}{n}.$$

This basically solves to

$$k \gtrsim \gamma^{-2} \ln \left( \frac{\gamma^2 n}{\mathcal{L}_{\mathbf{S}}^{\gamma}(w) \ln(e/\mathcal{L}_{\mathbf{S}}^{\gamma}(w))} \right) \geq \gamma^{-2} \ln \left( \gamma^2 n \right).$$

Inserting this $k$ in (11) recovers the bound by Grønlund et al. [2020a] stated in (4).

**Barriers.** In light of the above discussion, we identify some key barriers for the previous proof technique. Concretely, if we examine (11), the term $\sqrt{\mathcal{L}_{\mathbf{S}}^{\gamma}(w)k/n}$ requires us to choose $k$ no larger than $c\gamma^{-2} \ln(e/\mathcal{L}_{\mathbf{S}}^{\gamma}(w))$ to match the optimal bound we get in Theorem 2. Unfortunately, the additive $\exp(-\gamma^2 k/c')$ term originating from handling (8) and (9) then becomes $\text{poly}(\mathcal{L}_{\mathbf{S}}^{\gamma}(w))$, which is too expensive. In fact, even the additive $\exp(-k/c)$ term from handling (7) is too expensive for e.g. constant $\gamma$. Nonetheless, we will in fact choose such $k$ and identify a tighter strategy for analysing $\mathcal{L}_{\mathcal{D}}(w) - \mathcal{L}_{\mathbf{S}}^{\gamma}(w)$.

### 2.2 Our Key Improvements

Our first main observation is that the two upper bounds in (5) and (6) are not completely tight, i.e. they are inequalities, not equalities. In (5) we for instance ignore points $(x, y)$ that had a margin greater than 0 for $w$, but where the margin of $(Ax, y)$ is less than $\gamma/2$ for $h_{A,t}(w)$. Taking these into accounts, we get the tighter bounds

$$\begin{aligned}
\mathcal{L}_{\mathcal{D}}(w) = \mathcal{L}_{A\mathcal{D}}^{\gamma/2}(h_{A,t}(w)) &+ \mathbb{P}_{(\mathbf{x},\mathbf{y})\sim\mathcal{D}}[\mathbf{y}\langle w, \mathbf{x}\rangle \leq 0 \wedge \mathbf{y}\langle h_{A,t}(w), A\mathbf{x}\rangle > \gamma/2] \\
&- \mathbb{P}_{(\mathbf{x},\mathbf{y})\sim\mathcal{D}}[\mathbf{y}\langle w, \mathbf{x}\rangle > 0 \wedge \mathbf{y}\langle h_{A,t}(w), A\mathbf{x}\rangle \leq \gamma/2],
\end{aligned}$$

and

$$\begin{aligned}
\mathcal{L}_{S}^{\gamma}(w) = \mathcal{L}_{AS}^{\gamma/2}(h_{A,t}(w)) &- \mathbb{P}_{(\mathbf{x},\mathbf{y})\sim S}[\mathbf{y}\langle w, \mathbf{x}\rangle > \gamma \wedge \mathbf{y}\langle h_{A,t}(w), A\mathbf{x}\rangle \leq \gamma/2] \\
&+ \mathbb{P}_{(\mathbf{x},\mathbf{y})\sim S}[\mathbf{y}\langle w, \mathbf{x}\rangle \leq \gamma \wedge \mathbf{y}\langle h_{A,t}(w), A\mathbf{x}\rangle > \gamma/2].
\end{aligned}$$

With these refined bounds, we can now split $\mathcal{L}_{\mathcal{D}}(w) - \mathcal{L}_{S}^{\gamma}(w)$ into a sum of three terms:

$$\begin{aligned}
\mathcal{L}_{A\mathcal{D}}^{\gamma/2}&(h_{A,t}(w)) - \mathcal{L}_{AS}^{\gamma/2}(h_{A,t}(w)) \\
&+ \mathbb{P}_{\mathcal{D}}[\mathbf{y}\langle w, \mathbf{x}\rangle \leq 0 \wedge \mathbf{y}\langle h_{A,t}(w), A\mathbf{x}\rangle > \gamma/2] - \mathbb{P}_{S}[\mathbf{y}\langle w, \mathbf{x}\rangle \leq \gamma \wedge \mathbf{y}\langle h_{A,t}(w), A\mathbf{x}\rangle > \gamma/2]
\end{aligned} \tag{12}$$

$$+ \mathbb{P}_{S}[\mathbf{y}\langle w, \mathbf{x}\rangle > \gamma \wedge \mathbf{y}\langle h_{A,t}(w), A\mathbf{x}\rangle \leq \gamma/2] - \mathbb{P}_{\mathcal{D}}[\mathbf{y}\langle w, \mathbf{x}\rangle > 0 \wedge \mathbf{y}\langle h_{A,t}(w), A\mathbf{x}\rangle \leq \gamma/2]. \tag{13}$$

The first line is the same as (7) from before, but (12) and (13) improves over (8) and (9) by subtracting off a term. Intuitively, our more refined bounds allow us to argue that if the randomized rounding creates a big difference between $\mathcal{L}_{\mathcal{D}}(w)$ and $\mathcal{L}_{\mathcal{D}}^{\gamma/2}(h_{A,t}(w))$, then it creates a comparably large difference between $\mathcal{L}_{S}^{\gamma}(w)$ and $\mathcal{L}_{S}^{\gamma/2}(h_{A,t}(w))$, thereby canceling out. We will carefully exploit this in the following. Let us focus on (12) and remark that (13) is handled symmetrically. For (12), we see that

$$\mathbb{P}_{(\mathbf{x},\mathbf{y})\sim S}[\mathbf{y}\langle w, \mathbf{x}\rangle \leq \gamma \wedge \mathbf{y}\langle h_{A,t}(w), A\mathbf{x}\rangle > \gamma/2] \geq \mathbb{P}_{(\mathbf{x},\mathbf{y})\sim S}[\mathbf{y}\langle w, \mathbf{x}\rangle \leq 0 \wedge \mathbf{y}\langle h_{A,t}(w), A\mathbf{x}\rangle > \gamma/2],$$

and thus (12) is at most

$$\mathbb{P}_{(\mathbf{x},\mathbf{y})\sim\mathcal{D}}[\mathbf{y}\langle w, \mathbf{x}\rangle \leq 0 \wedge \mathbf{y}\langle h_{A,t}(w), A\mathbf{x}\rangle > \gamma/2] - \mathbb{P}_{(\mathbf{x},\mathbf{y})\sim S}[\mathbf{y}\langle w, \mathbf{x}\rangle \leq 0 \wedge \mathbf{y}\langle h_{A,t}(w), A\mathbf{x}\rangle > \gamma/2].$$

Now introducing the expectation over the randomized rounding $\mathbf{A}$ and $\mathbf{t}$ as in the previous proof, and using linearity of expectation, we want to bound the following expression with probability $1 - \delta$ over

$$\mathbf{S} \sim \mathcal{D}^n$$

$$\sup_{w \in \mathcal{S}^{d-1}} \Bigg( \mathbb{E}_{\mathbf{A},\mathbf{t}}[\mathbb{P}_{(\mathbf{x},\mathbf{y}) \sim \mathcal{D}}[\mathbf{y}\langle w, \mathbf{x}\rangle \leq 0 \wedge \mathbf{y}\langle h_{\mathbf{A},\mathbf{t}}(w), \mathbf{A}\mathbf{x}\rangle > \gamma/2]] -$$

$$\mathbb{E}_{\mathbf{A},\mathbf{t}}[\mathbb{P}_{(\mathbf{x},\mathbf{y}) \sim S}[\mathbf{y}\langle w, \mathbf{x}\rangle \leq 0 \wedge \mathbf{y}\langle h_{\mathbf{A},\mathbf{t}}(w), \mathbf{A}\mathbf{x}\rangle > \gamma/2]] \Bigg) =$$

$$\sup_{w \in \mathcal{S}^{d-1}} \Bigg( \mathbb{E}_{(\mathbf{x},\mathbf{y}) \sim \mathcal{D}}[\mathbb{P}_{\mathbf{A},\mathbf{t}}[\mathbf{y}\langle w, \mathbf{x}\rangle \leq 0 \wedge \mathbf{y}\langle h_{\mathbf{A},\mathbf{t}}(w), \mathbf{A}\mathbf{x}\rangle > \gamma/2]] -$$

$$\mathbb{E}_{(\mathbf{x},\mathbf{y}) \sim \mathbf{S}}[\mathbb{P}_{\mathbf{A},\mathbf{t}}[\mathbf{y}\langle w, \mathbf{x}\rangle \leq 0 \wedge \mathbf{y}\langle h_{\mathbf{A},\mathbf{t}}(w), \mathbf{A}\mathbf{x}\rangle > \gamma/2]] \Bigg). \tag{14}$$

This now has a form that looks familiar. Concretely, we have a function

$$\psi_w(x,y) = 1\{y\langle w, x\rangle \leq 0\} \cdot \mathbb{P}_{\mathbf{A},\mathbf{t}}[y\langle h_{\mathbf{A},\mathbf{t}}(w), \mathbf{A}x\rangle > \gamma/2]. \tag{15}$$

for each $w \in \mathcal{S}^{d-1}$, and wish to bound $\sup_w \mathbb{E}_{(\mathbf{x},\mathbf{y}) \sim \mathcal{D}}[\psi_w(\mathbf{x},\mathbf{y})] - \mathbb{E}_{(\mathbf{x},\mathbf{y}) \sim \mathbf{S}}[\psi_w(\mathbf{x},\mathbf{y})]$ with high probability over $\mathbf{S} \sim \mathcal{D}^n$. Rademacher complexity (see e.g. Shalev-Shwartz and Ben-David [2014]) is one key tool for bounding such differences. In particular, the contraction principle from Ledoux and Talagrand [1991] allows us to bound such a supremum when the functions $\psi_w$ are composite functions $\psi_w = f \circ g_w$ with $f : \mathbb{R} \to \mathbb{R}$ having bounded Lipschitz constant. In (Bartlett and Mendelson [2002]) this method is used, with $f$ being the ramp loss, resulting in a bound on (14) of $\sqrt{1/(\gamma^2 n)}$. We wish to take a similar approach for our $\psi_w$ in (15).

To argue that $\psi_w = f \circ g_w$ with $g_w(x,y) = y\langle w, x\rangle$, we need the probability in (15) to only depend on the original margin $y\langle w, x\rangle$. This is precisely the statement of Claim 3, which is proven in Appendix B.1. We thus proceed to bound the Lipschitz constant of the function $f$ in (15). To avoid discontinuities, we have to alter $\psi_w(x,y)$ somewhat to not include the discontinuous indicator function, and we eventually bound the Lipschitz constant $L$ by roughly

$$L \lesssim \gamma^{-1}\mathbb{P}_{\mathbf{A},\mathbf{t}}[y\langle h_{\mathbf{A},\mathbf{t}}(w), \mathbf{A}x\rangle > \gamma/2 \mid y\langle w, x\rangle = 0].$$

With a slight abuse of notation, we write $\mathbb{P}_{\mathbf{A},\mathbf{t}}[y\langle h_{\mathbf{A},\mathbf{t}}(w), \mathbf{A}x\rangle > \gamma/2 \mid y\langle w, x\rangle = 0]$ to denote the probability $\mathbb{P}_{\mathbf{A},\mathbf{t}}[y\langle h_{\mathbf{A},\mathbf{t}}(w), \mathbf{A}x\rangle > \gamma/2]$ for an arbitrary $x, w \in \mathcal{S}^{d-1}$ and $y \in \{-1, 1\}$ with $y\langle w, x\rangle = 0$ as $y\langle w, x\rangle$ completely determines this probability as argued above.

Since our randomized rounding preserves inner products to within $\gamma/2$ except with probability $\exp(-\gamma^2 k/c)$, we get $L \lesssim \gamma^{-1}\exp(-\gamma^2 k/c)$. This finally bounds (14) by

$$c \cdot \sqrt{\frac{\exp(-\gamma^2 k/c)}{\gamma^2 n}}.$$

This should be compared to proof by Grønlund et al. [2020a] that got a bound of $c\exp(-\gamma^2 k/c)$ and the $\sqrt{1/(\gamma^2 n)}$ bound mentioned above. This improvement is precisely enough to derive our tight Theorem 2. Indeed, as mentioned in (10), we can bound $\mathcal{L}_{\mathbf{A}\mathcal{D}}^{\gamma/2}(h_{\mathbf{A},\mathbf{t}}(w)) - \mathcal{L}_{\mathbf{A}S}^{\gamma/2}(h_{\mathbf{A},\mathbf{t}}(w))$ by

$$\sqrt{\mathcal{L}_{\mathbf{S}}^{\gamma}(w) \cdot \frac{k + \ln(e/\delta)}{n}} + \frac{k + \ln(e/\delta)}{n} + \exp(-k/c).$$

If we ignore the $\exp(-k/c)$ term and set $k = c'\gamma^{-2}\ln(e/\mathcal{L}_{\mathbf{S}}^{\gamma}(w))$, this gives the tight bound in Theorem 2.

Unfortunately, we cannot afford to ignore the $\exp(-k/c)$ term and we need additional ideas for dealing with it. Recall that in the previous proof by Grønlund et al. [2020a], it originates from bounding

$$\mathbb{E}_{\mathbf{A},\mathbf{t}}[\mathcal{L}_{\mathbf{A}\mathcal{D}}^{\gamma/2}(h_{\mathbf{A},\mathbf{t}}(w)) - \mathcal{L}_{\mathbf{A}S}^{\gamma/2}(h_{\mathbf{A},\mathbf{t}}(w))] \leq \mathbb{E}_{\mathbf{A},\mathbf{t}}[\sup_{h \in \mathcal{G}} \mathcal{L}_{\mathbf{A}\mathcal{D}}^{\gamma/2}(h) - \mathcal{L}_{\mathbf{A}S}^{\gamma/2}(h)] + \mathbb{P}_{\mathbf{A},\mathbf{t}}[h_{\mathbf{A},\mathbf{t}}(w) \notin \mathcal{G}],$$

and upper bounding $\mathbb{P}_{\mathbf{A},\mathbf{t}}[h_{\mathbf{A},\mathbf{t}}(w) \notin \mathcal{G}]$ by $\exp(-k/c)$. Here we instead consider an infinite sequence of discretizations/grids $\mathcal{G}_0, \mathcal{G}_1, \ldots$, and argue that the random rounding $\mathbf{A}, \mathbf{t}$ and training set $\mathbf{S}$ is simultaneously *good* (for some appropriate definition) for all grids with high probability. Here the grids $\mathcal{G}_i$ correspond to increasingly large norms of $h_{\mathbf{A},\mathbf{t}}(w)$, i.e. $\mathcal{G}_i$ contains all vectors of norm at most $2^{i+1}\mathbb{B}_2^d$ and all coordinates integer multiples of $k^{-1/2}$. Multiple careful applications of Cauchy-Schwartz, Jensen's inequality and upper bounds on the probability that $h_{\mathbf{A},\mathbf{t}}(w) \notin \mathcal{G}_i$ allows us to finally get rid of the $\exp(-k/c)$ factor.

# 3 Main Proof

We now set out to prove Theorem 2 following the proof outline sketched in Section 2. We start by a series of reductions that allow us to focus on a simpler task of establishing Theorem 2 only for a small range of $\gamma$ and $\mathcal{L}_{\mathbf{S}}^{\gamma}(w)$. We describe these reductions in Section 3.1 and then proceed to the main arguments in Section 3.2.

## 3.1 Setup

When eventually bounding the Lipschitz constant, as discussed in Section 2, the task turns out to be simpler if $\|x\|_2 = 1$ (and not just $\|x\|_2 \leq 1$) for all $x$ in the support of $\mathcal{D}$ and if $|\langle w, x \rangle| < c_{\gamma}$ for a constant $c_{\gamma}$ sufficiently smaller than 1 for all hypotheses $w$ and data points $(x, y)$ in the support of $\mathcal{D}$. We reduce to this case in Appendix A. The reduction maps every $w \in \mathcal{S}^{d-1}$ to a vector in $\mathcal{H} := \mathcal{S}^{d-1} \times \{0\}$, and every $x$ in the support of $\mathcal{D}$ to a vector in $\mathcal{X}$, where $\mathcal{X}$ is the set of all vectors $x'$ in $\mathcal{S}^{d}$ where the norm of $x'$ without its $(d+1)$'st coordinate is at most $c_{\gamma}$.

From hereon, we let $\mathcal{D}$ be an arbitrary distribution over $\mathcal{X} \times \{-1, 1\}$, and set out to prove that there is a constant $c > 1$, such that with probability at least $1 - \delta$ over $\mathbf{S} \sim \mathcal{D}^n$, it holds for all margins $\gamma \in (n^{-1/2}, c_{\gamma}]$ and all $w \in \mathcal{H}$ that

$$\mathcal{L}_{\mathcal{D}}(w) \leq \mathcal{L}_{\mathbf{S}}^{\gamma}(w) + c \left( \sqrt{\mathcal{L}_{\mathbf{S}}^{\gamma}(w) \cdot \left( \frac{\ln(e/\mathcal{L}_{\mathbf{S}}^{\gamma}(w))}{\gamma^2 n} + \frac{\ln(e/\delta)}{n} \right)} + \frac{\ln(e\gamma^2 n)}{\gamma^2 n} + \frac{\ln(e/\delta)}{n} \right). \quad (16)$$

Theorem 2 follows as a corollary.

**Smaller Tasks.** We now break the task of establishing (16) into smaller tasks, where we consider margins $\gamma$ in a small range $(\gamma_i, \gamma_{i+1}]$ and only vectors $w \in \mathcal{H}$ with $\mathcal{L}_{\mathcal{D}}^{(3/4)\gamma_i}(w)$ in a small range $(\ell_j, \ell_{j+1}]$. The purpose here is, that for one sub-task, we can treat margins and margin losses as the same within constant factors. A union bound over all the sub-tasks then suffices to establish (16).

For a given distribution $\mathcal{D}$, partition the range of values of the margin $\gamma \in (n^{-1/2}, c_{\gamma}]$ into intervals $\Gamma_i = (2^{i-1}n^{-1/2}, 2^i n^{-1/2}]$ for $i = 1, \ldots, \lg_2(c_{\gamma} n^{1/2})$. Similarly, partition the possible values of $\mathcal{L}_{\mathcal{D}}^{\gamma}(w) \in [0, 1]$ into intervals $L_0 = [0, n^{-1}]$ and $L_i = (2^{i-1}n^{-1}, 2^i n^{-1}]$ with $i = 1, \ldots \lg_2 n$.

For a pair $(\Gamma_i, L_j)$ with $\Gamma_i = (\gamma_i, \gamma_{i+1}]$, define

$$\mathcal{H}(\Gamma_i, L_j) = \{w \in \mathcal{H} : \mathcal{L}_{\mathcal{D}}^{(3/4)\gamma_i}(w) \in L_j\}.$$

For each pair $(\Gamma_i, L_j)$ we now prove an equivalent of (16), but tailored to the sub-task. The result is stated in the following lemma

**Lemma 3.** *There is a constant $c > 1$, such that for any $0 < \delta < 1$ and any pair $(\Gamma_i, L_j) = ((\gamma_i, \gamma_{i+1}], (\ell_j, \ell_{j+1}])$, it holds with probability at least $1 - \delta$ over a random sample $\mathbf{S} \sim \mathcal{D}^n$ that*

$$\sup_{\substack{w \in \mathcal{H}(\Gamma_i, L_j) \\ \gamma \in \Gamma_i}} |\mathcal{L}_{\mathcal{D}}(w) - \mathcal{L}_{\mathbf{S}}^{\gamma}(w)| \leq c \left( \sqrt{\ell_{j+1} \left( \frac{\ln(e/\ell_{j+1})}{\gamma_{i+1}^2 n} + \frac{\ln(e/\delta)}{n} \right)} + \frac{\ln(e/\ell_{j+1})}{\gamma_{i+1}^2 n} + \frac{\ln(e/\delta)}{n} \right).$$
$$(17)$$

Observe that while (16) depends on $\gamma$ and (17) depends on $\gamma_{i+1}$, this is fine since $\gamma \leq \gamma_{i+1}$ for all $\gamma \in \Gamma_i$. However, recall that $\mathcal{H}(\Gamma_i, L_j)$ refers to $w \in \mathcal{H}$ with $\mathcal{L}_{\mathcal{D}}^{(3/4)\gamma_i}(w) \in L_j = (\ell_j, \ell_{j+1}]$. But the $\ell_{j+1}$ terms in (17) need to be replaced by $\mathcal{L}_{\mathbf{S}}^{\gamma}(w)$ to obtain (16). Thus we relate the two via the following lemma

**Lemma 4.** *There is a constant $c > 1$, such that for any $0 < \delta < 1$ and any $\Gamma_i = (\gamma_i, \gamma_{i+1}]$, it holds with probability at least $1 - \delta$ over a random sample $\mathbf{S} \sim \mathcal{D}^n$ that*

$$\forall w \in \mathcal{H} : \mathcal{L}_{\mathbf{S}}^{\gamma_i}(w) \geq \frac{\mathcal{L}_{\mathcal{D}}^{(3/4)\gamma_i}(w)}{4} - c \left( \frac{\ln(e\gamma_{i+1}^2 n)}{\gamma_{i+1}^2 n} + \frac{\ln(e/\delta)}{n} \right). \quad (18)$$

We combine the sub-tasks and conclude

**Claim 1.** *For any $0 < \delta < 1$, it holds with probability $1 - \delta$ over $\mathbf{S} \sim \mathcal{D}^n$ that equations* (17) *and* (18) *simultaneously hold for all $(\Gamma_i, L_j)$ and $\Gamma_i$, with slightly different constants c.*

Since Claim 1 follows by a simple union bound, exploiting that for different values of $\ell_{j+1}$ and $\gamma_{i+1}$, we can afford to use different $\delta_{i,j} \approx \delta \exp(-\gamma_{i+1}^2 \ln(e/\ell_{j+1}))$ and $\delta_i \approx \delta \exp(-\gamma_{i+1}^{-2} \ln(e\gamma_{i+1}^2 n))$, we have deferred the proof to Appendix E.

A simple combination of (17) and (18) now gives

**Claim 2.** *For any $0 < \delta < 1$ and training set S, if equations* (17) *and* (18) *hold simultaneously for all $(\Gamma_i, L_j)$ and $\Gamma_i$, then equation* (16) *holds for all $\gamma \in (n^{-1/2}, c_\gamma]$ and all $w \in \mathcal{H}$ for a large enough constant $c > 1$ in* (16).

Claim 2 follows by using that $\gamma \leq \gamma_{i+1}$ for $\gamma \in \Gamma_i$, and by using Lemma 4 to relate all occurrences of $\ell_{j+1}$ in (17) to $\mathcal{L}_S^\gamma(w)$. As this is rather straight forward calculations, we have deferred the proof to Appendix E.

What remains is thus to establish Lemma 3 and Lemma 4, where we may now focus on a small range of $\gamma$ and $\mathcal{L}_{\mathcal{D}}^{(3/4)\gamma_i}(w)$. While both require substantial work and non-trivial arguments, the proof of Lemma 4 follows mostly the previous work by Grønlund et al. [2020a] and has thus been deferred to Appendix D.

## 3.2 Random Discretization

We now set out to prove Lemma 3. So let $0 < \delta < 1$, and fix a pair $(\Gamma_i, L_j)$. Following the proof outline in Section 2, we now consider the following random discretization of hypotheses in $\mathcal{H}(\Gamma_i, L_j)$: Let $k = k(i, j)$ be an integer parameter to be determined. Sample a random $k \times d$ matrix $\mathbf{A}$ with each entry $\mathcal{N}(0, 1/k)$ distributed as well as $k$ random offsets $\mathbf{t} = (\mathbf{t}_1, \ldots, \mathbf{t}_k)$ all independent and uniformly distributed in $[0, 1]$.

Let $\mathcal{G}$ be the set of all vectors in $\mathbb{R}^k$ with coordinates in

$$\{(1/2)(10\sqrt{k})^{-1} + z(10\sqrt{k})^{-1} \mid z \in \mathbb{Z}\}.$$

For $w \in \mathcal{H}$ and an outcome $(A, t)$ of $(\mathbf{A}, \mathbf{t})$, define $h_{A,t}(w) \in \mathcal{G}$ as the vector obtained as follows: Consider each coordinate $(Aw)_i$ and let $z_i$ denote the integer such that

$$(1/2)(10\sqrt{k})^{-1} + z_i(10\sqrt{k})^{-1} \leq (Aw)_i < (1/2)(10\sqrt{k})^{-1} + (z_i + 1)(10\sqrt{k})^{-1}.$$

Let $(h_{A,t}(w))_i$ equal $(1/2)(10\sqrt{k})^{-1} + z_i(10\sqrt{k})^{-1}$ if $t_i \leq p((Aw)_i)$ $((Aw)_i$ rounded down) and otherwise let it equal $(1/2)(10\sqrt{k})^{-1} + (z_i + 1)(10\sqrt{k})^{-1}$. By standard arguments, which we have deferred to Appendix E, we can choose $p((Aw)_i) \in [0, 1]$ such that the expected value of the coordinates satisfy $\mathbb{E}_\mathbf{t}[(h_{A,\mathbf{t}}(w))_i] = (Aw)_i$. The random discretization has the desirable property that it approximately preserves margins/inner products as stated in the following

**Lemma 5.** *There is a constant $c > 0$, such that for any integer $k \geq 1$, $w \in \mathcal{H}, x \in \mathcal{X}$ and any $\gamma \in (0, 1]$, it holds that $\mathbb{P}_{\mathbf{A},\mathbf{t}}[|\langle h_{\mathbf{A},\mathbf{t}}(w), \mathbf{A}x \rangle - \langle w, x \rangle| > \gamma] < c\exp(-\gamma^2 k/c)$.*

The proof of Lemma 5 follows the work by Alon and Klartag [2017] in their work on lower bounds for the Johnson-Lindenstrauss transform, and has thus been deferred to Appendix E. We now observe that

$$\mathcal{L}_{\mathcal{D}}(w) = \mathcal{L}_{\mathbf{A}\mathcal{D}}^{\gamma_i/2}(h_{\mathbf{A},\mathbf{t}}(w)) + \mathbb{P}_{(\mathbf{x},\mathbf{y})\sim\mathcal{D}}[y\langle h_{\mathbf{A},\mathbf{t}}(w), \mathbf{A}\mathbf{x} \rangle > \gamma_i/2 \wedge \mathbf{y}\langle w, \mathbf{x} \rangle \leq 0]$$
$$- \mathbb{P}_{(\mathbf{x},\mathbf{y})\sim\mathcal{D}}[\mathbf{y}\langle h_{\mathbf{A},\mathbf{t}}(w), \mathbf{A}\mathbf{x} \rangle \leq \gamma_i/2 \wedge \mathbf{y}\langle w, \mathbf{x} \rangle > 0].$$

Similarly, we have for $\gamma \in \Gamma_i$ and any training set $S$ that

$$\mathcal{L}_S^\gamma(w) = \mathcal{L}_{\mathbf{A}S}^{\gamma_i/2}(h_{\mathbf{A},\mathbf{t}}(w)) + \mathbb{P}_{(\mathbf{x},\mathbf{y})\sim S}[\mathbf{y}\langle h_{\mathbf{A},\mathbf{t}}(w), \mathbf{A}\mathbf{x} \rangle > \gamma_i/2 \wedge \mathbf{y}\langle w, \mathbf{x} \rangle \leq \gamma]$$
$$- \mathbb{P}_{(\mathbf{x},\mathbf{y})\sim S}[\mathbf{y}\langle h_{\mathbf{A},\mathbf{t}}(w), \mathbf{A}\mathbf{x} \rangle \leq \gamma_i/2 \wedge \mathbf{y}\langle w, \mathbf{x} \rangle > \gamma].$$

We now have for any $\gamma \in \Gamma_i$ that

$$\sup_{w \in \mathcal{H}(\Gamma_i, L_j)} \left( \mathcal{L}_\mathcal{D}(w) - \mathcal{L}_S^\gamma(w) \right) =$$

$$\sup_{w \in \mathcal{H}(\Gamma_i, L_j)} \Big( \mathbb{E}_{\mathbf{A},\mathbf{t}}[\mathcal{L}_{\mathbf{A}\mathcal{D}}^{\gamma_i/2}(h_{\mathbf{A},\mathbf{t}}(w)) - \mathcal{L}_{\mathbf{A}S}^{\gamma_i/2}(h_{\mathbf{A},\mathbf{t}}(w))] +$$

$$\mathbb{E}_{\mathbf{A},\mathbf{t}}[\mathbb{P}_\mathcal{D}[\mathbf{y}\langle h_{\mathbf{A},\mathbf{t}}(w), \mathbf{Ax}\rangle > \gamma_i/2 \wedge \mathbf{y}\langle w, \mathbf{x}\rangle \le 0] - \mathbb{P}_S[\mathbf{y}\langle h_{\mathbf{A},\mathbf{t}}(w), \mathbf{Ax}\rangle > \gamma_i/2 \wedge \mathbf{y}\langle w, \mathbf{x}\rangle \le \gamma]] +$$

$$\mathbb{E}_{\mathbf{A},\mathbf{t}}[\mathbb{P}_S[\mathbf{y}\langle h_{\mathbf{A},\mathbf{t}}(w), \mathbf{Ax}\rangle \le \gamma_i/2 \wedge \mathbf{y}\langle w, \mathbf{x}\rangle > \gamma] - \mathbb{P}_\mathcal{D}[\mathbf{y}\langle h_{\mathbf{A},\mathbf{t}}(w), \mathbf{Ax}\rangle \le \gamma_i/2 \wedge \mathbf{y}\langle w, \mathbf{x}\rangle > 0]] \Big). \tag{19}$$

A critical observation is that the distribution of $y\langle h_{\mathbf{A},\mathbf{t}}(w), \mathbf{A}x\rangle$ depends only on $y\langle w, x\rangle$.

**Claim 3.** *For any $(x, y) \in \mathcal{X} \times \{-1, 1\}$ and any $w \in \mathcal{H}$, the distribution of $y\langle h_{\mathbf{A},\mathbf{t}}(w), \mathbf{A}x\rangle$ is completely determined from $y\langle w, x\rangle$.*

We prove Claim 3 in Appendix B.1 by exploiting that the entries of $\mathbf{A}$ are i.i.d. $\mathcal{N}(0, 1/k)$ distributed and using the rotational invariance of the Gaussian distribution.

As outlined in the proof overview in Section 2, we can now use Claim 3 together with the contraction inequality of Rademacher complexity to bound several of the terms in (19). Similarly to the introduction of the ramp loss in classic proofs of generalization for large-margin halfspaces, we need to introduce a continuous function upper bounding the probabilities above. With this in mind, we now define the following functions $\phi$ and $\rho$:

$$\phi(\alpha) = \begin{cases} \mathbb{P}_{\mathbf{A},\mathbf{t}}[y\langle h_{\mathbf{A},\mathbf{t}}(w), \mathbf{A}x\rangle > \gamma_i/2 \mid y\langle w, x\rangle = \alpha] & \text{if } -c_\gamma \le \alpha \le 0 \\ \frac{(\gamma_i - \alpha)}{\gamma_i} \mathbb{P}_{\mathbf{A},\mathbf{t}}[y\langle h_{\mathbf{A},\mathbf{t}}(w), \mathbf{A}x\rangle > \gamma_i/2 \mid y\langle w, x\rangle = 0] & \text{if } 0 < \alpha \le \gamma_i \\ 0 & \text{if } \gamma_i < \alpha \le c_\gamma \end{cases}$$

$$\rho(\alpha) = \begin{cases} \mathbb{P}_{\mathbf{A},\mathbf{t}}[y\langle h_{\mathbf{A},\mathbf{t}}(w), \mathbf{A}x\rangle \le \gamma_i/2 \mid y\langle w, x\rangle = \alpha] & \text{if } \gamma_i < \alpha \le c_\gamma \\ \frac{\alpha}{\gamma_i} \mathbb{P}_{\mathbf{A},\mathbf{t}}[y\langle h_{\mathbf{A},\mathbf{t}}(w), \mathbf{A}x\rangle \le \gamma_i/2 \mid y\langle w, x\rangle = \gamma_i] & \text{if } 0 < \alpha \le \gamma_i \\ 0 & \text{if } -c_\gamma \le \alpha \le 0 \end{cases}.$$

Here we slightly abuse notation and write $\mathbb{P}_{\mathbf{A},\mathbf{t}}[y\langle h_{\mathbf{A},\mathbf{t}}(w), \mathbf{A}x\rangle > \gamma_i/2 \mid y\langle w, x\rangle = \alpha]$ to denote the probability $\mathbb{P}_{\mathbf{A},\mathbf{t}}[y\langle h_{\mathbf{A},\mathbf{t}}(w), \mathbf{A}x\rangle > \gamma_i/2]$ for an arbitrary $w \in \mathcal{H}, (x, y) \in \mathcal{X} \times \{-1, 1\}$ with $y\langle w, x\rangle = \alpha$ and remark that this probability is the same for all such $w, x, y$ by Claim 3.

We now observe that $\phi$ and $\rho$ upper and lower bounds the terms in (19)

**Remark 6.** *For any training set $S$ and distribution $\mathcal{D}$ over $\mathcal{X} \times \{-1, 1\}$, we have*

$$\mathbb{E}_{\mathbf{A},\mathbf{t}}[\mathbb{P}_{(\mathbf{x},\mathbf{y})\sim\mathcal{D}}[\mathbf{y}\langle h_{\mathbf{A},\mathbf{t}}(w), \mathbf{Ax}\rangle > \gamma_i/2 \wedge \mathbf{y}\langle w, \mathbf{x}\rangle \le 0]] \le \mathbb{E}_{(\mathbf{x},\mathbf{y})\sim\mathcal{D}}[\phi(\mathbf{y}\langle w, \mathbf{x}\rangle)]$$

$$\mathbb{E}_{\mathbf{A},\mathbf{t}}[\mathbb{P}_{(\mathbf{x},\mathbf{y})\sim S}[\mathbf{y}\langle h_{\mathbf{A},\mathbf{t}}(w), \mathbf{Ax}\rangle > \gamma_i/2 \wedge \mathbf{y}\langle w, \mathbf{x}\rangle \le \gamma]] \ge \mathbb{E}_{(\mathbf{x},\mathbf{y})\sim S}[\phi(\mathbf{y}\langle w, \mathbf{x}\rangle)]$$

$$\mathbb{E}_{\mathbf{A},\mathbf{t}}[\mathbb{P}_{(\mathbf{x},\mathbf{y})\sim S}[\mathbf{y}\langle h_{\mathbf{A},\mathbf{t}}(w), \mathbf{Ax}\rangle \le \gamma_i/2 \wedge \mathbf{y}\langle w, \mathbf{x}\rangle > \gamma]] \le \mathbb{E}_{(\mathbf{x},\mathbf{y})\sim S}[\rho(\mathbf{y}\langle w, \mathbf{x}\rangle)]$$

$$\mathbb{E}_{\mathbf{A},\mathbf{t}}[\mathbb{P}_{(\mathbf{x},\mathbf{y})\sim\mathcal{D}}[\mathbf{y}\langle h_{\mathbf{A},\mathbf{t}}(w), \mathbf{Ax}\rangle \le \gamma_i/2 \wedge \mathbf{y}\langle w, \mathbf{x}\rangle > 0]] \ge \mathbb{E}_{(\mathbf{x},\mathbf{y})\sim\mathcal{D}}[\rho(\mathbf{y}\langle w, \mathbf{x}\rangle)].$$

The proof of Remark 6 follows from the definition of $\phi$ and $\rho$, along with monotonicity of $\mathbb{P}_{\mathbf{A},\mathbf{t}}[y\langle h_{\mathbf{A},\mathbf{t}}(w), \mathbf{A}x\rangle > \gamma_i \mid y\langle w, x\rangle = \alpha]$ as a function of $\alpha$. The proofs have been deferred to Appendix E. Continuing from (19) using Remark 6, linearity of expectation and the triangle inequality, we have for any $\gamma \in \Gamma_i$ that

$$\sup_{w \in \mathcal{H}(\Gamma_i, L_j)} \mathcal{L}_\mathcal{D}(w) - \mathcal{L}_S^\gamma(w) \le \sup_{w \in \mathcal{H}(\Gamma_i, L_j)} \left| \mathbb{E}_{\mathbf{A},\mathbf{t}}[\mathcal{L}_\mathcal{D}^{\gamma_i/2}(h_{\mathbf{A},\mathbf{t}}(w)) - \mathcal{L}_S^{\gamma_i/2}(h_{\mathbf{A},\mathbf{t}}(w))] \right| \tag{20}$$

$$+ \sup_{w \in \mathcal{H}(\Gamma_i, L_j)} \left| \mathbb{E}_{(\mathbf{x},\mathbf{y})\sim\mathcal{D}}[\phi(\mathbf{y}\langle w, \mathbf{x}\rangle)] - \mathbb{E}_{(\mathbf{x},\mathbf{y})\sim S}[\phi(\mathbf{y}\langle w, \mathbf{x}\rangle)] \right| \tag{21}$$

$$+ \sup_{w \in \mathcal{H}(\Gamma_i, L_j)} \left| \mathbb{E}_{(\mathbf{x},\mathbf{y})\sim\mathcal{D}}[\rho(\mathbf{y}\langle w, \mathbf{x}\rangle)] - \mathbb{E}_{(\mathbf{x},\mathbf{y})\sim S}[\rho(\mathbf{y}\langle w, \mathbf{x}\rangle)] \right|. \tag{22}$$

In Appendix C, we carefully use Bernstein's plus a (highly non-trivial) union bound over infinitely many grids of increasing size to bound (20) as follows

**Lemma 7.** *There is a constant $c > 0$ such that with probability at least $1 - \delta$ over $\mathbf{S} \sim \mathcal{D}^n$ we have*

$$(20) \leq c \left( \sqrt{\frac{(\ell_{j+1} + \exp(-\gamma_{i+1}^2 k/c))(k + \ln(e/\delta))}{n}} + \frac{(k + \ln(e/\delta))}{n} \right).$$

In Appendix B, we then use Rademacher complexity and a bound on the Lipschitz constants of $\phi$ and $\rho$ to bound (21) and (22) as follows

**Lemma 8.** *There are constants $c, c' > 0$ such that when $k \geq c'\gamma_{i+1}^{-2}$, it holds with probability at least $1 - \delta$ over $\mathbf{S} \sim \mathcal{D}^n$ that*

$$\max\{(21),(22)\} \leq c \exp(-\gamma_{i+1}^2 k/c) \cdot \sqrt{(k + \gamma_{i+1}^{-2} + \ln(e/\delta))/n}.$$

To balance the expressions in Lemma 7 and Lemma 8, we now set $k = c\gamma_{i+1}^{-2} \ln(e/\ell_{j+1})$ for a sufficiently large constant $c > 0$ so that $\exp(-\gamma_{i+1}^2 k/c) \leq \ell_{j+1}/e$ and $k \geq c'\gamma_{i+1}^{-2}$. Combining Lemma 7 and Lemma 8 via a union bound with $\delta' = \delta/2$ and inserting into (20), (21) and (22) gives

$$\sup_{w \in \mathcal{H}(\Gamma_i, L_j)} \mathcal{L}_{\mathcal{D}}(w) - \mathcal{L}_S^{\gamma}(w) \leq c \left( \ell_{j+1} \sqrt{(\gamma_{i+1}^{-2} \ln(e/\ell_{j+1}) + \ln(e/\delta))/n} \right)$$

$$+ c \left( \sqrt{\frac{\ell_{j+1}(\gamma_{i+1}^{-2} \ln(e/\ell_{j+1}) + \ln(e/\delta))}{n}} + \frac{\gamma_{i+1}^{-2} \ln(e/\ell_{j+1}) + \ln(e/\delta)}{n} \right),$$

for a constant $c > 0$. This completes the proof of Lemma 3, which together with Lemma 4 completes the proof of our main result, Theorem 2.

## 4 Conclusion

We have established the first asymptotically tight generalization bound for large-margin halfspaces, resolving a long-standing gap between upper and lower bounds in this fundamental setting. Our main theorem precisely characterizes the interplay between the margin, empirical margin loss, sample size, and confidence parameter. The proof introduces several new analytical techniques, including a refined initial analysis and a Rademacher-based treatment of randomized rounding.

Beyond settling the generalization theory of large margin halfspaces, our framework provides novel tools that may prove useful for deriving tight bounds in related models. Concretely, our techniques are in essence a refinement of the techniques introduced by Schapire et al. [1998] for proving generalization of large margin voting classifiers. For voting classifiers, the best known generalization upper (Gao and Zhou [2013]) and lower bounds (Grønlund et al. [2020b]) for finite hypothesis sets have a similar logarithmic gap as our techniques managed to remove for halfspaces. Another related topic is kernel methods, in particular in the context of support vector machines. Here there are also prior works deriving generalization bounds based on margins, see e.g. Cortes et al. [2010], however these works are not "first-order bounds", i.e. the $\sqrt{\cdot}$ terms in the generalization bounds do not decrease with $\mathcal{L}_S^{\gamma}(w)$. We are hopeful that our techniques may also yield improvements in these areas of interest. In particular, it is worth mentioning that the original Johnson-Lindenstrauss transform Johnson and Lindenstrauss [1984] also provides dimensionality reduction from infinite-dimensional Hilbert spaces (e.g. kernel space), thus suggesting that a similar randomized discretization might be possible.

## Acknowledgments and Disclosure of Funding

The authors would like to thank Clement Svendsen for valuable measure theoretic insight.

Kasper Green Larsen is co-funded by a DFF Sapere Aude Research Leader Grant No. 9064-00068B by the Independent Research Fund Denmark and co-funded by the European Union (ERC, TUCLA, 101125203). Natascha Schalburg is funded by the European Union (ERC, TUCLA, 101125203). Views and opinions expressed are however those of the author(s) only and do not necessarily reflect those of the European Union or the European Research Council. Neither the European Union nor the granting authority can be held responsible for them.

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

# A  Simplifying the Setting

As is usual in analysis of halfspace classifiers, we restrict to data $x$ of at most unit norm and normalized halfspaces $w$ without bias. Our main result 2 is stated under this restriction. The more general result, for non-homogenous halfspaces and data $x$ of any norm, stated below, follows from the homogeneous case.

**Theorem 9.** *There is a constant $c > 0$ such that for any radius $R > 0$ and distribution $\mathcal{D}$ over $R \cdot \mathbb{B}_2^d \times \{-1, 1\}$, it holds with probability at least $1 - \delta$ over $\mathbf{S} \sim \mathcal{D}^n$ that for every $w \in \mathcal{S}^{d-1}, b \in \mathbb{R}$ and every margin $Rn^{-1/2} \leq \gamma \leq R$, we have*

$$\mathcal{L}_\mathcal{D}(w, b) \leq \mathcal{L}_{\mathbf{S}}^\gamma(w, b)$$
$$+ c\left(\sqrt{\mathcal{L}_{\mathbf{S}}^\gamma(w, b) \cdot \left(\frac{R^2 \ln(e/\mathcal{L}_{\mathbf{S}}^\gamma(w, b))}{\gamma^2 n} + \frac{\ln(e/\delta)}{n}\right)} + \frac{R^2 \ln(e\gamma^2 n/R^2)}{\gamma^2 n} + \frac{\ln(e/\delta)}{n}\right),$$

*where $\mathcal{L}_\mathcal{D}(w, b) := \mathbb{P}_{(\mathbf{x}, \mathbf{y}) \sim \mathcal{D}}[\text{sign}(\langle w, \mathbf{x}\rangle + b) \neq \mathbf{y}]$ and $\mathcal{L}_S^\gamma(w, b) := |\{(x, y) \in S : y(\langle w, x\rangle + b) \leq \gamma\}|/|S|$ for a set of samples $S$.*

*Proof.* First observe that for points in the ball of radius $R$, any halfspace $(w, b)$ with $|b| > R$ makes the same classifications as the halfspace $(w, \text{sign}(b)R)$. We thus assume wlog. that $|b| \leq R$. Let $\mathcal{D}$ be a distribution over $R \cdot \mathbb{B}_2^d \times \{-1, 1\}$. Map any sample $(x, y)$ in the support of $\mathcal{D}$ to the sample $(x', y)$ where $x'$ is obtained by prepending a coordinate hardcoded to $R$ and then scaling the resulting vector by $1/\sqrt{2R^2}$. Similarly, map any non-homogeneous halfspace $(w, b) \in \mathcal{S}^{d-1} \times [-R, R]$ to the homogeneous halfspace $w'$ obtained by prepending a coordinate to $w$ that is hardcoded to $b/R$ and then scaling the resulting vector by $1/\sqrt{(b/R)^2 + 1}$. We have $\|w'\| = 1$ and $\|x'\| \leq 1$. Furthermore $\langle w', x'\rangle = (2R^2((b/R)^2 + 1))^{-1/2}(b + \langle w, x\rangle)$. It follows that $w'$ makes the same prediction on $x'$ as $(w, b)$ does on $x$. Furthermore, the resulting margin is at least a factor $(2R^2((b/R)^2 + 1))^{-1/2} \geq 1/(2R)$ of the original margin. Theorem 9 now follows from Theorem 2 by replacing all occurrences of $\gamma$ by $\gamma/(2R)$. $\qquad\square$

In the main proof we reduce to a setting more favorable for analysis, here we give the details for the reduction, and the argument that no generality was lost.

**Observation 1.** *Without loss of generality, we can assume that the vectors $x$ in the support of $\mathcal{D}$ have unit norm and all margins lie in the range $[-c_\gamma, c_\gamma]$ for a constant $0 < c_\gamma < 1$.*

*Proof.* Consider the following distribution $\mathcal{D}'$ obtained by sampling an $(\mathbf{x}, \mathbf{y}) \sim \mathcal{D}$ and replacing $\mathbf{x}$ by $\mathbf{x}' = (c_\gamma \mathbf{x}) \times \{\sqrt{1 - c_\gamma^2 \|\mathbf{x}\|_2^2}\} \in \mathcal{S}^d$ for a sufficiently small constant $0 < c_\gamma < 1$. That is, scale down all coordinates of $\mathbf{x}$ by $c_\gamma$ and append a $(d + 1)$'st coordinate taking the value $\sqrt{1 - c_\gamma^2 \|\mathbf{x}\|_2^2}$. Then the norm of the resulting point $\mathbf{x}'$ is $\sqrt{c_\gamma^2 \|\mathbf{x}\|_2^2 + 1 - c_\gamma^2 \|\mathbf{x}\|_2^2} = 1$. Similarly, for any $w \in \mathcal{S}^{d-1}$, consider instead the hypothesis $w' = w \times \{0\}$. We observe that for any $x, w$, we have that $\langle w', x'\rangle = \langle w, c_\gamma x\rangle = c_\gamma \langle w, x\rangle$ and thus lies in the range $[-c_\gamma, c_\gamma]$ by Cauchy-Schwartz. This also implies that $\text{sign}(\langle w', x'\rangle) = \text{sign}(\langle w, x\rangle)$ and thus the generalization error of $w'$ under $\mathcal{D}'$ and $w$ under $\mathcal{D}$ are the same.

Theorem 2 follows as an immediate corollary of (16), since margins change by a $c_\gamma$ factor in our transformation of the input distribution. Since $c_\gamma$ is a constant, this disappears in the constant factor $c$ in Theorem 2 (note that for margins $\gamma \in [n^{-1/2}, c_\gamma^{-1} n^{-1/2})$ in Theorem 2, we cannot use the reduction, but here Theorem 2 follows trivially as $c \ln(e\gamma^2 n)/(\gamma^2 n) > 1$ for sufficiently large $c$). $\quad\square$

# B  Rademacher Bounds

In this section, we use Rademacher complexity and the contraction inequality to prove Lemma 8. We focus on bounding (21) and note that (22) is handled symmetrically.

For a training set $S \in (\mathcal{X} \times \{-1, 1\})^n$, consider the empirical Rademacher complexity (for $\sigma = (\sigma_1, \ldots, \sigma_n)$ a vector of independent and uniform variables in $\{-1, 1\}$):

$$\hat{\mathcal{R}}_{\phi, \mathcal{H}(\Gamma_i, L_j)}(S) = \frac{1}{n} \cdot \mathbb{E}_\sigma \left[ \sup_{w \in \mathcal{H}(\Gamma_i, L_j)} \sum_{(x_i, y_i) \in S} \sigma_i \phi(y_i \langle w, x_i \rangle) \right]$$

$$\leq \frac{1}{n} \cdot \mathbb{E}_\sigma \left[ \sup_{w \in \mathcal{H}} \sum_{(x_i, y_i) \in S} \sigma_i \phi(y_i \langle w, x_i \rangle) \right].$$

If $\phi$ is $L_\phi$-Lipschitz, then the contraction inequality from Ledoux and Talagrand [1991] gives that

$$\hat{\mathcal{R}}_{\phi, \mathcal{H}}(S) \leq \frac{L_\phi}{n} \cdot \mathbb{E}_\sigma \left[ \sup_{w \in \mathcal{H}} \sum_{(x_i, y_i) \in S} \sigma_i y_i \langle w, x_i \rangle \right].$$

Using Cauchy-Schwartz, this is bounded by

$$\hat{\mathcal{R}}_{\phi, \mathcal{H}}(S) \leq \frac{L_\phi}{n} \cdot \mathbb{E}_\sigma \left[ \sup_{w \in \mathcal{H}} \left\langle w, \sum_{(x_i, y_i) \in S} \sigma_i y_i x_i \right\rangle \right]$$

$$\leq \frac{L_\phi}{n} \cdot \left( \sup_{w \in \mathcal{H}} \|w\|_2 \right) \cdot \mathbb{E}_\sigma \left[ \left\| \sum_{(x_i, y_i) \in S} \sigma_i y_i x_i \right\|_2 \right]$$

$$\leq \frac{L_\phi}{n} \cdot \sqrt{\mathbb{E}_\sigma \left[ \left\| \sum_{(x_i, y_i) \in S} \sigma_i y_i x_i \right\|_2^2 \right]}$$

$$= \frac{L_\phi}{\sqrt{n}} \cdot \sqrt{\sum_{(x_i, y_i) \in S} \sum_{(x_j, y_j) \in S} \mathbb{E}_\sigma[\sigma_i \sigma_j] y_i y_j \langle x_i, x_j \rangle}$$

$$= \frac{L_\phi}{\sqrt{n}}.$$

Since this inequality holds for all $S$ with each $(x, y) \in S$ satisfying $\|x\|_2 = 1$, we have for the distribution $\mathcal{D}$ that the Rademacher complexity

$$\mathcal{R}_{\mathcal{D}, \phi, \mathcal{H}}(n) = \mathbb{E}_{\mathbf{S} \sim \mathcal{D}^n}[\hat{\mathcal{R}}_{\phi, \mathcal{H}}(\mathbf{S})],$$

satisfies $\mathcal{R}_{\mathcal{D}, \phi, \mathcal{H}}(n) \leq L_\phi / \sqrt{n}$. By Lemma 5 and $\gamma_i = \gamma_{i+1}/2$, we have that $\phi$ is bounded by

$$0 \leq \phi(\alpha) \leq \max_{-c_\gamma \leq \alpha \leq 0} \mathbb{P}_{\mathbf{A}, \mathbf{t}}[y \langle h_{\mathbf{A}, \mathbf{t}}(w), \mathbf{A}x \rangle > \gamma_i/2 \mid y \langle w, x \rangle = \alpha] \leq c \exp(-k \gamma_{i+1}^2/c),$$

for a constant $c > 0$. We conclude from standard results on Rademacher complexity (see e.g. Shalev-Shwartz and Ben-David [2014]), that with probability $1 - \delta$ over a sample $\mathbf{S} \sim \mathcal{D}^n$ it holds that

$$\sup_{w \in \mathcal{H}(\Gamma_i, L_j)} \left| \mathbb{E}_{(\mathbf{x}, \mathbf{y}) \sim \mathcal{D}}[\phi(\mathbf{y} \langle w, \mathbf{x} \rangle)] - \mathbb{E}_{(\mathbf{x}, \mathbf{y}) \sim \mathbf{S}}[\phi(\mathbf{y} \langle w, \mathbf{x} \rangle)] \right| \leq$$

$$2\mathcal{R}_{\mathcal{D}, \phi, \mathcal{H}}(n) + c_R \left( c \exp(-k \gamma_{i+1}^2/c) \sqrt{\frac{\ln(1/\delta)}{n}} \right) \leq$$

$$\frac{2L_\phi}{\sqrt{n}} + c_R \left( c \exp(-k \gamma_{i+1}^2/c) \sqrt{\frac{\ln(1/\delta)}{n}} \right).$$

where $c_R > 0$ is a constant. Symmetric arguments bounds $\rho$ by the same, with the Lipschitz constant $L_\rho$ of $\rho$ in place of $L_\phi$.

We now use the following bound on the Lipschitz constants of $\phi$ and $\rho$

**Lemma 10.** *There are constants $c_L, c > 0$ such that the Lipschitz constants $L_\phi$ and $L_\rho$ of $\phi$ and $\rho$ are bounded by*

$$c_L \exp(-\gamma_{i+1}^2 k/c_L) \cdot \left(\sqrt{k} + \gamma_{i+1}^{-1}\right),$$

*when $k \geq c\gamma_{i+1}^{-2}$.*

We prove this lemma in the next section. We thus conclude that with probability at least $1 - \delta$ over $\mathbf{S} \sim \mathcal{D}^n$, we have

$$\sup_{w \in \mathcal{H}(\Gamma_i, L_j)} \left|\mathbb{E}_{(\mathbf{x}, \mathbf{y}) \sim \mathcal{D}}[\phi(\mathbf{y}\langle w, \mathbf{x}\rangle)] - \mathbb{E}_{(\mathbf{x}, \mathbf{y}) \sim \mathbf{S}}[\phi(\mathbf{y}\langle w, \mathbf{x}\rangle)]\right| \leq$$

$$2 \cdot \frac{c_L \exp(-\gamma_{i+1}^2 k/c_L)\left(\sqrt{k} + \gamma_{i+1}^{-1}\right)}{\sqrt{n}} + c_R \left(c \exp(-k\gamma_{i+1}^2/c)\sqrt{\frac{\ln(1/\delta)}{n}}\right).$$

The same bound holds for $\rho$ via Lemma 10, which completes the proof of Lemma 8.

## B.1 Bounding the Lipschitz Constants

In this section, we proceed to bound the Lipschitz constants of $\phi$ and $\rho$ and thereby prove Lemma 10. We split it into two tasks depending on the value of $\alpha$. The simplest case is the following

**Lemma 11.** *There is a constant $c > 0$ such that the Lipschitz constants of $\phi$ and $\rho$, when $0 < \alpha \leq \gamma_i$, are less than:*

$$\frac{c \exp(-k\gamma_{i+1}^2/c)}{\gamma_{i+1}}.$$

*Proof.* Since $\phi$ is linear when $0 < \alpha \leq \gamma_i$, its Lipschitz constant equals the slope of the line, i.e.

$$\frac{1}{\gamma_i} \cdot \mathbb{P}_{\mathbf{A}, \mathbf{t}}[y\langle h_{\mathbf{A}, \mathbf{t}}(w), \mathbf{A}x\rangle > \gamma_i/2 \mid y\langle w, x\rangle = 0].$$

By Lemma 5 and using $\gamma_i = \gamma_{i+1}/2$, this is bounded by

$$\frac{c \exp(-\gamma_{i+1}^2 k/c)}{\gamma_{i+1}},$$

for a constant $c > 0$. The same arguments applies immediately to $\rho$. $\qquad\square$

The trickier case is when $\alpha \in [-c_\gamma, 0]$ for $\phi$ and when $\alpha \in (\gamma_i, c_\gamma]$ for $\rho$. If we set $c_\gamma \leq 1/\sqrt{2}$, then we have

**Lemma 12.** *There is a constant $c > 0$ such that the Lipschitz constant of $\phi$ when $\alpha \in [-1/\sqrt{2}, 0]$ and $\rho$ when $\alpha \in (\gamma_i, 1/\sqrt{2}]$ is less than*

$$c \exp\left(-\gamma_{i+1}^2 k/c\right) \sqrt{k},$$

*for $k \geq c\gamma_{i+1}^{-2}$.*

Combining this result with Lemma 11 completes the proof of Lemma 10.

To prove Lemma 12, we need to bound the Lipschitz constants of $\phi$ when $\alpha \in [-1/\sqrt{2}, 0]$ and $\rho$ when $\alpha \in (\gamma_i, 1/\sqrt{2}]$. We will go through the details for $\phi$, and comment how the argument for $\rho$ differs along the way.

First recall the following claim

**Restatement of Claim 3.** *For any $(x, y) \in \mathcal{X} \times \{-1, 1\}$ and any $w \in \mathcal{H}$, the distribution of $y\langle h_{\mathbf{A}, \mathbf{t}}(w), \mathbf{A}x\rangle$ is completely determined from $y\langle w, x\rangle$.*

As we need to understand the distribution of the random variable $y\langle h_{\mathbf{A}, \mathbf{t}}(w), \mathbf{A}x\rangle$ to bound the Lipschitz constants of $\phi$ and $\rho$, we proceed to give the proof of Claim 3 while introducing convenient notation for establishing Lemma 12.

*Proof.* Firstly, write $h_{\mathbf{A},\mathbf{t}}(w) = \mathbf{A}w + \mathbf{v}$ with $\mathbf{v} = (h_{\mathbf{A},\mathbf{t}}(w) - \mathbf{A}w)$. Then observe that $(\mathbf{A}w)_i = \langle \mathbf{a}_i, w \rangle \sim \mathcal{N}(0, \|w\|^2/k) \stackrel{d}{=} \mathcal{N}(0, 1/k)$ where $\mathbf{a}_i$ denotes the $i$'th row of $\mathbf{A}$. Here $\stackrel{d}{=}$ denotes equality in distribution. Now write $x = \langle w, x \rangle w + u$ where $\langle u, w \rangle = 0$ and $\|u\|^2 = \|x\|^2 - \langle w, x \rangle^2 = 1 - \langle w, x \rangle^2$ (i.e. a Gram-Schmidt step). We have $(\mathbf{A}x)_i = \langle \mathbf{a}_i, w \rangle \langle w, x \rangle + \langle \mathbf{a}_i, u \rangle$. By rotational invariance of the Gaussian distribution and orthogonality of $w$ and $u$, we have that $\langle \mathbf{a}_i, u \rangle \sim \mathcal{N}(0, (1 - \langle w, x \rangle^2)/k)$ and that this is independent of $\langle \mathbf{a}_i, w \rangle$. Using the independence, we also conclude that if we condition on any fixed outcome of $c_i = (\mathbf{A}w)_i$, we have that $c_i \langle \mathbf{a}_i, u \rangle$ is $\mathcal{N}(0, c_i^2(1 - \langle w, x \rangle^2)/k)$ distributed.

We now argue that we can sample from the distribution of $y \langle h_{\mathbf{A},\mathbf{t}}(w), \mathbf{A}x \rangle = \langle h_{\mathbf{A},\mathbf{t}}(w), y\mathbf{A}x \rangle$ knowing only $y \langle w, x \rangle$ as follows: Sample independent $\mathcal{N}(0, 1/k)$ distributed random variables $\mathbf{X}_1, \ldots, \mathbf{X}_k$. Next sample independent $\mathcal{N}(0, (1 - y^2 \langle w, x \rangle^2)/k)$ distributed random variables $\mathbf{Y}_1, \ldots, \mathbf{Y}_k$ and let $\mathbf{Z}_i = y \langle w, x \rangle \mathbf{X}_i + \mathbf{Y}_i \stackrel{d}{=} y \langle w, x \rangle \mathbf{X}_i + y\mathbf{Y}_i$, where the last step follows from independence of $\mathbf{X}_i$ and $\mathbf{Y}_i$ and symmetry in the distribution of $\mathbf{Y}_i$. Let $\mathbf{X}$ be the vector with the $\mathbf{X}_i$'s as entries and $\mathbf{Z}$ similarly. Then the joint distribution of $(\mathbf{X}, \mathbf{Z})$ is equal to the joint distribution of $(\mathbf{A}w, y\mathbf{A}x)$. Finally draw offsets $\mathbf{t}'_1, \ldots, \mathbf{t}'_k$ uniformly and independently in $[0, 1]$ and round $\mathbf{X}_i$ to a number of the form $(1/2)(10\sqrt{k})^{-1} + z(10\sqrt{k})^{-1}$ for $z \in \mathbb{Z}$ as in the definition of $h_{\mathbf{A},\mathbf{t}}$. The resulting variables $\mathbf{X}'_i$ satisfy that $y \langle h_{\mathbf{A},\mathbf{t}}(w), \mathbf{A}x \rangle \stackrel{d}{=} \langle \mathbf{X}', \mathbf{Z} \rangle$. $\qquad\square$

With Claim 3 established, we will use the notation in the proof as we proceed with bounding the Lipschitz constants of $\phi$ and $\rho$.

Let $\alpha = y \langle w, x \rangle$ for some $w \in \mathcal{H}$ and $(x, y) \in \mathcal{X} \times \{-1, 1\}$ and $\alpha \in [-1/\sqrt{2}, 0]$ (for $\rho$, let $\alpha \in (\gamma_i, 1/\sqrt{2}]$). Let $\mathbf{X}_i \sim \mathcal{N}(0, 1/k)$, $\mathbf{Y}_i \sim \mathcal{N}(0, (1 - \alpha^2)/k)$ and let $\mathbf{X}'_i$ be the random rounding of $\mathbf{X}_i$. We argued, in the proof of Claim 3, that $y \langle h_{\mathbf{A},\mathbf{t}}(w), \mathbf{A}x \rangle \stackrel{d}{=} \langle \mathbf{X}', \alpha\mathbf{X} + \mathbf{Y} \rangle$. Let additionally $E_i$ be the event that $\mathbf{X}'_i$ is rounded up. For notational convenience, let $\mathbf{M}_i = \sqrt{k}\mathbf{X}_i$ and observe that $\mathbf{M}_i \sim \mathcal{N}(0, 1)$. With this notation, we have that $\mathbf{X}'_i$ has the form

$$\mathbf{X}'_i = \frac{1}{10\sqrt{k}} \left( \left\lfloor 10\mathbf{M}_i - \frac{1}{2\sqrt{k}10} \right\rfloor + 1\{E_i\} + \frac{1}{2} \right).$$

Hence,

$$y \langle h_{\mathbf{A},\mathbf{t}}(w), \mathbf{A}x \rangle \stackrel{d}{=} \langle \mathbf{X}', \alpha\mathbf{X} + \mathbf{Y} \rangle = \frac{\alpha}{\sqrt{k}} \langle \mathbf{X}', \mathbf{M} \rangle + \langle \mathbf{X}', \mathbf{Y} \rangle.$$

Recall that the variables $\mathbf{t}_i$ and $\mathbf{M}_i$ determine $\mathbf{X}_i$, $E_i$ and thus also $\mathbf{X}'_i$. If we condition on an outcome $\mathbf{t}_i = t_i$ and $\mathbf{M}_i = M_i$, only $\mathbf{Y}_i$ remains random. We may thus write

$$\mathbb{P}[y \langle h_{\mathbf{A},\mathbf{t}}(w), \mathbf{A}x \rangle > \gamma_i/2] =$$
$$\mathbb{P}\left[ \frac{\alpha}{\sqrt{k}} \langle \mathbf{X}', \mathbf{M} \rangle + \langle \mathbf{X}', \mathbf{Y} \rangle > \gamma_i/2 \right] =$$
$$\int_{\mathbb{R}^k \times [0,1]^k} f_{\mathbf{M},\mathbf{t}}(M, t) \mathbb{P}\left[ \frac{\alpha}{\sqrt{k}} \langle \mathbf{X}', \mathbf{M} \rangle + \langle \mathbf{X}', \mathbf{Y} \rangle > \gamma_i/2 \;\middle|\; \mathbf{M}_i = M_i, \mathbf{t}_i = t_i \right] d(M, t) =$$
$$\int_{\mathbb{R}^k \times [0,1]^k} f_{\mathbf{M},\mathbf{t}}(M, t) \mathbb{P}\left[ \frac{\alpha}{\sqrt{k}} \langle X', M \rangle + \langle X', \mathbf{Y} \rangle > \gamma_i/2 \right] d(M, t),$$

where $f_{M,t}(M, t)$ is the joint probability density function of $\mathbf{M}$ and $\mathbf{t}$.

Let us now define $\mathbf{N}_i$ such that $\mathbf{Y}_i = \sqrt{\frac{1-\alpha^2}{k}}\mathbf{N}_i$ and let $\mathbf{N} = (\mathbf{N}_1, \ldots, \mathbf{N}_k)$. Then $\mathbf{N}_i \sim \mathcal{N}(0, 1)$ and the event

$$\frac{\alpha}{\sqrt{k}} \langle X', M \rangle + \langle X', \mathbf{Y} \rangle > \gamma_i/2,$$

may be rewritten as

$$\frac{\alpha}{\sqrt{k}}\langle X', M\rangle + \langle X', \mathbf{Y}\rangle > \gamma_i/2 \iff \langle X', \mathbf{Y}\rangle > \gamma_i/2 - \frac{\alpha}{\sqrt{k}}\langle X', M\rangle \iff$$

$$\sqrt{\frac{1-\alpha^2}{k}}\langle X', \mathbf{N}\rangle > \gamma_i/2 - \frac{\alpha}{\sqrt{k}}\langle X', M\rangle \iff$$

$$\sqrt{\frac{1-\alpha^2}{k}}\|X'\|_2\langle X'/\|X'\|_2, \mathbf{N}\rangle > \gamma_i/2 - \frac{\alpha}{\sqrt{k}}\langle X', M\rangle \iff$$

$$\langle X'/\|X'\|_2, \mathbf{N}\rangle > \frac{\sqrt{k}\gamma_i/2 - \alpha\langle X', M\rangle}{\sqrt{1-\alpha^2}\|X'\|_2}.$$

Observe that $\langle X'/\|X'\|_2, \mathbf{N}\rangle \sim \mathcal{N}(0,1)$. If we let $\Phi$ denote the cumulative density function of a standard normal distribution, then we have established

$$\mathbb{P}[y\langle h_{\mathbf{A},\mathbf{t}}(w), \mathbf{A}x\rangle > \gamma_i/2] = \int_{\mathbb{R}^k \times [0,1]^k} f_{M,t}(M,t)\left(1 - \Phi\left(\frac{\sqrt{k}\gamma_i/2 - \alpha\langle X', M\rangle}{\sqrt{1-\alpha^2}\|X'\|_2}\right)\right) d(M,t)$$

$$= \int_{\mathbb{R}^k \times [0,1]^k} f_{\mathbf{M}}(M)\left(1 - \Phi\left(\frac{\sqrt{k}\gamma_i/2 - \alpha\langle X', M\rangle}{\sqrt{1-\alpha^2}\|X'\|_2}\right)\right) d(M,t).$$
(23)

In the last equality, we use that $\mathbf{M}$ and $\mathbf{t}$ are independent and that the probability density function of $\mathbf{t}$ is 1 since each $\mathbf{t}_i$ is uniform in $[0,1]$. This reduces $f_{M,t}(M,t)$ to the probability density function $f_{\mathbf{M}}(M)$ of $\mathbf{M}$ alone.

The same arguments for $\rho$ also gives the integral (23), with the small difference that $(1 - \Phi(\cdot))$ is replaced by $\Phi(\cdot)$. This difference is irrelevant, since to bound the Lipschitz constant, we will differentiate and bound the differential's absolute value.

Let $g(M, t, \alpha)$ be the integrant above, we want to differentiate $\int_{\mathbb{R}^k \times [0,1]^k} g(M, t, \alpha) \ d(M, t)$ by differentiating under the integral. Standard measure theory results (Theorem 6.28, Klenke [2020]) allows us to do this if we satisfy three conditions. These conditions are, in this case, equivalent to the following

  i) for all constant $\alpha$, the integral $\int_{\mathbb{R}^k \times [0,1]^k} g(M, t, \alpha) \ d(M, t)$ is finite.

  ii) for all constant $M, t$, the partial differential of $g(M, t, \alpha)$ with respect to $\alpha$ exists.

  iii) There exists a function $h(M,t)$, where $\int_{\mathbb{R}^k \times [0,1]^k} h(M,t) \ d(M,t)$ is finite and such that $|\frac{\partial}{\partial\alpha}g(M,t,\alpha)| \le h(M,t)$ for all $\alpha$.

The first two conditions are straightforward: The integral is equal to a probability, which is finite. And $g$ is a combination of differentiable functions making it differentiable itself. The last condition is more cumbersome, but the goal of this proof is to upperbound the integral by a constant, which clearly doesn't depend on $\alpha$. Hence the last condition will be satisfied during the proof.

Hence we can continue with our differentiation by differentiating under the integral.

$$\left|\frac{\partial}{\partial\alpha}\mathbb{P}[y\langle h_{\mathbf{A},\mathbf{t}}(w), \mathbf{A}x\rangle > \gamma_i/2]\right| =$$

$$\left|\int_{\mathbb{R}^k \times [0,1]^k} \frac{\partial}{\partial\alpha}f_{\mathbf{M}}(M)\left(1 - \Phi\left(\frac{\sqrt{k}\gamma_i/2 - \alpha\langle X', M\rangle}{\sqrt{1-\alpha^2}\|X'\|_2}\right)\right) d(M,t)\right| =$$

$$\frac{1}{2\pi}\left|\int_{\mathbb{R}^k \times [0,1]^k} f_{\mathbf{M}}(M)\exp\left(-\frac{1}{2}\left(\frac{\sqrt{k}\gamma_i/2 - \alpha\langle X', M\rangle}{\sqrt{1-\alpha^2}\|X'\|_2}\right)^2\right)\left(\frac{\frac{\alpha\sqrt{k}\gamma_i}{2} - \langle X', M\rangle}{(1-\alpha^2)^{3/2}\|X'\|_2}\right) d(M,t)\right| \le$$

$$\frac{1}{2\pi}\int_{\mathbb{R}^k \times [0,1]^k} f_{\mathbf{M}}(M)\exp\left(-\frac{(\sqrt{k}\gamma_i/2 - \alpha\langle X', M\rangle)^2}{2(1-\alpha^2)\|X'\|_2^2}\right)\left|\frac{\frac{\alpha\sqrt{k}\gamma_i}{2} - \langle X', M\rangle}{(1-\alpha^2)^{3/2}\|X'\|_2}\right| d(M,t).$$

For both $\alpha \in [-1/\sqrt{2}, 0]$ and $\alpha \in (\gamma_i, 1/\sqrt{2}]$, we have that $1 \geq (1 - \alpha^2) \geq 1/2$ and thus, for both $\phi$ and $\rho$, the above is upper bounded by

$$\frac{2^{3/2}}{2\pi} \cdot \int_{\mathbb{R}^k \times [0,1]^k} f_{\mathbf{M}}(M) \exp\left(-\frac{(\sqrt{k}\gamma_i/2 - \alpha\langle X', M\rangle)^2}{2\|X'\|_2^2}\right) \frac{\sqrt{k}\gamma_i + |\langle X', M\rangle|}{\|X'\|_2} \, d(M, t)$$

$$\leq \int_{\mathbb{R}^k \times [0,1]^k} f_{\mathbf{M}}(M) \exp\left(-\frac{(\sqrt{k}\gamma_i/2 - \alpha\langle X', M\rangle)^2}{2\|X'\|_2^2}\right) \left(\frac{\sqrt{k}\gamma_i}{\|X'\|_2} + \|M\|_2\right) \, d(M, t).$$

We now use that $|X'_i| \geq (1/2)(10\sqrt{k})^{-1}$ for all $i$. This implies $\|X'\|_2 \geq \sqrt{k(1/4)(10\sqrt{k})^{-2}} = 1/20$. We may thus further upper bound the above by

$$20 \cdot \int_{\mathbb{R}^k \times [0,1]^k} f_{\mathbf{M}}(M) \exp\left(-\frac{(\sqrt{k}\gamma_i/2 - \alpha\langle X', M\rangle)^2}{2\|X'\|_2^2}\right) \left(\sqrt{k}\gamma_i + \|M\|_2\right) \, d(M, t). \quad (24)$$

We will bound (24), by splitting it into 3 cases:

$$i) \ \|M\|_2^2 \leq \frac{9}{10}k \qquad\qquad ii) \ \frac{9}{10}k \leq \|M\|_2^2 \leq \frac{4}{3}k \qquad\qquad iii) \ \|M\|_2^2 \geq \frac{4}{3}k.$$

The arguments for cases *i)* and *iii)* do not depend on $\alpha$, and hence are identical for $\rho$ and $\phi$. In those cases, we simply exploit that $\|\mathbf{M}\|_2^2 \sim \chi_k^2$ and thus these cases are very unlikely. This implies that the integral over $f_M(M)$ is so small that we can afford to upper bound the exponential term in (24) by 1. For case *ii)*, we can use the assumptions on $\|M\|_2^2$ to show that the exponential term is no more than $c\exp(-\gamma_i^2 k/c)$ for a constant $c > 0$. We proceed to the three cases.

**case *i)*.** We simply upper bound the exponential term in (24) by 1 and use the assumption that $\|M\|_2^2 \leq \frac{9}{10}k$ to conclude

$$\exp\left(-\frac{(\sqrt{k}\gamma_i/2 - \alpha\langle X', M\rangle)^2}{2\|X'\|_2^2}\right) \left(\sqrt{k}\gamma_i + \|M\|_2\right) \leq 2\sqrt{k}.$$

Now since $\mathbf{M}$ is multivariate standard normal, $\|\mathbf{M}\|_2^2$ is $\chi_k^2$ distributed. Let $\mathbf{Z} \sim \chi_k^2$ with probability density function $f_Z(z)$. Then the integral in (24) in is bounded by:

$$40\sqrt{k} \int_{(\sqrt{9k/10})\mathbb{B}_2^k} f_{\mathbf{M}}(M) \, dM = 40\sqrt{k} \int_0^{9k/10} f_Z(z) \, dz = 40\sqrt{k} \cdot \mathbb{P}[\mathbf{Z} < 9k/10],$$

which by Theorem 17 is less than

$$80\sqrt{k}\exp\left(-k\gamma_i^2/800\right).$$

**case *ii)*.** We use the assumption that $\frac{9}{10}k \leq \|M\|_2^2 \leq \frac{4}{3}k$ together with the following observations

**Remark 13.** *If $\|X\|_2^2 \leq 4/3$, then $\|X'\|_2^2 < 2$.*

**Remark 14.** *If $\|X\|_2^2 \geq 9/10$, then $(8/9)\|X\|_2^2 \leq \langle X, X'\rangle \leq (10/9)\|X\|_2^2$.*

We prove Remark 13 and Remark 14 in Appendix E. Since $\sqrt{k}X = M$, Remark 14 gives $\langle X', M\rangle \geq (8/10)\sqrt{k}$ and hence

$$\alpha > \gamma_i \implies \frac{\sqrt{k}\gamma_i}{2} - \alpha\langle X', M\rangle \leq -\frac{3\sqrt{k}\gamma_i}{10} \leq 0 \implies -\left(\frac{\sqrt{k}\gamma_i}{2} - \alpha\langle X', M\rangle\right)^2 \leq -\frac{9k\gamma_i^2}{100}$$

$$\alpha < 0 \implies \frac{\sqrt{k}\gamma_i}{2} - \alpha\langle X', M\rangle \geq \frac{\sqrt{k}\gamma_i}{2} \geq 0 \implies -\left(\frac{\sqrt{k}\gamma_i}{2} - \alpha\langle X', M\rangle\right)^2 \leq -\frac{k\gamma_i^2}{4}.$$

Hence for both $\phi$ and $\rho$, the last two factors of the integral in (24) are bounded by:

$$\exp\left(-\frac{(\sqrt{k}\gamma_i/2 - \alpha\langle X', M\rangle)^2}{2\|X'\|_2^2}\right) \left(\sqrt{k}\gamma_i + \|M\|_2\right) \leq \frac{5}{2}\sqrt{k}\exp\left(-\frac{9k\gamma_i^2}{100 \cdot 4}\right).$$

Which gives the following:

$$50\sqrt{k}\exp\left(-\frac{k\gamma_i^2}{50}\right) \int_{(\sqrt{9k/10}\cdot\mathbb{B}_2^k)^C \cap (\sqrt{4k/3}\cdot\mathbb{B}_2^k)} f_{\mathbf{M}}(M) \, dM \leq 50\sqrt{k}\exp\left(-\frac{k\gamma_i^2}{50}\right).$$

**case *iii*).** We bound the last two factors of the integral (24) under the assumption that $\|M\|_2^2 \geq \frac{4}{3}k$. Here we simply upper bound the exponential by 1 and get

$$\exp\left(-\frac{(\sqrt{k}\gamma_i/2 - \alpha\langle X', M\rangle)^2}{2\|X'\|_2^2}\right)\left(\sqrt{k}\gamma_i + \|M\|_2\right) \leq 2\|M\|_2,$$

hence the integral (24) is bounded by:

$$40\int_{(\sqrt{4k/3}\cdot\mathbb{B}_2^k)^C} f_{\mathbf{M}}(M)\|M\|_2 \ dM. \tag{25}$$

Recall that $\|\mathbf{M}\|_2^2 \sim \chi_k^2$ and let $\mathbf{Z} \sim \chi_k^2$ with probability density function $f_Z(z)$. Then the integral (25) is equal to

$$40\int_{4k/3}^{\infty} f_Z(z)\sqrt{z} \ dz.$$

Let also $L_i = \left[\frac{4}{3}k\cdot 2^i, \frac{4}{3}k\cdot 2^{i+1}\right)$ for $i \in \mathbb{Z}_{\geq 0}$. By definition, the $L_i$'s partition $\left[\frac{4}{3}k, \infty\right)$, and we upper bound with:

$$40\sum_{i=0}^{\infty}\int_{L_i} f_Z(z)\sqrt{\frac{4}{3}k\cdot 2^{i+1}} \ dz \leq 70\sqrt{k}\sum_{i=0}^{\infty}\mathbb{P}[\mathbf{Z} \in L_i]2^{i/2} \leq 70\sqrt{k}\sum_{i=0}^{\infty}\mathbb{P}\left[\mathbf{Z} \geq \frac{4}{3}k\cdot 2^i\right]2^{i/2}.$$

Using the following remark:

**Remark 15** (Laurent and Massart [2000], equation 4.3, page 1326)**.**
*Let* $\mathbf{Z} \sim \chi_k^2$, $y > 0$ *then:*

$$\mathbb{P}[\mathbf{Z} \geq 2\sqrt{ky} + 2y + k] \leq \exp(-y).$$

With $y = c\frac{4}{3}k$, where $c = \frac{1}{8^2}2^i$. We have:

$$2\sqrt{ky} + 2y + k = \left(\sqrt{c}\frac{4}{\sqrt{3}} + c\frac{8}{3} + 1\right)k \leq \left(\frac{4\sqrt{3}}{24} + \frac{1}{24} + 1\right)k\cdot 2^i \leq \frac{4}{3}k\cdot 2^i.$$

Hence we can finish the bound for this case, using the assumption that $k \geq \gamma_i^{-2}72\ln(2)$

$$70\sqrt{k}\sum_{i=0}^{\infty}\mathbb{P}\left[\mathbf{Z} \geq \frac{4}{3}k\cdot 2^i\right]2^{i/2} \leq 70\sqrt{k}\sum_{i=0}^{\infty}\exp\left(-\frac{1}{48}k\cdot 2^i\right)2^{i/2}$$

$$\leq 70\sqrt{k}\exp\left(-\frac{1}{48}k\right)\sum_{i=0}^{\infty}\exp\left(-\frac{1}{48}k\right)^i 2^{i/2}$$

$$\leq 140\sqrt{k}\exp\left(-\frac{1}{48}k\gamma_i^2\right).$$

**Collecting the three cases.** Hence in total $|\frac{\partial}{\partial\alpha}\phi|$ for $\alpha \in [-1/\sqrt{2}, 0]$ and $|\frac{\partial}{\partial\alpha}\rho|$ for $\alpha \in \left(\gamma_i, 1\sqrt{2}\right]$ are bounded by:

$$80\sqrt{k}\exp\left(-k\gamma_i^2/800\right) + 50\sqrt{k}\exp\left(-\frac{k\gamma_i^2}{50}\right) + 140\sqrt{k}\exp\left(-\frac{1}{48}k\gamma_i^2\right)$$

$$\leq 140\sqrt{k}\exp\left(-k\gamma_i^2/800\right).$$

Using that $\gamma_{i+1} = 2\gamma_i$, this completes the proof of Lemma 12.

## C   Meet in the Middle Bound

The goal of this section is to prove the following

**Restatement of Lemma 7.** *There is a constant $c > 0$ such that with probability at least $1 - \delta$ over $\mathbf{S} \sim \mathcal{D}^n$ we have*

$$\sup_{w \in \mathcal{H}(\Gamma_i, L_j)} \left| \mathbb{E}_{\mathbf{A}, \mathbf{t}} [\mathcal{L}_{\mathbf{A}\mathcal{D}}^{\gamma_i/2}(h_{\mathbf{A}, \mathbf{t}}(w)) - \mathcal{L}_{\mathbf{A}\mathbf{S}}^{\gamma_i/2}(h_{\mathbf{A}, \mathbf{t}}(w))] \right| \leq$$

$$c \left( \sqrt{\frac{(\ell_{j+1} + \exp(-\gamma_{i+1}^2 k/c))(k + \ln(e/\delta))}{n}} + \frac{(k + \ln(e/\delta))}{n} \right).$$

Notice here that the two losses $\mathcal{L}_{\mathbf{A}\mathcal{D}}^{\gamma_i/2}(h_{\mathbf{A}, \mathbf{t}}(w))$ and $\mathcal{L}_{\mathbf{A}\mathbf{S}}^{\gamma_i/2}(h_{\mathbf{A}, \mathbf{t}}(w))$ refer to the same margin $\gamma_i/2$ and $h_{\mathbf{A}, \mathbf{t}}(w)$ has been discretized to have all coordinates of the form $(1/2)(10\sqrt{k})^{-1} + z(10\sqrt{k})^{-1}$ for integer $z$. Intuitively, we will try to exploit this discretization to union bound over a grid of finitely many hypotheses. Unfortunately, the random matrix $\mathbf{A}$ may increase the norm of $w$ arbitrarily much, and thus a single grid is insufficient. Instead, we need an infinite sequence of grids. For this, let $\mathcal{G}_0$ denote the set of all vectors in $4\mathbb{B}_2^k$ whose coordinates are of the form $(1/2)(10\sqrt{k})^{-1} + z(10\sqrt{k})^{-1}$ for integer $z$. More generally, let $\mathcal{G}_i$ for $i > 0$ denote the set of all vectors in $(2^i \cdot 4\mathbb{B}_2^k)$ whose coordinates are of this form. Since $\|x\|_1 \leq \sqrt{k}\|x\|_2$ for any $x \in \mathbb{R}^k$, we have that $\mathcal{G}_i \subset (2^i \cdot 4\mathbb{B}_2^k) \subseteq \sqrt{k}(2^i \cdot 4\mathbb{B}_1^k)$. For a vector $x \in \mathcal{G}_i$, let $i(x) = (i_1, \ldots, i_k)$ denote the integers so that $x = (10\sqrt{k})^{-1}i(x) + (1/2)(10\sqrt{k})^{-1}\mathbf{1}$ with $\mathbf{1} \in \mathbb{R}^k$ the all-1's vector. Then by the triangle inequality, we have $(10\sqrt{k})^{-1}\|i(x)\|_2 \leq \|x\|_2 + (1/2)(10\sqrt{k})^{-1}\|\mathbf{1}\|_2 \leq 2^i \cdot 4 + 1/20$. This implies $\|i(x)\|_1 \leq (10\sqrt{k})\sqrt{k}(2^i \cdot 4 + 1/20) \leq (5 \cdot 2^{i+3} + 1)k$. Since each coordinate of $i(x)$ is an integer, there are thus at most $2^k$ choices for the signs and $\sum_{t=0}^{(5 \cdot 2^{i+3}+1)k} \binom{k+t-1}{t}$ choices for the absolute values of the integers. That is, we have

$$|\mathcal{G}_i| \leq 2^k \cdot \sum_{t=0}^{(5 \cdot 2^{i+3}+1)k} \binom{k+t-1}{t} \leq 2^{(5 \cdot 2^{i+3}+3)k} \leq 2^{2^{i+7}k}. \tag{26}$$

We now start by considering a fixed outcome $A$ of the random matrix $\mathbf{A}$. For such a fixed $A$, the training set $\mathbf{S}$ behaves well in the sense that $\mathcal{L}_{A\mathcal{D}}^{\gamma}(w)$ and $\mathcal{L}_{A\mathbf{S}}^{\gamma}(w)$ are close with high probability for any $w$. This is formalized in the following remark

**Remark 16.** *For any distribution $\mathcal{D}$ over $\mathcal{X} \times \{-1, 1\}$, fixed $w \in \mathcal{H}$, margin $\gamma$ and any $A \in \mathbb{R}^{k \times d}$, it holds with probability at least $1 - \delta$ over $\mathbf{S} \sim \mathcal{D}^n$ that*

$$|\mathcal{L}_{A\mathcal{D}}^{\gamma}(w) - \mathcal{L}_{A\mathbf{S}}^{\gamma}(w)| \leq \sqrt{\frac{8\mathcal{L}_{A\mathcal{D}}^{\gamma}(w) \ln(1/\delta)}{n}} + \frac{2\ln(1/\delta)}{n}.$$

The proof of Remark 16 is a simple application of Bernstein's and can be found in Appendix E.

In Lemma 7, the matrix $\mathbf{A}$ is not fixed but random. Thus we need to find a formal property of the training set $\mathbf{S}$ under which $\mathcal{L}_{\mathbf{A}\mathcal{D}}^{\gamma_i/2}(h_{\mathbf{A}, \mathbf{t}}(w))$ and $\mathcal{L}_{\mathbf{A}\mathbf{S}}^{\gamma_i/2}(h_{\mathbf{A}, \mathbf{t}}(w))$ are close in expectation over the random choice of $\mathbf{A}$. With this goal in mind, we now say that a matrix $A$ in the support of $\mathbf{A}$ and a training set $S$ has *distortion* at least $\beta$, if there is a grid $\mathcal{G}_a$ and a vector $w \in \mathcal{G}_a$ such that

$$|\mathcal{L}_{A\mathcal{D}}^{\gamma_i/2}(w) - \mathcal{L}_{A\mathbf{S}}^{\gamma_i/2}(w)| > \beta \cdot \left( \sqrt{\frac{8\mathcal{L}_{A\mathcal{D}}^{\gamma_i/2}(w)(2^{a+7}k + \ln(1/\delta))}{n}} + \frac{2(2^{a+7}k + \ln(1/\delta))}{n} \right).$$

For a training set $S$, we use $D_\beta(S)$ to denote the set of matrices $A$ with distortion at least $\beta$ for $S$.

We observe that for a fixed matrix $A$, grid $\mathcal{G}_a$ and $\beta > 1$, we have by Remark 16 with $\delta_a' = (\delta/2^{2^{a+7}k})^\beta$ and a union bound over all $w \in \mathcal{G}_a$, that with probability at least $1 - |\mathcal{G}_a|\delta_a'$, it holds for

all $w \in \mathcal{G}_a$ that

$$|\mathcal{L}_{AD}^{\gamma_i/2}(w) - \mathcal{L}_{A\mathbf{S}}^{\gamma_i/2}(w)| \leq \sqrt{\frac{8\mathcal{L}_{AD}^{\gamma_i/2}(w)\ln(1/\delta_a')}{n}} + \frac{2\ln(1/\delta_a')}{n}$$

$$= \sqrt{\frac{8\mathcal{L}_{AD}^{\gamma_i/2}(w)(\beta 2^{a+7}k + \beta\ln(1/\delta))}{n}} + \frac{2(\beta 2^{a+7}k + \beta\ln(1/\delta))}{n}$$

$$\leq \beta \cdot \left( \sqrt{\frac{8\mathcal{L}_{AD}^{\gamma_i/2}(w)(2^{a+7}k + \ln(1/\delta))}{n}} + \frac{2(2^{a+7}k + \ln(1/\delta))}{n} \right).$$

Thus for $\beta \geq 2$, we have

$$\mathbb{P}_{\mathbf{S}}[A \in D_\beta(\mathbf{S})] \leq \sum_{a=0}^{\infty} |\mathcal{G}_a|\delta_a'$$

$$\leq \sum_{a=0}^{\infty} \delta^\beta \cdot 2^{-(\beta-1)2^{a+7}k}$$

$$\leq 2 \cdot \delta^\beta \cdot 2^{-(\beta-1)2^7 k}.$$

By Markov's inequality, we have

$$\mathbb{P}_{\mathbf{S}}[\mathbb{P}_{\mathbf{A}}[\mathbf{A} \in D_\beta(\mathbf{S})] > 2 \cdot \delta^{\beta/2} \cdot 2^{-(\beta-1)\cdot 2^6 k}] \leq \frac{\mathbb{E}_{\mathbf{S}}[\mathbb{P}_{\mathbf{A}}[\mathbf{A} \in D_\beta(\mathbf{S})]}{2 \cdot \delta^{\beta/2} \cdot 2^{-(\beta-1)\cdot 2^6 k}}$$

$$= \frac{\mathbb{E}_{\mathbf{A}}[\mathbb{P}_{\mathbf{S}}[\mathbf{A} \in D_\beta(\mathbf{S})]}{2 \cdot \delta^{\beta/2} \cdot 2^{-(\beta-1)\cdot 2^6 k}}$$

$$\leq \delta^{\beta/2} \cdot 2^{-(\beta-1)2^6 k}.$$

Now call a training set $S$ *representative* if it holds for every $\beta = 2^h$ with integer $h \geq 1$ that

$$\mathbb{P}_{\mathbf{A}}[\mathbf{A} \in D_\beta(\mathbf{S})] \leq 2 \cdot \delta^{\beta/2} \cdot 2^{-(\beta-1)\cdot 2^6 k}.$$

A union bound implies that $\mathbf{S}$ is representative with probability at least

$$1 - \sum_{h=1}^{\infty} 2 \cdot \delta^{2^{h-1}} \cdot 2^{-(2^h-1)2^6 k} \geq 1 - \frac{\delta}{2^{2^6 k - 2}} \geq 1 - \delta.$$

Now define for integer $h \geq 1$ the set

$$K_h(S) = D_{2^h}(S) \setminus \left( \cup_{b=h+1}^{\infty} D_{2^b}(S) \right).$$

Let $K_0(S)$ be defined as

$$K_0(S) = \mathrm{supp}(\mathbf{A}) \setminus \left( \cup_{b=1}^{\infty} D_{2^b}(S) \right).$$

For any $w \in \mathcal{H}$, we may use the triangle inequality to conclude

$$\left| \mathbb{E}_{\mathbf{A},\mathbf{t}}[\mathcal{L}_{\mathbf{A}D}^{\gamma_i/2}(h_{\mathbf{A},\mathbf{t}}(w)) - \mathcal{L}_{\mathbf{A}S}^{\gamma_i/2}(h_{\mathbf{A},\mathbf{t}}(w))] \right| \leq$$

$$\sum_{h=0}^{\infty} \mathbb{E}_{\mathbf{A},\mathbf{t}} \left[ \left| \mathcal{L}_{\mathbf{A}D}^{\gamma_i/2}(h_{\mathbf{A},\mathbf{t}}(w)) - \mathcal{L}_{\mathbf{A}S}^{\gamma_i/2}(h_{\mathbf{A},\mathbf{t}}(w)) \right| \mid \mathbf{A} \in K_h(S) \right] \mathbb{P}_{\mathbf{A}}[\mathbf{A} \in K_h(S)].$$

Now consider an $A \in K_h(S)$. Then $A$ has distortion no more than $2^{h+1}$ by definition of $K_h(S)$. This implies that if $h_{A,t}(w)$ is in $\mathcal{G}_a$ but not $\mathcal{G}_b$ for $b < a$, then $\|h_{A,t}(w)\|_2 \geq 2^{a+1}$ by definition of $\mathcal{G}_b$ and we get

$$|\mathcal{L}_{AD}^{\gamma_i/2}(h_{A,t}(w)) - \mathcal{L}_{A\mathbf{S}}^{\gamma_i/2}(h_{A,t}(w))| \leq$$

$$2^{h+1} \cdot \left( \sqrt{\frac{8\mathcal{L}_{AD}^{\gamma_i/2}(w)(2^{a+7}k + \ln(1/\delta))}{n}} + \frac{2(2^{a+7}k + \ln(1/\delta))}{n} \right) \leq$$

$$2^{h+8}\|h_{A,t}(w)\|_2 \cdot \left( \sqrt{\frac{8\mathcal{L}_{AD}^{\gamma_i/2}(w)(k + \ln(1/\delta))}{n}} + \frac{2(k + \ln(1/\delta))}{n} \right).$$

Using Cauchy-Schwartz, we thus get for any $w \in \mathcal{H}$ that

$$\left| \mathbb{E}_{\mathbf{A},\mathbf{t}}[\mathcal{L}_{\mathbf{A}\mathcal{D}}^{\gamma_i/2}(h_{\mathbf{A},\mathbf{t}}(w)) - \mathcal{L}_{\mathbf{A}S}^{\gamma_i/2}(h_{\mathbf{A},\mathbf{t}}(w))] \right| \leq$$

$$\sum_{h=0}^{\infty} 2^{h+8} \mathbb{E}_{\mathbf{A},\mathbf{t}} \left[ \|h_{\mathbf{A},\mathbf{t}}(w)\|_2 \cdot \left( \sqrt{\frac{8\mathcal{L}_{\mathbf{A}\mathcal{D}}^{\gamma_i/2}(w)(k+\ln(1/\delta))}{n}} + \right. \right.$$

$$\left. \left. \frac{2(k+\ln(1/\delta))}{n} \right) \mid \mathbf{A} \in K_h(S) \right] \mathbb{P}_{\mathbf{A}}[\mathbf{A} \in K_h(S)] \leq$$

$$\sum_{h=0}^{\infty} 2^{h+8} \sqrt{\mathbb{E}_{\mathbf{A},\mathbf{t}}\left[\|h_{\mathbf{A},\mathbf{t}}(w)\|_2^2 \mid \mathbf{A} \in K_h(S)\right]} \cdot$$

$$\sqrt{\mathbb{E}_{\mathbf{A},\mathbf{t}}\left[ \left( \sqrt{\frac{8\mathcal{L}_{\mathbf{A}\mathcal{D}}^{\gamma_i/2}(w)(k+\ln(1/\delta))}{n}} + \frac{2(k+\ln(1/\delta))}{n} \right)^2 \mid \mathbf{A} \in K_h(S) \right]} \mathbb{P}_{\mathbf{A}}[\mathbf{A} \in K_h(S)].$$

By Cauchy-Schwartz, this is at most

$$\sqrt{\sum_{h=0}^{\infty} 2^{2h+16} \mathbb{E}_{\mathbf{A},\mathbf{t}}[\|h_{\mathbf{A},\mathbf{t}}(w)\|_2^2 \mid \mathbf{A} \in K_h(S)] \mathbb{P}_{\mathbf{A}}[\mathbf{A} \in K_h(S)]} \cdot$$

$$\sqrt{\sum_{h=0}^{\infty} \mathbb{E}_{\mathbf{A},\mathbf{t}}\left[ \left( \sqrt{\frac{8\mathcal{L}_{\mathbf{A}\mathcal{D}}^{\gamma_i/2}(w)(k+\ln(1/\delta))}{n}} + \frac{2(k+\ln(1/\delta))}{n} \right)^2 \mid \mathbf{A} \in K_h(S) \right] \mathbb{P}_{\mathbf{A}}[\mathbf{A} \in K_h(S)]}.$$

Using Cauchy-Schwartz again and Jensen's inequality, the first sum is bounded by

$$\sum_{h=0}^{\infty} 2^{2h+16} \mathbb{E}_{\mathbf{A},\mathbf{t}}[\|h_{\mathbf{A},\mathbf{t}}(w)\|_2^2 \mid \mathbf{A} \in K_h(S)] \mathbb{P}_{\mathbf{A}}[\mathbf{A} \in K_h(S)] \leq$$

$$\sqrt{\sum_{h=0}^{\infty} 2^{4h+64} \mathbb{P}_{\mathbf{A}}[\mathbf{A} \in K_h(S)]} \cdot \sqrt{\sum_{h=0}^{\infty} \mathbb{E}_{\mathbf{A},\mathbf{t}}[\|h_{\mathbf{A},\mathbf{t}}(w)\|_2^2 \mid \mathbf{A} \in K_h(S)]^2 \mathbb{P}_{\mathbf{A}}[\mathbf{A} \in K_h(S)]} \leq$$

$$\sqrt{\sum_{h=0}^{\infty} 2^{4h+64} \mathbb{P}_{\mathbf{A}}[\mathbf{A} \in D_{2^h}(S)]} \cdot \sqrt{\sum_{h=0}^{\infty} \mathbb{E}_{\mathbf{A},\mathbf{t}}[\|h_{\mathbf{A},\mathbf{t}}(w)\|_2^4 \mid \mathbf{A} \in K_h(S)] \mathbb{P}_{\mathbf{A}}[\mathbf{A} \in K_h(S)]} \leq$$

$$\sqrt{\sum_{h=0}^{\infty} 2^{4h+64} 2(\delta/2^{2^7 k+1})^{(2^h-1)/2}} \cdot \sqrt{\mathbb{E}_{\mathbf{A},\mathbf{t}}[\|h_{\mathbf{A},\mathbf{t}}(w)\|_2^4]} \leq$$

$$2^{33} \cdot \sqrt{\mathbb{E}_{\mathbf{A},\mathbf{t}}[\|h_{\mathbf{A},\mathbf{t}}(w)\|_2^4]}.$$

Using Jensen's inequality on the second sum, we find that

$$\sum_{h=0}^{\infty} \mathbb{E}_{\mathbf{A},\mathbf{t}}\left[ \left( \sqrt{\frac{8\mathcal{L}_{\mathbf{A}\mathcal{D}}^{\gamma_i/2}(w)(k+\ln(1/\delta))}{n}} + \frac{2(k+\ln(1/\delta))}{n} \right)^2 \mid \mathbf{A} \in K_h(S) \right] \mathbb{P}_{\mathbf{A}}[\mathbf{A} \in K_h(S)] =$$

$$\mathbb{E}_{\mathbf{A},\mathbf{t}}\left[ \left( \sqrt{\frac{8\mathcal{L}_{\mathbf{A}\mathcal{D}}^{\gamma_i/2}(w)(k+\ln(1/\delta))}{n}} + \frac{2(k+\ln(1/\delta))}{n} \right)^2 \right].$$

For positive constants $c_0, c_1, c_2$, we have that the function $f(t) = (\sqrt{c_0 t + c_1} + c_2)^2$ is concave for $t \geq 0$. To see this, we compute its derivative

$$f'(t) = 2(\sqrt{c_0 t + c_1} + c_2) \cdot \frac{c_0}{2\sqrt{c_0 t + c_1}} = c_0 + \frac{c_0 c_2}{\sqrt{c_0 t + c_1}},$$

and its second derivative

$$f''(t) = \frac{-c_0^2 c_2}{2(c_0 t + c_1)^{3/2}}.$$

This is a negative function for $t \geq 0$. We thus use Jensen's inequality to conclude

$$\mathbb{E}_{\mathbf{A},\mathbf{t}}\left[\left(\sqrt{\frac{8\mathcal{L}_{\mathbf{A}\mathcal{D}}^{\gamma_i/2}(w)(k+\ln(1/\delta))}{n}} + \frac{2(k+\ln(1/\delta))}{n}\right)^2\right] \leq$$

$$\left(\sqrt{\frac{8\mathbb{E}_{\mathbf{A},\mathbf{t}}\left[\mathcal{L}_{\mathbf{A}\mathcal{D}}^{\gamma_i/2}(w)\right](k+\ln(1/\delta))}{n}} + \frac{2(k+\ln(1/\delta))}{n}\right)^2.$$

Combining it all, we have thus shown

$$\left|\mathbb{E}_{\mathbf{A},\mathbf{t}}[\mathcal{L}_{\mathbf{A}\mathcal{D}}^{\gamma_i/2}(h_{\mathbf{A},\mathbf{t}}(w)) - \mathcal{L}_{\mathbf{A}S}^{\gamma_i/2}(h_{\mathbf{A},\mathbf{t}}(w))]\right| \leq$$

$$\sqrt{2^{33} \cdot \sqrt{\mathbb{E}_{\mathbf{A},\mathbf{t}}[\|h_{\mathbf{A},\mathbf{t}}(w)\|_2^4]}} \cdot \sqrt{\left(\sqrt{\frac{8\mathbb{E}_{\mathbf{A},\mathbf{t}}\left[\mathcal{L}_{\mathbf{A}\mathcal{D}}^{\gamma_i/2}(w)\right](k+\ln(1/\delta))}{n}} + \frac{2(k+\ln(1/\delta))}{n}\right)^2} \leq$$

$$2^{17} \cdot \mathbb{E}_{\mathbf{A},\mathbf{t}}[\|h_{\mathbf{A},\mathbf{t}}(w)\|_2^4]^{1/4} \cdot \left(\sqrt{\frac{8\mathbb{E}_{\mathbf{A},\mathbf{t}}\left[\mathcal{L}_{\mathbf{A}\mathcal{D}}^{\gamma_i/2}(w)\right](k+\ln(1/\delta))}{n}} + \frac{2(k+\ln(1/\delta))}{n}\right).$$

We now bound $\mathbb{E}_{\mathbf{A},\mathbf{t}}[\|h_{\mathbf{A},\mathbf{t}}(w)\|_2^4]$ as follows

$$\mathbb{E}_{\mathbf{A},\mathbf{t}}[\|h_{\mathbf{A},\mathbf{t}}(w)\|_2^4] =$$

$$\mathbb{E}_{\mathbf{A},\mathbf{t}}[\|\mathbf{A}w + (h_{\mathbf{A},\mathbf{t}}(w) - \mathbf{A}w)\|_2^4] \leq$$

$$\mathbb{E}_{\mathbf{A},\mathbf{t}}[(\|\mathbf{A}w\|_2 + \|h_{\mathbf{A},\mathbf{t}}(w) - \mathbf{A}w\|_2)^4] \leq$$

$$\mathbb{E}_{\mathbf{A},\mathbf{t}}\left[\left(\|\mathbf{A}w\|_2 + \sqrt{k(10\sqrt{k})^{-2}}\right)^4\right] =$$

$$\mathbb{E}_{\mathbf{A},\mathbf{t}}\left[(\|\mathbf{A}w\|_2 + 1/10)^4\right] =$$

$$\sum_{b=0}^{4}\binom{4}{b}\mathbb{E}_{\mathbf{A},\mathbf{t}}[\|\mathbf{A}w\|_2^b]10^{-(4-b)}.$$

Recalling that $\|\mathbf{A}w\|_2^2 \sim (1/k)\chi_k^2$, we have from the moments of the chi-square distribution that for even $k \geq 4$:

$$\mathbb{E}_{\mathbf{A},\mathbf{t}}[\|\mathbf{A}w\|_2^b] \leq \mathbb{E}_{\mathbf{A},\mathbf{t}}[\|\mathbf{A}w\|_2^4] = k^{-2}\mathbb{E}_{\mathbf{A},\mathbf{t}}[(k\|\mathbf{A}w\|_2^2)^2] = k^{-2}2^2\frac{(2+k/2)!}{(k/2)!} \leq 4.$$

Hence

$$\mathbb{E}_{\mathbf{A},\mathbf{t}}[\|h_{\mathbf{A},\mathbf{t}}(w)\|_2^4] \leq \sum_{b=0}^{4}\binom{4}{b}4 \cdot 10^{-(4-b)} \leq (4+1/10)^4 < 5^4.$$

We thus have

$$\left|\mathbb{E}_{\mathbf{A},\mathbf{t}}[\mathcal{L}_{\mathbf{A}\mathcal{D}}^{\gamma_i/2}(h_{\mathbf{A},\mathbf{t}}(w)) - \mathcal{L}_{\mathbf{A}S}^{\gamma_i/2}(h_{\mathbf{A},\mathbf{t}}(w))]\right| \leq$$

$$2^{20} \cdot \left(\sqrt{\frac{8\mathbb{E}_{\mathbf{A},\mathbf{t}}\left[\mathcal{L}_{\mathbf{A}\mathcal{D}}^{\gamma_i/2}(w)\right](k+\ln(1/\delta))}{n}} + \frac{2(k+\ln(1/\delta))}{n}\right).$$

Finally, we exploit that for any $w \in \mathcal{H}(\Gamma_i, L_j)$, we have by definition that $\mathcal{L}_{\mathcal{D}}^{(3/4)\gamma_i}(w) \leq \ell_{j+1}$. Thus for any such $w$, we have

$$
\begin{aligned}
\mathbb{E}_{\mathbf{A},\mathbf{t}}[\mathcal{L}_{\mathbf{A}\mathcal{D}}^{\gamma_i/2}(w)] &= \mathbb{E}_{\mathbf{A},\mathbf{t}}[\mathbb{P}_{(\mathbf{x},\mathbf{y})\sim\mathcal{D}}[\mathbf{y}\langle h_{\mathbf{A},\mathbf{t}}(w), \mathbf{A}\mathbf{x}\rangle \leq \gamma_i/2]] \\
&= \mathbb{E}_{(\mathbf{x},\mathbf{y})\sim\mathcal{D}}[\mathbb{P}_{\mathbf{A},\mathbf{t}}[\mathbf{y}\langle h_{\mathbf{A},\mathbf{t}}(w), \mathbf{A}\mathbf{x}\rangle \leq \gamma_i/2]] \\
&\leq \mathbb{P}_{(\mathbf{x},\mathbf{y})\sim\mathcal{D}}[\mathbf{y}\langle w, \mathbf{x}\rangle \leq (3/4)\gamma_i] \\
&\quad + \mathbb{E}_{(\mathbf{x},\mathbf{y})\sim\mathcal{D}}[\mathbb{P}_{\mathbf{A},\mathbf{t}}[\mathbf{y}\langle h_{\mathbf{A},\mathbf{t}}(w), \mathbf{A}\mathbf{x}\rangle \leq \gamma_i/2] \mid \mathbf{y}\langle w, \mathbf{x}\rangle > (3/4)\gamma_i] \\
&\leq \mathcal{L}_{\mathcal{D}}^{(3/4)\gamma_i}(w) + \sup_{\mu > (3/4)\gamma_i}[\mathbb{P}_{\mathbf{A},\mathbf{t}}[\langle h_{\mathbf{A},\mathbf{t}}(w), \mathbf{A}x\rangle \leq \gamma_i/2 \mid y\langle w, x\rangle = \mu].
\end{aligned}
$$

Using Lemma 5 and that $\mathcal{L}_{\mathcal{D}}^{(3/4)\gamma_i}(w) \in L_j$ by definition of $\mathcal{H}(\Gamma_i, L_j)$, there is a constant $c > 0$ such that this is bounded by

$$
\begin{aligned}
&\leq \mathcal{L}_{\mathcal{D}}^{(3/4)\gamma_i}(w) + c\exp(-k(\gamma_i/4)^2/c) \\
&\leq \ell_{j+1} + c\exp(-k\gamma_{i+1}^2/(16c)).
\end{aligned}
$$

We have thus reached the conclusion that there is a constant $c > 0$, such that with probability at least $1 - \delta$ over $\mathbf{S} \sim \mathcal{D}^n$, it holds that

$$
\sup_{w \in \mathcal{H}(\Gamma_i, L_j)} \left| \mathbb{E}_{\mathbf{A},\mathbf{t}}[\mathcal{L}_{\mathbf{A}\mathcal{D}}^{\gamma_i/2}(h_{\mathbf{A},\mathbf{t}}(w)) - \mathcal{L}_{\mathbf{A}S}^{\gamma_i/2}(h_{\mathbf{A},\mathbf{t}}(w))] \right| \leq
$$

$$
c \cdot \left( \sqrt{\frac{(\ell_{j+1} + \exp(-k\gamma_{i+1}^2/c))(k + \ln(1/\delta))}{n}} + \frac{k + \ln(1/\delta)}{n} \right).
$$

This completes the proof of Lemma 7.

## D Within Constant Factors

In this section we prove

**Restatement of Lemma 4.** *There is a constant $c > 1$, such that for any $0 < \delta < 1$ and any $\Gamma_i = (\gamma_i, \gamma_{i+1}]$, it holds with probability at least $1 - \delta$ over a random sample $\mathbf{S} \sim \mathcal{D}^n$ that*

$$
\forall w \in \mathcal{H} : \mathcal{L}_{\mathbf{S}}^{\gamma_i}(w) \geq \frac{\mathcal{L}_{\mathcal{D}}^{(3/4)\gamma_i}(w)}{4} - c\left(\frac{\ln(e\gamma_{i+1}^2 n)}{\gamma_{i+1}^2 n} + \frac{\ln(e/\delta)}{n}\right).
$$

The proof follows mostly the ideas in Grønlund et al. [2020a] that were outlined in the proof overview in Section 2.

*Proof.* Let $k \geq 1$ be a parameter to be determined and consider the random construction of $\mathbf{A}$ and $\mathbf{t}$ as defined in Section 3.2. Let $\mathcal{G}_a$ be defined as in Section C, i.e. $\mathcal{G}_a$ contains all vectors in $2^a \cdot 4\mathbb{B}_2^k$. We say that a matrix $A$ in the support of $\mathbf{A}$ and a training set $S$ is $\alpha$-*unusual*, if there is a vector $w \in \mathcal{G}_0$ such that

$$
\mathcal{L}_{AS}^{(7/8)\gamma_i}(w) < \frac{\mathcal{L}_{A\mathcal{D}}^{(7/8)\gamma_i}(w)}{2} - \frac{2^{11}k + \ln(1/\alpha)}{n}.
$$

For a fixed matrix $A$ and vector $w \in \mathcal{W}_0$, we have by Bernstein's inequality and $\mathbb{E}_{\mathbf{S}}[\mathcal{L}_{AS}^{(7/8)\gamma_i}(w)] = \mathcal{L}_{A\mathcal{D}}^{(7/8)\gamma_i}(w)$ that

$$
\mathbb{P}_{\mathbf{S}}\left[\left|\mathcal{L}_{AS}^{(7/8)\gamma_i}(w) - \mathcal{L}_{A\mathcal{D}}^{(7/8)\gamma_i}(w)\right| > t/n\right] < \exp\left(-\frac{\frac{1}{2}t^2}{n\mathcal{L}_{A\mathcal{D}}^{(7/8)\gamma_i}(w) + \frac{1}{3}t}\right).
$$

Setting

$$
t = n \cdot \left(\frac{\mathcal{L}_{A\mathcal{D}}^{(7/8)\gamma_i}(w)}{2} + Z\right),
$$

with $Z = 16\ln(1/\alpha)/n$ gives

$$\mathbb{P}_{\mathbf{S}}\left[\left|\mathcal{L}_{AS}^{(7/8)\gamma_i}(w) - \mathcal{L}_{A\mathcal{D}}^{(7/8)\gamma_i}(w)\right| > \left(\frac{\mathcal{L}_{A\mathcal{D}}^{(7/8)\gamma_i}(w)}{2} + Z\right)\right]$$

$$< \exp\left(-\frac{\frac{n^2}{2}\left(\frac{\mathcal{L}_{A\mathcal{D}}^{(7/8)\gamma_i}(w)}{2} + Z\right)^2}{n\mathcal{L}_{A\mathcal{D}}^{(7/8)\gamma_i}(w) + \frac{n}{3}\left(\frac{\mathcal{L}_{A\mathcal{D}}^{(7/8)\gamma_i}(w)}{2} + Z\right)}\right)$$

$$\leq \exp\left(-\frac{\frac{n^2}{8}\max\{\mathcal{L}_{A\mathcal{D}}^{(7/8)\gamma_i}(w), Z\}^2}{2n\max\{\mathcal{L}_{A\mathcal{D}}^{(7/8)\gamma_i}(w), Z\}}\right)$$

$$\leq \exp\left(-\frac{nZ}{16}\right)$$

$$= \alpha.$$

A union bound over all $w \in \mathcal{G}_0$ with $\alpha' = \alpha/e^{2^7 k}$ gives that a fixed matrix $A$ is $\alpha$-unusual for $\mathbf{S} \sim \mathcal{D}^n$ with probability at most

$$|\mathcal{G}_0|\frac{\alpha}{e^{2^7 k}} < \alpha.$$

Now call a training set $S$ $\alpha$-representative if $\mathbf{A}$ is $\alpha$-unusual for $S$ with probability less than $1/4$. By Markov's inequality, we have

$$\mathbb{P}_{\mathbf{S}}[\mathbb{P}_{\mathbf{A}}[(\mathbf{S}, \mathbf{A}) \text{ is } \alpha\text{-unusual}] \geq 1/4] \leq \frac{\mathbb{E}_{\mathbf{S}}[\mathbb{P}_{\mathbf{A}}[(\mathbf{S}, \mathbf{A}) \text{ is } \alpha\text{-unusual}]]}{1/4}$$

$$= 4 \cdot \mathbb{E}_{\mathbf{A}}[\mathbb{P}_{\mathbf{S}}[(\mathbf{S}, \mathbf{A}) \text{ is } \alpha\text{-unusual}]]$$

$$\leq 4\alpha.$$

Thus

$$\mathbb{P}_{\mathbf{S}}[\mathbf{S} \text{ is } \alpha\text{-representative}] \geq 1 - 4\alpha. \tag{27}$$

We claim that if the training set $S$ is $\delta$-representative, then it holds for all $w \in \mathcal{H}$ that

$$\mathcal{L}_S^\gamma(w) \geq \frac{\mathcal{L}_{\mathcal{D}}^{(3/4)\gamma_i}(w)}{4} - \frac{2^{11}k + \ln(4/\delta)}{n} - 30\exp(-k\gamma_{i+1}^2/2^{14}).$$

To see this, consider an arbitrary such $S$ and a $w \in \mathcal{H}$. Sample $\mathbf{A}$ and $\mathbf{t}$ as in the previous section. Call $\mathbf{A}, \mathbf{t}$ *good* for $w$ if it satisfies both $\|h_{\mathbf{A},\mathbf{t}}(w)\|_2 \leq 4$ and $\mathcal{L}_{\mathbf{A}\mathcal{D}}^{(7/8)\gamma_i}(h_{\mathbf{A},\mathbf{t}}(w)) \geq \mathcal{L}_{\mathcal{D}}^{(3/4)\gamma_i}(w) - 25\exp(-k\gamma_{i+1}^2/2^{14})$. For ease of notation, let $G_w$ denote the set of $(A, t)$ that are good for $w$. Similarly, let $U_S$ denote the set of $A$ where $A$ is $\delta$-unusual for $S$.

For all $w \in \mathcal{H}$, $\gamma \in \Gamma_i$, $A$ and $t$, we have that

$$\mathcal{L}_S^\gamma(w) \geq \mathcal{L}_{AS}^{(7/8)\gamma_i}(h_{A,\mathbf{t}}(w)) - \mathbb{P}_{(\mathbf{x},\mathbf{y})\sim S}[\mathbf{y}\langle w, \mathbf{x}\rangle > \gamma \wedge \mathbf{y}\langle h_{A,\mathbf{t}}(w), A\mathbf{x}\rangle \leq (7/8)\gamma_i].$$

Thus

$$\mathcal{L}_S^\gamma(w) \geq \mathbb{E}_{\mathbf{A},\mathbf{t}}[\mathcal{L}_{\mathbf{A}S}^{(7/8)\gamma_i}(h_{\mathbf{A},\mathbf{t}}(w)) - \mathbb{P}_{(\mathbf{x},\mathbf{y})\sim S}[\mathbf{y}\langle w, \mathbf{x}\rangle > \gamma \wedge \mathbf{y}\langle h_{\mathbf{A},\mathbf{t}}(w), \mathbf{A}\mathbf{x}\rangle \leq (7/8)\gamma_i]]$$

$$\geq \mathbb{E}_{\mathbf{A},\mathbf{t}}[\mathcal{L}_{\mathbf{A}S}^{(7/8)\gamma_i}(h_{\mathbf{A},\mathbf{t}}(w)) \mid (\mathbf{A}, \mathbf{t}) \in G_w \wedge \mathbf{A} \notin U_S]\mathbb{P}_{\mathbf{A},\mathbf{t}}[(\mathbf{A}, \mathbf{t}) \in G_w \wedge \mathbf{A} \notin U_S] \tag{28}$$

$$- \mathbb{E}_{\mathbf{A},\mathbf{t}}[\mathbb{P}_{(\mathbf{x},\mathbf{y})\sim S}[\mathbf{y}\langle w, \mathbf{x}\rangle > \gamma \wedge \mathbf{y}\langle h_{\mathbf{A},\mathbf{t}}(w), \mathbf{A}\mathbf{x}\rangle \leq (7/8)\gamma_i]]. \tag{29}$$

For the term (28), we observe that conditioned on $(\mathbf{A}, \mathbf{t}) \in G_w$, we have that $h_{\mathbf{A},\mathbf{t}}(w) \in \mathcal{G}_0$ since $\|h_{\mathbf{A},\mathbf{t}}(w)\|_2 \leq 4$. Secondly, when $\mathbf{A} \notin U_S$, this implies by the definition of $\delta$-unusual that

$$\mathcal{L}_{AS}^{(7/8)\gamma_i}(h_{\mathbf{A},\mathbf{t}}(w)) \geq \frac{\mathcal{L}_{A\mathcal{D}}^{(7/8)\gamma_i}(h_{\mathbf{A},\mathbf{t}}(w))}{2} - \frac{2^{11}k + \ln(1/\delta)}{n}.$$

Hence

$$\mathbb{E}_{\mathbf{A},\mathbf{t}}[\mathcal{L}_{\mathbf{A}S}^{(7/8)\gamma_i}(h_{\mathbf{A},\mathbf{t}}(w)) \mid (\mathbf{A}, \mathbf{t}) \in G_w \wedge \mathbf{A} \notin U_S]\mathbb{P}_{\mathbf{A},\mathbf{t}}[(\mathbf{A}, \mathbf{t}) \in G_w \wedge \mathbf{A} \notin U_S] \geq$$

$$\mathbb{E}_{\mathbf{A},\mathbf{t}}\left[\frac{\mathcal{L}_{A\mathcal{D}}^{(7/8)\gamma_i}(h_{\mathbf{A},\mathbf{t}}(w))}{2} \,\middle|\, (\mathbf{A}, \mathbf{t}) \in G_w \wedge \mathbf{A} \notin U_S\right]\mathbb{P}_{\mathbf{A},\mathbf{t}}[(\mathbf{A}, \mathbf{t}) \in G_w \wedge \mathbf{A} \notin U_S] - \frac{2^{11}k + \ln(1/\delta)}{n}.$$

$$\tag{30}$$

Using again that $(\mathbf{A}, \mathbf{t}) \in G_w$, we have that (30) is at least

$$\frac{\mathcal{L}_{\mathcal{D}}^{(3/4)\gamma_i}(w)}{2}\mathbb{P}_{\mathbf{A},\mathbf{t}}[(\mathbf{A},\mathbf{t}) \in G_w \wedge \mathbf{A} \notin U_S] - \frac{2^{11}k + \ln(1/\delta)}{n} - 25\exp(-k\gamma_{i+1}^2/2^{14}). \quad (31)$$

We now bound $\mathbb{P}[(\mathbf{A}, \mathbf{t}) \in G_w]$ and $\mathbb{P}[\mathbf{A} \notin U_S]$. For this, we recall that $\|\mathbf{A}w\|_2^2 \sim (1/k)\chi_2^k$. Thus $\mathbb{E}[\|\mathbf{A}w\|_2^2] = 1$ and by Markov's, we get $\mathbb{P}[\|\mathbf{A}w\|_2^2 \geq 9] \leq 1/9$. Conditioned on $\|\mathbf{A}w\|_2^2 < 9$, we have $\|h_{\mathbf{A},\mathbf{t}}(w)\|_2 \leq \|\mathbf{A}w\|_2 + \|h_{\mathbf{A},\mathbf{t}}(w) - \mathbf{A}w\|_2 \leq \sqrt{9} + \sqrt{k(10\sqrt{k})^{-2}} < 4$. Next observe that

$$\mathcal{L}_{\mathbf{A}\mathcal{D}}^{(7/8)\gamma_i}(h_{\mathbf{A},\mathbf{t}}(w)) \geq \mathcal{L}_{\mathcal{D}}^{(3/4)\gamma_i}(w) - \mathbb{P}_{(\mathbf{x},\mathbf{y})\sim\mathcal{D}}[\mathbf{y}\langle w, \mathbf{x}\rangle \leq (3/4)\gamma_i \wedge \mathbf{y}\langle h_{\mathbf{A},\mathbf{t}}(w), \mathbf{A}\mathbf{x}\rangle > (7/8)\gamma_i].$$

We have by Lemma 5 that there is a constant $c > 0$ so that

$$\mathbb{E}_{\mathbf{A},\mathbf{t}}[\mathbb{P}_{(\mathbf{x},\mathbf{y})\sim\mathcal{D}}[\mathbf{y}\langle w, \mathbf{x}\rangle \leq (3/4)\gamma_i \wedge \mathbf{y}\langle h_{\mathbf{A},\mathbf{t}}(w), \mathbf{A}\mathbf{x}\rangle > (7/8)\gamma_i]] =$$
$$\mathbb{E}_{(\mathbf{x},\mathbf{y})\sim\mathcal{D}}[\mathbb{P}_{\mathbf{A},\mathbf{t}}[\mathbf{y}\langle w, \mathbf{x}\rangle \leq (3/4)\gamma_i \wedge \mathbf{y}\langle h_{\mathbf{A},\mathbf{t}}(w), \mathbf{A}\mathbf{x}\rangle > (7/8)\gamma_i]] \leq$$
$$\sup_{x\in\mathcal{X}:\langle w,x\rangle\leq(3/4)\gamma_i} \mathbb{P}_{\mathbf{A},\mathbf{t}}[\langle h_{\mathbf{A},\mathbf{t}}(w), \mathbf{A}x\rangle > (7/8)\gamma_i] \leq$$
$$c\exp(-k(\gamma_i/8)^2/c) \leq$$
$$c\exp(-k\gamma_{i+1}^2/(2^8 c)).$$

Thus by Markov's inequality, we conclude

$$\mathbb{P}_{\mathbf{A},\mathbf{t}}[\mathcal{L}_{\mathbf{A}\mathcal{D}}^{(7/8)\gamma_i}(h_{\mathbf{A},\mathbf{t}}(w)) < \mathcal{L}_{\mathcal{D}}^{(3/4)\gamma_i}(w) - 5c\exp(-k\gamma_{i+1}^2/(2^8 c))] \leq$$
$$\mathbb{P}_{\mathbf{A},\mathbf{t}}[\mathbb{P}_{(\mathbf{x},\mathbf{y})\sim\mathcal{D}}[\mathbf{y}\langle w, \mathbf{x}\rangle \leq (3/4)\gamma_i \wedge \mathbf{y}\langle h_{\mathbf{A},\mathbf{t}}(w), \mathbf{A}\mathbf{x}\rangle > (7/8)\gamma_i] > 5c\exp(-k\gamma_{i+1}^2/(2^8 c))] < 1/5.$$

Finally, since we assumed $S$ is $\delta$-representative, we have $\mathbb{P}_{\mathbf{A}}[\mathbf{A} \in U_S] \leq 1/4$ by definition of $\delta$-representative. We conclude by a union bound that

$$\mathbb{P}_{\mathbf{A},\mathbf{t}}[(\mathbf{A},\mathbf{t}) \in G_w \wedge \mathbf{A} \notin U_S] \geq 1 - 1/9 - 1/5 - 1/4 \geq 1/2.$$

In summary, we have shown that (31) is at least

$$\frac{\mathcal{L}_{\mathcal{D}}^{(3/4)\gamma_i}(w)}{2} \cdot \frac{1}{2} - \frac{2^{11}k + \ln(1/\delta)}{n} - 5c\exp(-k\gamma_{i+1}^2/(2^8 c)).$$

Recalling that $(28) \geq (31)$ gives

$$\mathbb{E}_{\mathbf{A},\mathbf{t}}[\mathcal{L}_{\mathbf{A}S}^{(7/8)\gamma_i}(h_{\mathbf{A},\mathbf{t}}(w)) \mid (\mathbf{A},\mathbf{t}) \in G_w \wedge \mathbf{A} \notin U_S]\mathbb{P}_{\mathbf{A},\mathbf{t}}[(\mathbf{A},\mathbf{t}) \in G_w \wedge \mathbf{A} \notin U_S] \geq$$
$$\frac{\mathcal{L}_{\mathcal{D}}^{(3/4)\gamma_i}(w)}{4} - \frac{2^{11}k + \ln(1/\delta)}{n} - 5c\exp(-k\gamma_{i+1}^2/(2^8 c)).$$

The term (29) can be bounded using Lemma 5 by

$$\mathbb{E}_{\mathbf{A},\mathbf{t}}[\mathbb{P}_{(\mathbf{x},\mathbf{y})\sim S}[\mathbf{y}\langle w, \mathbf{x}\rangle > \gamma \wedge \mathbf{y}\langle h_{\mathbf{A},\mathbf{t}}(w), \mathbf{A}\mathbf{x}\rangle \leq (7/8)\gamma_i]] =$$
$$\mathbb{E}_{(\mathbf{x},\mathbf{y})\sim S}[\mathbb{P}_{\mathbf{A},\mathbf{t}}[\mathbf{y}\langle w, \mathbf{x}\rangle > \gamma \wedge \mathbf{y}\langle h_{\mathbf{A},\mathbf{t}}(w), \mathbf{A}\mathbf{x}\rangle \leq (7/8)\gamma_i]] \leq$$
$$\sup_{x\in\mathcal{X}:\langle w,x\rangle>\gamma} \mathbb{P}_{\mathbf{A},\mathbf{t}}[\langle h_{\mathbf{A},\mathbf{t}}(w), \mathbf{A}x\rangle \leq (7/8)\gamma_i] \leq$$
$$c\exp(-k(\gamma - (7/8)\gamma_i)^2/c) \leq$$
$$c\exp(-k\gamma_i^2/(64c)) \leq$$
$$c\exp(-k\gamma_{i+1}^2/(2^8 c)).$$

In summary, we have shown that for $(\delta/4)$-representative $S$, it holds for all $w \in \mathcal{H}$ that

$$\mathcal{L}_S^{\gamma}(w) \geq \frac{\mathcal{L}_{\mathcal{D}}^{(3/4)\gamma_i}(w)}{4} - \frac{2^{11}k + \ln(4/\delta)}{n} - 6c\exp(-k\gamma_{i+1}^2/(2^8 c)).$$

We finally conclude from (27) that with probability at least $1 - \delta$ over $\mathbf{S}$, it holds for all $w \in \mathcal{H}$ that

$$\mathcal{L}_{\mathbf{S}}^{\gamma}(w) \geq \frac{\mathcal{L}_{\mathcal{D}}^{(3/4)\gamma_i}(w)}{4} - \frac{2^{11}k + \ln(4/\delta)}{n} - 6c\exp(-k\gamma_{i+1}^2/(2^8 c)).$$

Picking $k = 2^8 c\gamma_{i+1}^{-2}\ln(\gamma_{i+1}^2 n)$ finally results in

$$\mathcal{L}_{\mathbf{S}}^{\gamma}(w) \geq \frac{\mathcal{L}_{\mathcal{D}}^{(3/4)\gamma_i}(w)}{4} - \frac{2^{20}c\ln(\gamma_{i+1}^2 n)}{\gamma_{i+1}^2 n} - \frac{2\ln(e/\delta)}{n}.$$

This completes the proof. $\square$

# E   Auxiliary Results

In this section, we prove a number of auxiliary results used throughout the paper. For this, we need the following concentration inequality:

**Theorem 17** (Wainwright [2019], example 2.11). *Let $Y \sim \chi_k^2$, then for any $x \in (0,1)$ it holds that*

$$\mathbb{P}\left[\left|\frac{Y}{k} - 1\right| \geq x\right] \leq 2\exp(-kx^2/8).$$

**Restatement of Claim 1.** *For any $0 < \delta < 1$, it holds with probability $1 - \delta$ over $\mathbf{S} \sim \mathcal{D}^n$ that (17) and (18) simultaneously hold for all $(\Gamma_i, L_j)$ and $\Gamma_i$, with slightly different constants c.*

*Proof.* Let $(\gamma_i, \gamma_{i+1}]$ be such that $\gamma_{i+1} := 2^i n^{-1/2}$. Similarly, let $(\ell_j, \ell_{j+1}]$ be such that $\ell_{j+1} := 2^j n^{-1}$. Do a union bound over all $(\Gamma_i, L_j)$ for $i = 1, \ldots, \lg_2(c_\gamma n^{1/2})$ and $j = 0, \ldots, \lg_2 n$ with $\delta_{i,j} := (\delta/e)^3 \exp(-\gamma_{i+1}^{-2} \ln(e/\ell_{j+1}))$ in (17). We see that

$$\sum_{i=1}^{\lg_2(c_\gamma n^{1/2})} \sum_{j=0}^{\lg_2 n} \delta_{i,j} = \sum_{i=1}^{\lg_2(c_\gamma n^{1/2})} \sum_{j=0}^{\lg_2 n} (\delta/e)^3 \exp(-\gamma_{i+1}^{-2} \ln(e/\ell_{j+1}))$$

$$= \sum_{i=1}^{\lg_2(c_\gamma n^{1/2})} \sum_{j=0}^{\lg_2 n} (\delta/e)^3 \exp(-2^{-2i} n \ln(en2^{-j}))$$

$$= \sum_{i=1}^{\lg_2(c_\gamma n^{1/2})} \sum_{j=0}^{\lg_2 n} (\delta/e)^3 (en2^{-j})^{-2^{-2i}n}.$$

Doing the substitutions $j \leftarrow \lg_2 n - j$ and $i \leftarrow \lg_2(c_\gamma n^{1/2}) + 1 - i$, this equals

$$= \sum_{i=1}^{\lg_2(c_\gamma n^{1/2})} \sum_{j=0}^{\lg_2 n} (\delta/e)^3 (e2^j)^{-2^{2i-2}c_\gamma^{-2}}$$

$$\leq \sum_{i=1}^{\lg_2(c_\gamma n^{1/2})} \sum_{j=0}^{\lg_2 n} (\delta/e)^3 e^{-2^{2i-2}} 2^{-j}$$

$$\leq \sum_{i=1}^{\lg_2(c_\gamma n^{1/2})} 2(\delta/e)^3 e^{-2^{2i-2}}$$

$$\leq \delta/2.$$

Similarly, do a union bound over all $\Gamma_i$ with $\delta_i := (\delta/e)^3 \exp(-\gamma_{i+1}^{-2} \ln(e\gamma_{i+1}^2 n))$ in (18). We have

$$\sum_{i=1}^{\lg_2(c_\gamma n^{1/2})} \delta_i = \sum_{i=1}^{\lg_2(c_\gamma n^{1/2})} (\delta/e)^3 \exp(-\gamma_{i+1}^{-2} \ln(e\gamma_{i+1}^2 n))$$

$$\leq \sum_{i=1}^{\lg_2(c_\gamma n^{1/2})} (\delta/e)^3 \exp(-\ln(e\gamma_{i+1}^2 n))$$

$$= \sum_{i=1}^{\lg_2(c_\gamma n^{1/2})} (\delta/e)^3 \frac{1}{e\gamma_{i+1}^2 n}$$

$$= \sum_{i=1}^{\lg_2(c_\gamma n^{1/2})} (\delta/e)^3 \frac{n}{e2^{2i}n}$$

$$\leq (\delta/e)^3$$

$$\leq \delta/2.$$

We thus have that with probability at least $1 - \delta$ that for all $(\Gamma_i, L_j)$, we have

$$\sup_{w \in \mathcal{H}(\Gamma_i, L_j), \gamma \in \Gamma_i} |\mathcal{L}_{\mathcal{D}}(w) - \mathcal{L}_{\mathbf{S}}^{\gamma}(w)| \leq$$

$$c \left( \sqrt{\ell_{j+1} \left( \frac{\ln(e/\ell_{j+1})}{\gamma_{i+1}^2 n} + \frac{\ln(e/\delta_{i,j})}{n} \right)} + \frac{\ln(e/\ell_{j+1})}{\gamma_{i+1}^2 n} + \frac{\ln(e/\delta_{i,j})}{n} \right) =$$

$$c \left( \sqrt{\ell_{j+1} \left( \frac{\ln(e/\ell_{j+1})}{\gamma_{i+1}^2 n} + \frac{\ln(e/\delta_{i,j})}{n} \right)} + 2 \cdot \frac{\ln(e/\ell_{j+1})}{\gamma_{i+1}^2 n} + 3 \cdot \frac{\ln(e/\delta)}{n} \right).$$

and for all $\Gamma_i$, we have

$$\inf_{w \in \mathcal{H}} \mathcal{L}_{\mathbf{S}}^{\gamma_i}(w) \geq \frac{\mathcal{L}_{\mathcal{D}}^{(3/4)\gamma_i}(w)}{4} - c \left( \frac{\ln(e\gamma_{i+1}^2 n)}{\gamma_{i+1}^2 n} - \frac{\ln(e/\delta_i)}{n} \right)$$

$$= \frac{\mathcal{L}_{\mathcal{D}}^{(3/4)\gamma_i}(w)}{4} - c \left( 2 \cdot \frac{\ln(e\gamma_{i+1}^2 n)}{\gamma_{i+1}^2 n} - 3 \cdot \frac{\ln(e/\delta)}{n} \right).$$

$\square$

**Restatement of Claim 2.** *For any $0 < \delta < 1$ and training set S, if* (17) *and* (18) *hold simultaneously for all $(\Gamma_i, L_j)$ and $\Gamma_i$, then* (16) *holds for all $\gamma \in (n^{-1/2}, c_\gamma]$ and all $w \in \mathcal{H}$ for large enough constant $c > 1$ in* (16).

*Proof.* Let $0 < \delta < 1$ and assume as in the claim that (17) and (18) holds for all $(\Gamma_i, L_j)$ and $\Gamma_i$. Now consider an arbitrary $\gamma \in (n^{-1/2}, c_\gamma]$ and $w \in \mathcal{H}$. Let $i$ and $j$ be such that $\gamma \in (\gamma_i, \gamma_{i+1}]$ and $\mathcal{L}_{\mathcal{D}}^{(3/4)\gamma_i}(w) \in (\ell_j, \ell_{j+1}]$ with $\gamma_{i+1} = 2^i n^{-1/2}$ and $\ell_{j+1} = 2^j n^{-1}$. We consider two cases. Let $c_4 > 1$ be the constant in Lemma 4. First, if

$$\mathcal{L}_{\mathcal{D}}^{(3/4)\gamma_i}(w) \leq 16 \cdot c_4 \cdot \left( \frac{\ln(e\gamma_{i+1}^2 n)}{\gamma_{i+1}^2 n} + \frac{\ln(e/\delta)}{n} \right),$$

then since $\mathcal{L}_{\mathcal{D}}(w) \leq \mathcal{L}_{\mathcal{D}}^{(3/4)\gamma_i}(w)$ and $\gamma \leq \gamma_{i+1}$ (using that $\ln(e\gamma^2 n)/(\gamma^2 n)$ is decreasing in $\gamma$ for $\gamma \geq n^{-1/2}$), we have already shown (16) for sufficiently large constant $c$ in (16). So assume this is not the case. Our goal is to show that $\ell_{j+1}$ and $\mathcal{L}_S^{\gamma}(w)$ are within constant factors of each other so that we may replace occurrences of $\ell_{j+1}$ by $\mathcal{L}_S^{\gamma}(w)$ in (17). We first see that our assumption implies

$$\ell_{j+1} \geq \mathcal{L}_{\mathcal{D}}^{(3/4)\gamma_i}(w) \geq 16 \cdot c_4 \cdot \left( \frac{\ln(e\gamma_{i+1}^2 n)}{\gamma_{i+1}^2 n} + \frac{\ln(e/\delta)}{n} \right) \geq \frac{1}{\gamma_{i+1}^2 n} > n^{-1}. \quad (32)$$

This also implies $j \neq 0$ and hence $\ell_{j+1} = 2\ell_j$ and therefore $\ell_{j+1} \leq 2\mathcal{L}_{\mathcal{D}}^{(3/4)\gamma_i}(w)$. Letting $c_3$ be the constant in Lemma 3, we get from (17) and (32), that

$$\mathcal{L}_S^{\gamma}(w) \leq \mathcal{L}_{\mathcal{D}}(w) + c_3 \cdot \left( \sqrt{\ell_{j+1} \left( \frac{\ln(e/\ell_{j+1})}{\gamma_{i+1}^2 n} + \frac{\ln(e/\delta)}{n} \right)} + \frac{\ln(e/\ell_{j+1})}{\gamma_{i+1}^2 n} + \frac{\ln(e/\delta)}{n} \right)$$

$$\leq \mathcal{L}_{\mathcal{D}}^{(3/4)\gamma_i}(w) + c_3 \cdot \left( \sqrt{2\mathcal{L}_{\mathcal{D}}^{(3/4)\gamma_i}(w) \left( \frac{\ln(e\gamma_{i+1}^2 n)}{\gamma_{i+1}^2 n} + \frac{\ln(e/\delta)}{n} \right)} + \frac{\ln(e\gamma_{i+1}^2 n)}{\gamma_{i+1}^2 n} + \frac{\ln(e/\delta)}{n} \right)$$

$$\leq \mathcal{L}_{\mathcal{D}}^{(3/4)\gamma_i}(w) + c_3 \cdot \left( \sqrt{2\mathcal{L}_{\mathcal{D}}^{(3/4)\gamma_i}(w)\mathcal{L}_{\mathcal{D}}^{(3/4)\gamma_i}(w)} + \mathcal{L}_{\mathcal{D}}^{(3/4)\gamma_i}(w) \right)$$

$$\leq 3 \cdot c_3 \cdot \mathcal{L}_{\mathcal{D}}^{(3/4)\gamma_i}(w).$$

We thus also have $\ell_{j+1} \geq \mathcal{L}_{\mathcal{D}}^{(3/4)\gamma_i}(w) \geq (3 \cdot c_3)^{-1} \mathcal{L}_S^{\gamma}(w)$. Inserting this and (32) in (17) gives

$$\mathcal{L}_{\mathcal{D}}(w) \leq \mathcal{L}_S^{\gamma}(w) + c_3 \left( \sqrt{\ell_{j+1} \left( \frac{\ln(3ec_3/\mathcal{L}_S^{\gamma}(w))}{\gamma_{i+1}^2 n} + \frac{\ln(e/\delta)}{n} \right)} + \frac{\ln(e\gamma_{i+1}^2 n)}{\gamma_{i+1}^2 n} + \frac{\ln(e/\delta)}{n} \right).$$

$$(33)$$

Finally from (18) and $\gamma \geq \gamma_i$, we have

$$\mathcal{L}_S^\gamma(w) \geq \mathcal{L}_S^{\gamma_i}(w)$$

$$\geq \frac{\mathcal{L}_\mathcal{D}^{(3/4)\gamma_i}(w)}{4} - c_4 \left( \frac{\ln(e\gamma_{i+1}^2 n)}{\gamma_{i+1}^2 n} - \frac{\ln(e/\delta)}{n} \right)$$

$$\geq \frac{\ell_{j+1}}{8} - c_4 \left( \frac{\ln(e\gamma_{i+1}^2 n)}{\gamma_{i+1}^2 n} - \frac{\ln(e/\delta)}{n} \right).$$

From (32), this is at least $\ell_{j+1}/16$ and thus $\ell_{j+1} \leq 16\mathcal{L}_S^\gamma(w)$. Inserting this in (33) finally gives us

$$\mathcal{L}_\mathcal{D}(w) \leq \mathcal{L}_S^\gamma(w) + c_3 \left( \sqrt{16\mathcal{L}_S^\gamma(w) \left( \frac{\ln(2ec_3/\mathcal{L}_S^\gamma(w))}{\gamma_{i+1}^2 n} + \frac{\ln(e/\delta)}{n} \right)} + \frac{\ln(e\gamma_{i+1}^2 n)}{\gamma_{i+1}^2 n} + \frac{\ln(e/\delta)}{n} \right).$$

Since $\gamma \leq \gamma_{i+1}$, this completes the proof of Claim 2 for sufficiently large $c > 0$ in (16). $\qquad\square$

**Restatement of Lemma 5.** *There is a constant $c > 0$, such that for any integer $k \geq 1$, $w \in \mathcal{H}$, $x \in \mathcal{X}$ and any $\gamma \in (0, 1]$, it holds that $\mathbb{P}_{\mathbf{A},\mathbf{t}}[|\langle h_{\mathbf{A},\mathbf{t}}(w), \mathbf{A}x \rangle - \langle w, x \rangle| > \gamma] < c\exp(-\gamma^2 k/c)$.*

*Proof.* We start by observing that $\|\mathbf{A}w\|_2^2$, $\|\mathbf{A}x\|_2^2$ and $\|\mathbf{A}(w-x)\|_2^2/\|w-x\|_2^2$ are all $(1/k)\chi_k^2$ distributed. Using Theorem 17 with $x = \gamma/3$, we have with probability at least $1 - 6\exp(-k\gamma^2/72)$ that $\|\mathbf{A}w\|_2^2 \in 1 \pm \gamma/3$, $\|\mathbf{A}x\|_2^2 \in 1 \pm \gamma/3$ and $\|\mathbf{A}(w-x)\|_2^2 \in \|w-x\|_2^2(1 \pm \gamma/3)$. By the polar identity, this implies

$$\langle \mathbf{A}w, \mathbf{A}w \rangle = \frac{1}{4}\left( \|\mathbf{A}w\|_2^2 + \|\mathbf{A}x\|_2^2 - \|\mathbf{A}(w-x)\|_2^2 \right)$$

$$\in \frac{1}{4}\left( \|w\|_2^2 + \|x\|_2^2 - \|w-x\|_2^2 \right) \pm \frac{\gamma}{12}\left( \|w\|_2^2 + \|x\|_2^2 + \|w-x\|_2^2 \right)$$

$$\subseteq \langle w, x \rangle \pm \frac{\gamma}{12}(1 + 1 + 4)$$

$$= \langle w, x \rangle \pm \frac{\gamma}{2}.$$

Let us condition on an outcome $A$ of $\mathbf{A}$ satisfying the above. We then observe that

$$\langle h_{A,\mathbf{t}}(w), Ax \rangle = \langle h_{A,\mathbf{t}}(w) - Aw, Ax \rangle + \langle Aw, Ax \rangle.$$

By the randomized rounding procedure, we have that each coordinate $i$ satisfies $\mathbb{E}_{\mathbf{t}_i}[(h_{A,\mathbf{t}}(w))_i] = (Aw)_i$. Moreover, these coordinates are independent. Letting $\Delta_i = (h_{A,\mathbf{t}}(w))_i - (Aw)_i$, we then have that $\mathbb{E}[\Delta_i] = 0$ and that $\Delta_i$ lies in an interval of length $(10\sqrt{k})^{-1}$. Hoeffding's inequality implies

$$\mathbb{P}_{\mathbf{t}}[|\langle h_{A,\mathbf{t}}(w) - Aw, Ax \rangle| > \gamma/2] = \mathbb{P}_{\Delta_1,\ldots,\Delta_k}\left[ \left| \sum_{i=1}^k \Delta_i(Ax)_i \right| > \gamma/2 \right]$$

$$< 2\exp\left( -\frac{2(\gamma/2)^2}{\sum_{i=1}^k (10\sqrt{k})^{-2}(Ax)_i^2} \right)$$

$$= 2\exp\left( -\frac{50\gamma^2 k}{\|Ax\|_2^2} \right)$$

$$\leq 2\exp\left( -25\gamma^2 k \right).$$

In summary, it holds with probability at least $1 - 6\exp(-k\gamma^2/72) - 2\exp(-25\gamma^2 k) \geq 1 - 7\exp(-k\gamma^2/72)$ that

$$|\langle h_{\mathbf{A},\mathbf{t}}(w), \mathbf{A}x \rangle - \langle w, x \rangle| \leq |\langle h_{\mathbf{A},\mathbf{t}}(w), \mathbf{A}x \rangle - \langle \mathbf{A}w, \mathbf{A}x \rangle| + |\langle \mathbf{A}w, \mathbf{A}x \rangle - \langle w, x \rangle|$$

$$\leq |\langle h_{\mathbf{A},\mathbf{t}}(w) - \mathbf{A}w, \mathbf{A}x \rangle| + \gamma/2$$

$$< \gamma.$$

$\qquad\square$

**Remark 18.** *The value $p((Aw)_i)$ chosen such that $\mathbb{E}_{\mathbf{t}}[(h_{A,\mathbf{t}}(w))_i] = (Aw)_i$, i.e satisfying that*

$$(Aw)_i = p((Aw)_i)\left(\frac{1}{2\cdot 10\sqrt{k}} + \frac{z_i}{10\sqrt{k}}\right) + (1 - p((Aw)_i))\left(\frac{1}{2\cdot 10\sqrt{k}} + \frac{z_i + 1}{10\sqrt{k}}\right),$$

*where $z_i$ is an integer such that*

$$(1/2)(10\sqrt{k})^{-1} + z_i(10\sqrt{k})^{-1} \leq (Aw)_i < (1/2)(10\sqrt{k})^{-1} + (z_i + 1)(10\sqrt{k})^{-1}.$$

*is nonnegative, and less than 1.*

*Proof.* By definition of $z_i$.

$$((1/2)(10\sqrt{k})^{-1} + z_i(10\sqrt{k})^{-1}) + (1 - p((Aw)_i))(10\sqrt{k})^{-1} = (Aw)_i \Rightarrow$$
$$(Aw)_i - ((1/2)(10\sqrt{k})^{-1} + z_i(10\sqrt{k})^{-1}) = (1 - p((Aw)_i))(10\sqrt{k})^{-1}.$$

By definition of $z_i$, we have that the left hand side is a number in $[0, (10\sqrt{k})^{-1}]$ and thus we conclude

$$(1 - p((Aw)_i)) \in [0, 1] \Rightarrow p((Aw)_i) \in [0, 1].$$

$\square$

**Restatement of Remark 6.** *For any training set $S$ and distribution $\mathcal{D}$ over $\mathcal{X} \times \{-1, 1\}$, we have*

$$\mathbb{E}_{\mathbf{A},\mathbf{t}}[\mathbb{P}_{(\mathbf{x},\mathbf{y})\sim\mathcal{D}}[\mathbf{y}\langle h_{\mathbf{A},\mathbf{t}}(w), \mathbf{Ax}\rangle > \gamma_i/2 \wedge \mathbf{y}\langle w, \mathbf{x}\rangle \leq 0]] \leq \mathbb{E}_{(\mathbf{x},\mathbf{y})\sim\mathcal{D}}[\phi(\mathbf{y}\langle w, \mathbf{x}\rangle)]$$
$$\mathbb{E}_{\mathbf{A},\mathbf{t}}[\mathbb{P}_{(\mathbf{x},\mathbf{y})\sim S}[\mathbf{y}\langle h_{\mathbf{A},\mathbf{t}}(w), \mathbf{Ax}\rangle > \gamma_i/2 \wedge \mathbf{y}\langle w, \mathbf{x}\rangle \leq \gamma]] \geq \mathbb{E}_{(\mathbf{x},\mathbf{y})\sim S}[\phi(\mathbf{y}\langle w, \mathbf{x}\rangle)]$$
$$\mathbb{E}_{\mathbf{A},\mathbf{t}}[\mathbb{P}_{(\mathbf{x},\mathbf{y})\sim S}[\mathbf{y}\langle h_{\mathbf{A},\mathbf{t}}(w), \mathbf{Ax}\rangle \leq \gamma_i/2 \wedge \mathbf{y}\langle w, \mathbf{x}\rangle > \gamma]] \leq \mathbb{E}_{(\mathbf{x},\mathbf{y})\sim S}[\rho(\mathbf{y}\langle w, \mathbf{x}\rangle)]$$
$$\mathbb{E}_{\mathbf{A},\mathbf{t}}[\mathbb{P}_{(\mathbf{x},\mathbf{y})\sim\mathcal{D}}[\mathbf{y}\langle h_{\mathbf{A},\mathbf{t}}(w), \mathbf{Ax}\rangle \leq \gamma_i/2 \wedge \mathbf{y}\langle w, \mathbf{x}\rangle > 0]] \geq \mathbb{E}_{(\mathbf{x},\mathbf{y})\sim\mathcal{D}}[\rho(\mathbf{y}\langle w, \mathbf{x}\rangle)].$$

In the proof, we will need the following monotonicity properties

**Claim 4.** *We have $\mathbb{P}_{\mathbf{A},\mathbf{t}}[y\langle h_{\mathbf{A},\mathbf{t}}(w), \mathbf{A}x\rangle > \gamma_i/2 \mid y\langle w, x\rangle = \alpha_1] \leq \mathbb{P}_{\mathbf{A},\mathbf{t}}[y\langle h_{\mathbf{A},\mathbf{t}}(w), \mathbf{A}x\rangle > \gamma_i/2 \mid y\langle w, x\rangle = \alpha_2]$ for any $0 \leq \alpha_1 \leq \alpha_2 \leq \gamma_i$.*

**Claim 5.** *We have $\mathbb{P}_{\mathbf{A},\mathbf{t}}[y\langle h_{\mathbf{A},\mathbf{t}}(w), \mathbf{A}x\rangle \leq \gamma_i/2 \mid y\langle w, x\rangle = \alpha_2] \leq \mathbb{P}_{\mathbf{A},\mathbf{t}}[y\langle h_{\mathbf{A},\mathbf{t}}(w), \mathbf{A}x\rangle \leq \gamma_i/2 \mid y\langle w, x\rangle = \alpha_1]$ for any $0 < \alpha_1 \leq \alpha_2 \leq \gamma_i$.*

First we will prove Remark 6 using the two claims. Afterward, we will prove Claim 4 and Claim 5.

*Proof of Remark 6.* For convenience, let us recall the definitions of $\phi$ and $\rho$:

$$\phi(\alpha) = \begin{cases} \mathbb{P}_{\mathbf{A},\mathbf{t}}[y\langle h_{\mathbf{A},\mathbf{t}}(w), \mathbf{A}x\rangle > \gamma_i/2 \mid y\langle w, x\rangle = \alpha] & \text{if } -c_\gamma \leq \alpha \leq 0 \\ \frac{(\gamma_i - \alpha)}{\gamma_i}\mathbb{P}_{\mathbf{A},\mathbf{t}}[y\langle h_{\mathbf{A},\mathbf{t}}(w), \mathbf{A}x\rangle > \gamma_i/2 \mid y\langle w, x\rangle = 0] & \text{if } 0 < \alpha \leq \gamma_i \\ 0 & \text{if } \gamma_i < \alpha \leq c_\gamma \end{cases}$$

$$\rho(\alpha) = \begin{cases} \mathbb{P}_{\mathbf{A},\mathbf{t}}[y\langle h_{\mathbf{A},\mathbf{t}}(w), \mathbf{A}x\rangle \leq \gamma_i/2 \mid y\langle w, x\rangle = \alpha] & \text{if } \gamma_i < \alpha \leq c_\gamma \\ \frac{\alpha}{\gamma_i}\mathbb{P}_{\mathbf{A},\mathbf{t}}[y\langle h_{\mathbf{A},\mathbf{t}}(w), \mathbf{A}x\rangle \leq \gamma_i/2 \mid y\langle w, x\rangle = \gamma_i] & \text{if } 0 < \alpha \leq \gamma_i \\ 0 & \text{if } -c_\gamma \leq \alpha \leq 0. \end{cases}$$

We handle each of the inequalities in turn. First we see that

$$\mathbb{E}_{\mathbf{A},\mathbf{t}}[\mathbb{P}_{(\mathbf{x},\mathbf{y})\sim\mathcal{D}}[\mathbf{y}\langle h_{\mathbf{A},\mathbf{t}}(w), \mathbf{Ax}\rangle > \gamma_i/2 \wedge \mathbf{y}\langle w, \mathbf{x}\rangle \leq 0] =$$
$$\mathbb{E}_{(\mathbf{x},\mathbf{y})\sim\mathcal{D}}[\mathbb{P}_{\mathbf{A},\mathbf{t}}[\mathbf{y}\langle h_{\mathbf{A},\mathbf{t}}(w), \mathbf{Ax}\rangle > \gamma_i/2 \wedge \mathbf{y}\langle w, \mathbf{x}\rangle \leq 0]] \leq$$
$$\mathbb{E}_{(\mathbf{x},\mathbf{y})\sim\mathcal{D}}[\phi(\mathbf{y}\langle w, \mathbf{x}\rangle)].$$

Here the inequality follows from the observations that $\phi(y\langle w, x\rangle) \geq 0$ for $y\langle w, x\rangle > 0$, whereas $\mathbb{P}_{\mathbf{A},\mathbf{t}}[\mathbf{y}\langle h_{\mathbf{A},\mathbf{t}}(w), \mathbf{Ax}\rangle > \gamma_i/2 \wedge \mathbf{y}\langle w, \mathbf{x}\rangle \leq 0] = 0$ for such $y\langle w, x\rangle$. Similarly for $y\langle w, x\rangle = \alpha \leq 0$, we have $\phi(y\langle w, x\rangle) = \mathbb{P}_{\mathbf{A},\mathbf{t}}[y\langle h_{\mathbf{A},\mathbf{t}}(w), \mathbf{A}x\rangle > \gamma_i/2 \mid y\langle w, x\rangle = \alpha] = \mathbb{P}_{\mathbf{A},\mathbf{t}}[y\langle h_{\mathbf{A},\mathbf{t}}(w), \mathbf{A}x\rangle > \gamma_i/2 \wedge y\langle w, x\rangle \leq 0 \mid y\langle w, x\rangle = \alpha]$.

Similarly, we have

$$\mathbb{E}_{\mathbf{A},\mathbf{t}}[\mathbb{P}_{(\mathbf{x},\mathbf{y})\sim S}[\mathbf{y}\langle h_{\mathbf{A},\mathbf{t}}(w),\mathbf{A}\mathbf{x}\rangle > \gamma_i/2 \wedge \mathbf{y}\langle w,\mathbf{x}\rangle \le \gamma] =$$
$$\mathbb{E}_{(\mathbf{x},\mathbf{y})\sim S}[\mathbb{P}_{\mathbf{A},\mathbf{t}}[\mathbf{y}\langle h_{\mathbf{A},\mathbf{t}}(w),\mathbf{A}\mathbf{x}\rangle > \gamma_i/2 \wedge \mathbf{y}\langle w,\mathbf{x}\rangle \le \gamma]] \ge$$
$$\mathbb{E}_{(\mathbf{x},\mathbf{y})\sim S}[\mathbb{P}_{\mathbf{A},\mathbf{t}}[\mathbf{y}\langle h_{\mathbf{A},\mathbf{t}}(w),\mathbf{A}\mathbf{x}\rangle > \gamma_i/2 \wedge \mathbf{y}\langle w,\mathbf{x}\rangle \le \gamma_i]] \ge$$
$$\mathbb{E}_{(\mathbf{x},\mathbf{y})\sim S}[\phi(\mathbf{y}\langle w,\mathbf{x}\rangle)].$$

The last inequality follows by observing that if $y\langle w,x\rangle > \gamma_i$, we have $\phi(y\langle w,x\rangle) = 0$ and $\mathbb{P}_{\mathbf{A},\mathbf{t}}[y\langle h_{\mathbf{A},\mathbf{t}}(w),\mathbf{A}x\rangle > \gamma_i/2 \wedge y\langle w,x\rangle \le \gamma_i] = 0$. For $\alpha = y\langle w,x\rangle$ with $0 < \alpha \le \gamma_i$, we have $\phi(\alpha) = \frac{\gamma_i-\alpha}{\gamma_i}\mathbb{P}_{\mathbf{A},\mathbf{t}}[y\langle h_{\mathbf{A},\mathbf{t}}(w),\mathbf{A}x\rangle > \gamma_i/2 \mid y\langle w,x\rangle = 0] \le \mathbb{P}_{\mathbf{A},\mathbf{t}}[y\langle h_{\mathbf{A},\mathbf{t}}(w),\mathbf{A}x\rangle > \gamma_i/2 \mid y\langle w,x\rangle = \alpha] = \mathbb{P}_{\mathbf{A},\mathbf{t}}[y\langle h_{\mathbf{A},\mathbf{t}}(w),\mathbf{A}x\rangle > \gamma_i/2 \wedge y\langle w,x\rangle \le \gamma_i \mid y\langle w,x\rangle = \alpha]$. This uses the monotonicity in Claim 4. Finally for $y\langle w,x\rangle = \alpha \le 0$, the two coincide as in the above argument.

Symmetric arguments for $\rho$ gives

$$\mathbb{E}_{\mathbf{A},\mathbf{t}}[\mathbb{P}_{(\mathbf{x},\mathbf{y})\sim S}[\mathbf{y}\langle h_{\mathbf{A},\mathbf{t}}(w),\mathbf{A}\mathbf{x}\rangle \le \gamma_i/2 \wedge \mathbf{y}\langle w,\mathbf{x}\rangle > \gamma] =$$
$$\mathbb{E}_{(\mathbf{x},\mathbf{y})\sim S}[\mathbb{P}_{\mathbf{A},\mathbf{t}}[\mathbf{y}\langle h_{\mathbf{A},\mathbf{t}}(w),\mathbf{A}\mathbf{x}\rangle \le \gamma_i/2 \wedge \mathbf{y}\langle w,\mathbf{x}\rangle > \gamma] \le$$
$$\mathbb{E}_{(\mathbf{x},\mathbf{y})\sim S}[\mathbb{P}_{\mathbf{A},\mathbf{t}}[\mathbf{y}\langle h_{\mathbf{A},\mathbf{t}}(w),\mathbf{A}\mathbf{x}\rangle \le \gamma_i/2 \wedge \mathbf{y}\langle w,\mathbf{x}\rangle > \gamma_i] \le$$
$$\mathbb{E}_{(\mathbf{x},\mathbf{y})\sim S}[\rho(\mathbf{y}\langle w,\mathbf{x}\rangle)].$$

Here the last inequality follows from the following considerations. For $y\langle w,x\rangle = \alpha$ with $\alpha \le \gamma_i$, we have that $\mathbb{P}_{\mathbf{A},\mathbf{t}}[y\langle h_{\mathbf{A},\mathbf{t}}(w),\mathbf{A}x\rangle \le \gamma_i/2 \wedge y\langle w,x\rangle > \gamma_i] = 0$ and $\rho$ is always non-negative. For $\alpha > \gamma_i$, we have $\rho(\alpha) = \mathbb{P}_{\mathbf{A},\mathbf{t}}[y\langle h_{\mathbf{A},\mathbf{t}}(w),\mathbf{A}x\rangle \le \gamma_i/2 \mid y\langle w,x\rangle = \alpha] = \mathbb{P}_{\mathbf{A},\mathbf{t}}[y\langle h_{\mathbf{A},\mathbf{t}}(w),\mathbf{A}x\rangle \le \gamma_i/2 \wedge y\langle w,x\rangle > \gamma_i \mid y\langle w,x\rangle = \alpha]$ and the two coincide.

Finally, we have

$$\mathbb{E}_{\mathbf{A},\mathbf{t}}[\mathbb{P}_{(\mathbf{x},\mathbf{y})\sim \mathcal{D}}[\mathbf{y}\langle h_{\mathbf{A},\mathbf{t}}(w),\mathbf{A}\mathbf{x}\rangle \le \gamma_i/2 \wedge \mathbf{y}\langle w,\mathbf{x}\rangle > 0] =$$
$$\mathbb{E}_{(\mathbf{x},\mathbf{y})\sim \mathcal{D}}[\mathbb{P}_{\mathbf{A},\mathbf{t}}[\mathbf{y}\langle h_{\mathbf{A},\mathbf{t}}(w),\mathbf{A}\mathbf{x}\rangle \le \gamma_i/2 \wedge \mathbf{y}\langle w,\mathbf{x}\rangle > 0] \ge$$
$$\mathbb{E}_{(\mathbf{x},\mathbf{y})\sim \mathcal{D}}[\rho(\mathbf{y}\langle w,\mathbf{x}\rangle)].$$

Here the inequality follows by observing that for $y\langle w,x\rangle = \alpha$ with $\alpha \le 0$, both $\rho(\alpha)$ and $\mathbb{P}_{\mathbf{A},\mathbf{t}}[y\langle h_{\mathbf{A},\mathbf{t}}(w),\mathbf{A}x\rangle \le \gamma_i/2 \wedge y\langle w,x\rangle > 0]$ are 0. For $0 \le \alpha \le \gamma_i$ we have by definition that $\rho(\alpha) = \frac{\alpha}{\gamma_i}\mathbb{P}_{\mathbf{A},\mathbf{t}}[y\langle h_{\mathbf{A},\mathbf{t}}(w),\mathbf{A}x\rangle \le \gamma_i/2 \mid y\langle w,x\rangle = \gamma_i] \le \mathbb{P}_{\mathbf{A},\mathbf{t}}[y\langle h_{\mathbf{A},\mathbf{t}}(w),\mathbf{A}x\rangle \le \gamma_i/2 \mid y\langle w,x\rangle = \alpha] = \mathbb{P}_{\mathbf{A},\mathbf{t}}[y\langle h_{\mathbf{A},\mathbf{t}}(w),\mathbf{A}x\rangle \le \gamma_i/2 \wedge y\langle w,x\rangle > 0 \mid y\langle w,x\rangle = \alpha]$, where we used that $\mathbb{P}_{\mathbf{A},\mathbf{t}}[y\langle h_{\mathbf{A},\mathbf{t}}(w),\mathbf{A}x\rangle \le \gamma_i/2 \mid y\langle w,x\rangle = \alpha]$ is decreasing in $\alpha$ (as stated in Claim 5). Finally, for $\alpha > \gamma_i$, the two coincide as above. $\qquad\square$

*Proof of Claim 4.* Let $w_1, x_1, y_1$ be such that $\alpha_1 := y_1\langle w_1, x_1\rangle$ and let $w_2, x_2, y_2$ be such that $\alpha_2 := y_2\langle w_2, x_2\rangle$. Consider sampling $\mathbf{X}_i, \mathbf{Y}_i \sim \mathcal{N}(0, 1/k)$ independently. Also sample offsets $\mathbf{t}'_1, \ldots, \mathbf{t}'_k$ uniformly and independently in $[0,1]$ and let $\mathbf{X}'_i$ be $\mathbf{X}_i$ rounded based on $\mathbf{t}'_i$ as above. Let $\mathbf{Z}_1 = \mathbf{Y} = \alpha_1\mathbf{X} + \sqrt{1-\alpha_1^2}\mathbf{Y}$ and $\mathbf{Z}_2 = \alpha_2\mathbf{X} + \sqrt{1-\alpha_2^2}\mathbf{Y}$. Then the marginal distribution of $\langle \mathbf{X}', \mathbf{Z}_j\rangle$ equals the distribution of $\langle h_{\mathbf{A},\mathbf{t}}(w_j), y_j\mathbf{A}x_j\rangle = y_j\langle h_{\mathbf{A},\mathbf{t}}(w_j), \mathbf{A}x_j\rangle$.

Consider now an arbitrary outcome $X', X$ of $\mathbf{X}, \mathbf{X}'$. We have $\langle \mathbf{Z}_j, X'\rangle \ge \gamma_i/2$ if and only if $\alpha_j\langle X, X'\rangle + \sqrt{1-\alpha_j^2}\langle \mathbf{Y}, X'\rangle \ge \gamma_i/2$. We also have that $\langle \mathbf{Y}, X'\rangle \sim \mathcal{N}(0, \|X'\|_2^2/k)$ and thus

$$\mathbb{P}[\langle \mathbf{Z}_2, X'\rangle \ge \gamma_i/2] - \mathbb{P}[\langle \mathbf{Z}_1, X'\rangle \ge \gamma_i/2] =$$
$$\left(1 - \Phi\left(\sqrt{k}\cdot\frac{\gamma_i/2 - \alpha_2\langle X, X'\rangle}{\sqrt{1-\alpha_2^2}}\right)\right) - \left(1 - \Phi\left(\sqrt{k}\cdot\frac{\gamma_i/2 - \alpha_1\langle X, X'\rangle}{\sqrt{1-\alpha_1^2}}\right)\right) =$$
$$\Phi\left(\sqrt{k}\cdot\frac{\gamma_i/2 - \alpha_1\langle X, X'\rangle}{\sqrt{1-\alpha_1^2}}\right) - \Phi\left(\sqrt{k}\cdot\frac{\gamma_i/2 - \alpha_2\langle X, X'\rangle}{\sqrt{1-\alpha_2^2}}\right). \qquad (34)$$

Here $\Phi(\cdot)$ denotes the cumulative density function of the normal distribution with mean 0 and variance 1. Now let

$$u := \sqrt{k}\cdot\frac{\gamma_i/2 - \alpha_1\langle X, X'\rangle}{\sqrt{1-\alpha_1^2}},$$

and
$$\ell := \sqrt{k} \cdot \frac{\gamma_i/2 - \alpha_2\langle X, X'\rangle}{\sqrt{1-\alpha_2^2}}.$$

Consider now the derivative
$$\frac{\partial}{\partial \alpha} \sqrt{k} \cdot \frac{\gamma_i/2 - \alpha\langle X, X'\rangle}{\sqrt{1-\alpha^2}} = \sqrt{k} \cdot \frac{\alpha\gamma_i/2 - \langle X, X'\rangle}{(1-\alpha^2)^{3/2}}.$$

Assume first that $\|X\|_2^2 \geq 9/10$. Then $\langle X, X'\rangle \geq 8/9$ by Remark 14. Now since $\alpha\gamma_i/2 \leq \gamma_i^2/2 \leq c_\gamma^2/8 \leq 1/9$ for $c_\gamma$ small enough. Thus the derivative when $\|X\|_2^2 \geq 9/10$ is no more than
$$\sqrt{k} \cdot (1/9 - 8/9) \leq -7\sqrt{k}/9.$$

This implies $u - \ell \geq 7(\alpha_2 - \alpha_1)\sqrt{k}/9 > 0$ and therefore
$$\mathbb{P}[\langle \mathbf{Z}_2, X'\rangle \geq \gamma_i/2] \geq \mathbb{P}[\langle \mathbf{Z}_1, X'\rangle \geq \gamma_i/2].$$

If we in addition have that $\|X\|_2^2 \leq 4/3$, then we may even show that the difference in probabilities is large as a function of $\alpha_2 - \alpha_1$ as follows
$$\mathbb{P}[\langle \mathbf{Z}_2, X'\rangle \geq \gamma_i/2] - \mathbb{P}[\langle \mathbf{Z}_1, X'\rangle \geq \gamma_i/2] = \frac{1}{\sqrt{2\pi}} \cdot \int_{x=\ell}^u e^{-x^2/2}dx$$
$$\geq e^{-\max_{a\in[\ell,u]} a^2/2} \frac{7\sqrt{k}(\alpha_2 - \alpha_1)}{9\sqrt{2\pi}}.$$

Observing that
$$\max_{a\in[\ell,u]} a^2 \leq \frac{k}{1-c_\gamma^2} \cdot \max\{\gamma_i^2/2, \gamma_i^2\langle X, X'\rangle^2\},$$

we use Remark 14 to conclude $\langle X, X'\rangle \leq (10/9)\|X\|_2^2$ and thus $u^2 \leq 2k\gamma_i^2(10/9)^2 \leq 3k\gamma_i^2$ for $c_\gamma \leq 1/\sqrt{2}$. This gives us that for any $X$ with $9/10 \leq \|X\|_2^2 \leq 4/3$, it holds that
$$\mathbb{P}[\langle \mathbf{Z}_2, X'\rangle \geq \gamma_i/2] - \mathbb{P}[\langle \mathbf{Z}_1, X'\rangle \geq \gamma_i/2] \geq e^{-3k\gamma_i^2/2} \frac{7\sqrt{k}(\alpha_2 - \alpha_1)}{9\sqrt{2\pi}}.$$

For $\|X\|_2^2 < 9/10$, we have $\|X'\|_2 = \|X' - X + X\|_2 \leq \|X' - X\|_2 + \|X\|_2 \leq \sqrt{k(10\sqrt{k})^{-2}} + \sqrt{9/10} \leq 11/10$. It follows by Cauchy-Schwartz that $|\langle X, X'\rangle| \leq \|X\|_2 \cdot \|X'\|_2 \leq \sqrt{9/10} \cdot 11/10 \leq 11/10$. For $0 \leq \alpha \leq \gamma_i \leq c_\gamma/2 \leq 1/\sqrt{8}$ for $c_\gamma \leq 1/\sqrt{2}$, this upper bounds the derivative by
$$\sqrt{k} \cdot \frac{\gamma_i^2/2 + 11/10}{(1-1/8)^{3/2}} < 2\sqrt{k}.$$

If $u \geq \ell$, we already have that
$$\mathbb{P}[\langle \mathbf{Z}^2, X'\rangle \geq \gamma_i/2] - \mathbb{P}[\langle \mathbf{Z}^1, X'\rangle \geq \gamma_i/2] \geq 0.$$

So assume $u < \ell$. The bound on the derivative gives us that $\ell - u \leq 2\sqrt{k}(\alpha_2 - \alpha_1)$ and we conclude
$$\mathbb{P}[\langle \mathbf{Z}^2, X'\rangle \geq \gamma_i/2] - \mathbb{P}[\langle \mathbf{Z}^1, X'\rangle \geq \gamma_i/2] = -\frac{1}{\sqrt{2\pi}} \cdot \int_{x=u}^\ell e^{-x^2/2}dx$$
$$\geq -e^{-\min_{a\in[u,\ell]} a^2/2} \cdot \frac{2\sqrt{k}(\alpha_2 - \alpha_1)}{\sqrt{2\pi}}$$
$$\geq -\frac{2\sqrt{k}(\alpha_2 - \alpha_1)}{\sqrt{2\pi}}.$$

We finally conclude
$$\mathbb{P}[\langle \mathbf{Z}^2, X'\rangle \geq \gamma_i/2] - \mathbb{P}[\langle \mathbf{Z}^1, X'\rangle \geq \gamma_i/2] \geq$$
$$\mathbb{P}[9/10 \leq \|\mathbf{X}\|_2^2 \leq 4/3] \cdot e^{-3k\gamma_i^2/2} \cdot \frac{7\sqrt{k}(\alpha_2 - \alpha_1)}{9\sqrt{2\pi}} - \mathbb{P}[\|\mathbf{X}\|_2^2 < 9/10] \cdot \frac{2\sqrt{k}(\alpha_2 - \alpha_1)}{\sqrt{2\pi}}. \quad (35)$$

Using Theorem 17, we get

$$\mathbb{P}[9/10 \leq \|\mathbf{X}\|_2^2 \leq 4/3] \geq 1 - 2\exp(-k/800), \tag{36}$$

and

$$\mathbb{P}[\|\mathbf{X}\|_2^2 < 9/10] \leq 2\exp(-k/800).$$

For $k$ at least a sufficiently large constant, we have that (36) is at least $1/2$ and we get that (35) is at least

$$e^{-3k\gamma_i^2/2} \cdot \frac{7\sqrt{k}(\alpha_2 - \alpha_1)}{18\sqrt{2\pi}} - e^{-k/800} \cdot \frac{4\sqrt{k}(\alpha_2 - \alpha_1)}{\sqrt{2\pi}}.$$

For the constant $c_\gamma$ sufficiently small, this is positive as $\gamma_i \leq c_\gamma$. $\qquad\square$

*Proof of Claim 5.* Similarly to the proof of Claim 4, let $w_1, x_1, y_1$ by such that $\alpha_1 = y_1\langle w_1, x_1 \rangle$ and let $w_2, x_2, y_2$ be such that $\alpha_2 = y_2\langle w_2, x_2 \rangle$. Draw $\mathbf{X}, \mathbf{X}'$ and $\mathbf{Z}_1, \mathbf{Z}_2$ as above. Consider again an arbitrary outcome $X', X$ of $\mathbf{X}, \mathbf{X}'$. We have $\langle \mathbf{Z}_j, X' \rangle \leq \gamma_i/2$ if and only if $\alpha_j \langle X, X' \rangle + \sqrt{1 - \alpha_j^2}\langle \mathbf{Y}, X' \rangle \leq \gamma_i/2$. Hence

$$\mathbb{P}[\langle \mathbf{Z}_1, X' \rangle \leq \gamma_i/2] - \mathbb{P}[\langle \mathbf{Z}_2, X' \rangle \leq \gamma_i/2] =$$

$$\Phi\left(\sqrt{k} \cdot \frac{\gamma_i/2 - \alpha_1\langle X, X' \rangle}{\sqrt{1 - \alpha_1^2}}\right) - \Phi\left(\sqrt{k} \cdot \frac{\gamma_i/2 - \alpha_2\langle X, X' \rangle}{\sqrt{1 - \alpha_2^2}}\right).$$

This has the exact same constraints $0 \leq \alpha_1 \leq \alpha_2 \leq \gamma_i$ and exact same form as (34). The conclusion thus follows from the proof of Claim 4. $\qquad\square$

**Restatement of Remark 13.** *If $\|X\|_2^2 \leq 4/3$, then $\|X'\|_2^2 < 2$.*

*Proof.* By the triangle inequality, and using that all coordinates of $X - X'$ are bounded by $(10\sqrt{k})^{-1}$ in absolute value, we have

$$\begin{aligned}
\|X'\|_2^2 &= \|X' - X + X\|_2^2 \\
&\leq (\|X' - X\|_2 + \|X\|_2)^2 \\
&\leq \left(\sqrt{k(10\sqrt{k})^{-2}} + \sqrt{4/3}\right)^2 \\
&= (1/10 + \sqrt{4/3})^2 \\
&< 2.
\end{aligned}$$

$\qquad\square$

**Restatement of Remark 14.** *If $\|X\|_2^2 \geq 9/10$, then $(8/9)\|X\|_2^2 \leq \langle X, X' \rangle \leq (10/9)\|X\|_2^2$*

*Proof.* We have:

$$\begin{aligned}
\langle X', X \rangle &= \langle X' - X + X, X \rangle \\
&= \langle X' - X, X \rangle + \|X\|_2^2.
\end{aligned}$$

Since each coordinate of $X' - X$ is bounded by $(10\sqrt{k})^{-1}$ in absolute value, it follows by Cauchy-Schwartz that

$$\begin{aligned}
|\langle X' - X, X \rangle| &\leq \|X' - X\|_2 \cdot \|X\|_2 \\
&\leq \sqrt{k(10\sqrt{k})^{-2}} \cdot \frac{\|X\|_2^2}{\|X\|_2} \\
&\leq \frac{\|X\|_2^2}{10\sqrt{9/10}} \\
&\leq \|X\|_2^2/9.
\end{aligned}$$

The conclusion follows. $\qquad\square$

**Restatement of Remark 16.** *For any distribution $\mathcal{D}$ over $\mathcal{X} \times \{-1, 1\}$, fixed $w \in \mathcal{H}$, margin $\gamma$ and any $A \in \mathbb{R}^{k \times d}$, it holds with probability at least $1 - \delta$ over $\mathbf{S} \sim \mathcal{D}^n$ that*

$$|\mathcal{L}_{A\mathcal{D}}^{\gamma}(w) - \mathcal{L}_{A\mathbf{S}}^{\gamma}(w)| \leq \sqrt{\frac{8\mathcal{L}_{A\mathcal{D}}^{\gamma}(w)\ln(1/\delta)}{n}} + \frac{2\ln(1/\delta)}{n}.$$

*Proof.* Since $\mathcal{L}_{A\mathbf{S}}^{\gamma}(w)$ is an average of $n$ i.i.d. 0/1 random variables with mean $\mathcal{L}_{A\mathcal{D}}^{\gamma}(w)$, we get from Bernstein's inequality that

$$\mathbb{P}_{\mathbf{S}\sim\mathcal{D}}\left[|\mathcal{L}_{A\mathcal{D}}^{\gamma}(w) - \mathcal{L}_{A\mathbf{S}}^{\gamma}(w)| > \sqrt{\frac{8\mathcal{L}_{A\mathcal{D}}^{\gamma}(w)\ln(1/\delta)}{n}} + \frac{2\ln(1/\delta)}{n}\right] \leq$$

$$\exp\left(-\frac{\frac{1}{2} \cdot \left(\sqrt{8\mathcal{L}_{A\mathcal{D}}^{\gamma}(w)n\ln(1/\delta)} + 2\ln(1/\delta)\right)^2}{n\mathcal{L}_{A\mathcal{D}}^{\gamma}(w) + \frac{1}{3} \cdot \left(\sqrt{8\mathcal{L}_{A\mathcal{D}}^{\gamma}(w)\ln(1/\delta)n} + 2\ln(1/\delta)\right)}\right) \leq$$

$$\exp\left(-\frac{\frac{1}{2} \cdot \max\left\{8\mathcal{L}_{A\mathcal{D}}^{\gamma}(w)n, 4\ln(1/\delta)\right\}\ln(2/\delta)}{\frac{1}{8}\max\{n\mathcal{L}_{A\mathcal{D}}^{\gamma}(w), 4\ln(1/\delta)\} + \frac{1}{3} \cdot \sqrt{2 \cdot \max\{8\mathcal{L}_{A\mathcal{D}}^{\gamma}(w), 4\ln(1/\delta)\} \cdot \ln(1/\delta)}}\right).$$

Using that $\ln(1/\delta) \leq \frac{1}{4}\max\{8\mathcal{L}_{A\mathcal{D}}^{\gamma}(w), 4\ln(1/\delta)\}$, this is at most

$$\exp\left(-\frac{\frac{1}{2}\ln(1/\delta)}{\frac{1}{8} + \frac{1}{3} \cdot \sqrt{\frac{1}{2}}}\right) \leq \exp(-\ln(1/\delta)) = \delta.$$

$\square$