# OpenReview forum: "Tight Generalization Bounds for Large-Margin Halfspaces"
_NeurIPS.cc/2025/Conference — NeurIPS 2025 spotlight_

### Official Review · Reviewer_VQkK · 2025-07-01

**Clarity:** 3
**Significance:** 4
**Originality:** 4
**Rating:** 5
**Confidence:** 3

**Summary:**

The paper considers the problem of learning halfspaces with large-margin and provides tight generalization bounds for this problem. More precisely, assume that we have an unknown distribution $\mathcal{D}$ over $B^d\times\{0,1\}$ where $B_d$ is the $d$-dimensional unit-ball, and assume that we have a sample $S$ of $n$ i.i.d. samples $\mathcal{S}:=(x_i,y_i)_{1\leq i\leq n}\sim\mathcal{D}^n$. We are interested in relating the true loss $\mathcal{L}_D(w)$ for a unit vector $w\in\mathcal{D}$ (which is defined as the probability under $\mathcal{D}$ that $\mathrm{sign}(\langle w,x\rangle)\neq y$) to the $\gamma$-margin empirical loss, which is defined as the fraction of samples $(x,y)\in S$ satisfying $\langle w,x\rangle\leq\gamma$.

There are previously known results providing lower bound and upper bounds of $\mathcal{L}_D(w)$ in terms of $\mathcal{L}_S^\gamma(w)$. While the previous upper and lower bounds are very close and almost match, there were still a gap between the two. The paper provides a tighter upper bound which asymptotically matches the previously known lower bound, and hence it is (asymptotically) tight.

**Questions:**

- Page 1, line 17: What is $\mathbb{B}_2^d$? Is it the hypercube $\{0,1\}^d$ or is it a $d$-dimension ball?
> - If it is the hypercube, is this restriction important?
> - From the discussions in the text it seems to me that most likely it is a $d$-dimensional ball, but then what does the 2 subscript represent? Is it the radius?
Please explain and properly define the non-standard notations in the paper.

- Page 6, line 195: What is $\mathcal{X}$? I might have missed the definition but it doesn't seem to me that it was introduced anywhere. From the discussions in the paper, it seems to me that it is the set of all unit vectors satisfying $\langle w,x\rangle \leq c_\gamma$. Is this correct? If so, please explicitly say that.

- Page 7, line 224: Claims 2: I think that "with probability at least $1-\delta$" should be added to the claim, right?

- Page 8, line 243: Shouldn't "$p(z_i)$" rather depend on $(Aw)_i$? Perhaps it should be written $p((Aw)_i)$ to be more formal?

- Page 8, line 254: (Minor comment:) The formatting/rendering of equation (19) is a bit weird.

**Ethical Concerns:**

["NO or VERY MINOR ethics concerns only"]

**Final Justification:**

The authors have addressed the questions that I had (which were mostly minor), and I can see that the reviews of the other reviewers were similarly positive. I hence keep the score of 5.

**Limitations:**

Since this is a theoretical paper, I do not see any potential negative societal impact.

**Paper Formatting Concerns:**

I have not noticed major formatting concerns.

**Quality:**

3

**Strengths And Weaknesses:**

Strengths:
- The paper provides an asymptotically tight upper bound of the generalization loss on terms of the margin empirical loss. This closes the gap between the upper and lower bounds and settles the problem of characterizing the the generalization loss in this setting.
- In terms of novelty and originality, I find the idea of writing the difference between the loss as an exact sum of difference terms (including Equations (12) and (13)) particularly elegant. This ultimately enabled the authors to perform a very tight analysis of these difference leveraging the powerful machinery of Rademacher complexity.
- In terms of technical correctness, while I cannot 100% vet that all the details of the proof are correct as I have not read the appendices, I did not find major mistakes in the high-level description of the proof in the main body of the paper. Though I have not thoroughly checked all the math, the general proof strategy makes sense to me.
- In terms of clarity and presentation, I find that the structure of how the high-level ideas are presented to be generally done well: I like how the authors motivated their work by starting with a high-level description of the proof of the tightest known upper bounds. The authors described the barriers for such approach to achieve an upper bound matching the lower bound and then explained their strategy (to overcome these barriers in order) to get a tighter upper bound matching the lower bound.

Weaknesses:
- The paper sometimes use notations without properly defining/introducing them. The reader can generally understand the intended meaning from the context in most cases, but this is not always the case. Carefully defining the non-standard notations would definitely improve the readability of the paper. Please refer to the questions below.
- Even though the overall clarity is good, the math writing can be made slightly clearer and more formal in some cases.

---

> ### Author Rebuttal · Authors · 2025-07-28
>
> Thank you for your thorough review.
>
> As other reviewers have said as well, there are definitions missing. Thank you for pointing it out, and for the given examples. In the final version of the paper, we will take special care to make sure that the notation is clear.
>
> As for your questions, Page 1, line 17: What is $\mathbb{B}_2^d$? Is it the hypercube ${0,1}^d$ or is it a $d$-dimension ball?
>
> By $\mathbb{B}_2^d$ we refer to the unit ball, in $d$ dimensions, with respect to the $\ell_2$-norm. So it is the Euclidian unit ball. We will make this more explicit in the final version of the paper.
>
> Page 6, line 195: What is $\mathcal{X}$? I might have missed the definition but it doesn't seem to me that it was introduced anywhere. From the discussions in the paper, it seems to me that it is the set of all unit vectors satisfying $\langle w,x\rangle \leq c_\gamma$. Is this correct? If so, please explicitly say that.
>
> You are correct, it is already explicitly stated on page 6, line 193-194, but phrased a bit convoluted. The sentence "which is the set..." is meant to define $\mathcal{X}$.
>
> Page 7, line 224: Claims 2: I think that "with probability at least $1-\delta$" should be added to the claim, right?
>
> Actually not. We can see that this is phrased in a slightly confusing way. line 224 refers to eq. (17) and (18) both holding simultaneously. Indeed these two equations hold with probability 1-delta. But what we are claiming here is that whenever (17) and (18) hold for some value of delta (delta appears in those two equations), then (16) also holds. The claim that (17) and (18) together imply (16) doesn't really need to know what the probability is that (17) and (18) hold. Of course, we eventually use the claim via union bound over the events in (17) and (18) to establish the claim in lines 195-197.
>
> Page 8, line 243: Shouldn't "$p(z_i)$" rather depend on $(Aw)_i$? Perhaps it should be written $p((Aw)_i)$ to be more formal?
>
> Yes, you are correct, p(z_i) does depend on (Aw)_i. We will make that more clear in the final version of the paper.
>
> Page 8, line 254: (Minor comment:) The formatting/rendering of equation (19) is a bit weird.
>
> We agree that the formatting is weird, we will try our best to make it less weird.

---

> > ### Comment · Reviewer_VQkK · 2025-08-04
> >
> > I would like to thank the authors for the answers to my questions. My review remains positive about the paper and I remain the score.

---

### Official Review · Reviewer_kkYb · 2025-07-02

**Clarity:** 3
**Significance:** 3
**Originality:** 4
**Rating:** 5
**Confidence:** 3

**Summary:**

This paper presents a new, asymptotically tight generalization bound for large-margin halfspace classifiers. The primary contribution is a novel upper bound (Theorem 2) that characterizes the trade-off between the true error of a classifier and its empirical margin loss on a training set. This new bound matches the existing state-of-the-art lower bound.

The work builds upon the previous state-of-the-art bound by Grønlund et al. [2020], which had a small gap compared to the established lower bound. The authors identify and overcome key barriers in the prior proof technique, which relied on random discretization and a framework from Schapire et al. [1998].

**Questions:**

1. The proof relies on a sophisticated random discretization process involving a Johnson-Lindenstrauss transform and random snapping to a grid. While this is used as a proof technique, does it offer any algorithmic insights? Could a learning algorithm based on this discretization be practical or offer any advantages?
2. "While one might argue that our improvement is small in magnitude, this finally pins down the exact generalization performance of a classic learning model". Could you elaborate on the importance of closing this final gap? Does it resolve any long-standing theoretical questions or open up new avenues of inquiry that were previously blocked by the discrepancy in the bounds?

**Ethical Concerns:**

["NO or VERY MINOR ethics concerns only"]

**Limitations:**

yes

**Quality:**

4

**Strengths And Weaknesses:**

Strengths:

1. The paper proves the first generalization bound for large-margin halfspaces that is asymptotically tight. This result is significant as it settles the precise generalization performance of a classic and fundamental learning model.
2. The proof introduces several novel ideas that could be applicable to other generalization bounds. These include a more refined analysis of error terms that avoids inequalities used in prior work, a new application of Rademacher complexity to bound these refined terms, and a method using an infinite sequence of discretization grids to handle error probabilities more effectively.
3. The paper provides a clear and detailed proof overview to build intuition before presenting the full proof. The main theorem is rigorously proven with all assumptions stated, and complex arguments are broken down into smaller, more manageable lemmas with deferred proofs in the appendix for clarity.

Weakness:

N/A

---

> ### Author Rebuttal · Authors · 2025-07-28
>
> Thank you for the thorough review.
>
> As for your questions: The proof relies on a sophisticated random discretization process involving a Johnson-Lindenstrauss transform and random snapping to a grid. While this is used as a proof technique, does it offer any algorithmic insights? Could a learning algorithm based on this discretization be practical or offer any advantages?
>
> This is a very interesting question. We are aware of one previous work that has actually built on exactly this idea to develop a new learning algorithm. The paper "Replicable Learning of Large-Margin Halfspaces" from ICML'24 builds on the Gr\o nlund et al. proof technique, with JL + random snapping, to get a better learning algorithm. It seems highly likely that other applications exist.
>
> "While one might argue that our improvement is small in magnitude, this finally pins down the exact generalization performance of a classic learning model". Could you elaborate on the importance of closing this final gap? Does it resolve any long-standing theoretical questions or open up new avenues of inquiry that were previously blocked by the discrepancy in the bounds?
>
> Yes we believe it solves a long-standing open question. Proving generalisation bounds for large margin half spaces has been an active line of research at least since the first bounds by Bartlett and Shawe-Taylor in 1999. They considered the exact same setup and thus we believe it is fair to say that we close a line of research that has been going for 25+ years. As also mentioned in our reply to reviewers fuQC and kXmV, we believe our new technique may also be applicable to other learning problems now that we finally have a technique capable of proving completely tight bounds.

---

### Official Review · Reviewer_fuQC · 2025-07-06

**Clarity:** 2
**Significance:** 3
**Originality:** 4
**Rating:** 5
**Confidence:** 5

**Summary:**

This paper proves the first generalization bound for large-margin halfspaces that is asymptotically tight in the tradeoff between margin, fraction of training points with the given margin, failure probability, and number of training points. The authors improve upon previous work by Grønlund et al. (2020) by closing the gap between upper and lower bounds. The improvement in tightness pins down the exact generalization performance for the classical halfspace classifier model. The novel techniques used in this paper may also inspire future theoretical analysis on the generalization bounds of other ML models.

**Questions:**

## Major questions
1. **Future direction**: In lines 65-66 on page 3, the authors claim that “Furthermore, our proof of Theorem 2 brings several novel ideas that we hope may find further applications in generalization bounds”. What are the possible future research directions of this work? For example, could the tighter bound be applied to improve other theoretical results?
2. **Extension of the proposed method**: Do the techniques developed here have potential applications to other learning models beyond halfspaces?

## Minor Issues
Here are some minor punctuation problems and typos:
1. Eq. (5) requires a period (“.”) at the end.
2. Eq. (17) requires a period (“.”) at the end.
3. All the terms in Eq. (19) are right-aligned, which is inconsistent with the formatting of other equations in the paper. I could be wrong, but it might be clearer if the second and third terms on the right-hand side of Eq. (19) could be shortened by introducing intermediate notations, allowing for a more consistent left-aligned format.
4. There is an additional space before line 255 on page 8.
5. It would be better if the equation between lines 284 and 285 on page 9 could be left-aligned.

## Missing citations related to SVM
[1] Yuzhou Gu, Zhao Song, Lichen Zhang. “Faster Algorithms for Structured Linear and Kernel Support Vector Machines”. ICLR 2025.
[2] Vladimir Vapnik, Rauf Izmailov.  “Learning using privileged information: similarity control and knowledge transfer”. JMLR 2015.
## Missing citations related to Kernel method
[3] Timothy Chu, Josh Alman, Gary L. Miller, Shyam Narayanan, Mark Sellke, Zhao Song. “Metric Transforms and Low Rank Representations of Kernels for Fast Attention”. NeurIPS 2024.
[4] Josh Alman, Timothy Chu, Aaron Schild, Zhao Song. “Algorithms and Hardness for Linear Algebra on Geometric Graphs”. FOCS 2020.
[5] Corinna Cortes, Mehryar Mohri, Afshin Rostamizadeh. “Generalization Bounds for Learning Kernels”. ICML 2010.
[6] Jan van den Brand, Binghui Peng, Zhao Song, Omri Weinstein. “Training (Overparametrized) Neural Networks in Near-Linear Time”. ITCS 2021.

**Ethical Concerns:**

["NO or VERY MINOR ethics concerns only"]

**Final Justification:**

The paper proves a fundamental result, I recommend acceptance.

**Limitations:**

Yes

**Quality:**

4

**Strengths And Weaknesses:**

## Strengths:
1. **Clear paper organization**: The proof is thorough and well-structured, with clear organization into lemmas and auxiliary results. The authors carefully handle all technical details.
2.  **Well Justification**: Despite the technical complexity, the paper provides good intuition through a detailed proof overview (Section 2) that explains the key ideas and improvements over previous approaches.
3. **Novel technical contributions**: The proof introduces several innovative techniques that collectively enable the tight bound. First, the authors develop a more refined analysis that tracks both positive and negative changes in margins during random discretization, moving beyond previous one-sided approaches that only considered margin violations in one direction. Second, they employ a sophisticated application of Rademacher complexity theory with carefully bounded Lipschitz constants, achieving sharper control over the generalization gap through an improved use of the contraction principle. Third, to handle the challenge of varying norms in the discretized hypotheses, they introduce an infinite sequence of grids with an intricate union bound argument.
## Weaknesses:
1.  **Lack of a formal conclusion section**: The paper ends after completing the main proof without a dedicated conclusion section that summarizes the contributions, discusses the broader implications of achieving tight bounds, or suggests future research directions. This is a minor structural issue that could be easily addressed by adding a brief concluding section.
2.  **Lack of a related work section**: The paper lacks a related work section that positions the contribution within the broader landscape of generalization theory and margin-based learning. However, this is not a major concern, as the clear comparison with Grønlund et al. (2020) in Section 1 has already highlighted the novelty of the contribution, and this issue can be easily fixed by adding a related work section.

---

> ### Author Rebuttal · Authors · 2025-07-28
>
> Thank you for the thorough review.
>
> Thank you for suggesting the creation of additional conclusion and related work sections. We agree that the paper ends rather abruptly, and will strongly consider including a summarizing/concluding section in the final version. We were severely challenged by the page limit and went for giving as clear an overview of the proof structure as possible. As the (potential) final version for NeurIPS allows an additional page, we will prioritise a proper conclusion.
>
> In the introduction of the paper, we aimed to give an overview on how the result improves over the most closely related prior works. Given an additional page, we will put emphasis on discussing related literature more broadly as suggested by the reviewer.
>
> As for your questions, In lines 65-66 on page 3, the authors claim that “Furthermore, our proof of Theorem 2 brings several novel ideas that we hope may find further applications in generalization bounds”. What are the possible future research directions of this work? For example, could the tighter bound be applied to improve other theoretical results?
>
> Do the techniques developed here have potential applications to other learning models beyond halfspaces?
>
> We have one joint answer for the two questions (if we understand them correctly). As also mentioned in our response to reviewer kXmV, we believe our techniques also have a chance of improving known generalisation bounds for voting classifiers as the proofs there rely on the same random discretisation strategy and is lossy in the same way as the Gr\o nlund et al. paper. If the technique can indeed also be applied there, we see no reason why it shouldn't have other applications as well.
>
> Additionally, thank you for pointing out typos and additional references, we will carefully go over each of the proposed references to make sure an expanded related works sections covers all necessary references.

---

> > ### Comment · Reviewer_fuQC · 2025-08-07
> >
> > Thanks for your response. I decide to keep my position score. Looking forward to your revision!

---

### Official Review · Reviewer_kXmV · 2025-07-08

**Clarity:** 3
**Significance:** 3
**Originality:** 4
**Rating:** 5
**Confidence:** 2

**Summary:**

The paper studies the risk of large-margin halfspaces in a standard statistical learning setting.
The authors prove a novel bound on the generalization error $L\_D(w) - L\_S^{\\gamma}(w)$ for any $w$ in the unit $d$-dimensional sphere, where $L\_D(w)$ is the true risk of $w$, which exactly matches the best known lower bound for every margin $\\gamma$, empirical $\\gamma$-margin loss $L\_S^{\\gamma}(w)$, sample size $n$, and confidence $\\delta$.
By doing so, they close the residual gap in the generalization error upper and lower bounds from previous work.

**Questions:**

- Do the authors believe some of the techniques adopted can be easily extended to other settings, or are they particularly tailored to the geometric nature of the large-margin halfspaces?

**Ethical Concerns:**

["NO or VERY MINOR ethics concerns only"]

**Final Justification:**

I keep my already positive initial evaluation since no sudden concern arose during the discussion.

**Limitations:**

Yes

**Quality:**

4

**Strengths And Weaknesses:**

The main result of this paper is pretty clear: it pins down the generalization error of large-margin halfspaces up to a constant factor.
This optimality is achieved through a careful random discretization that first projects with Johnson-Lindenstrauss and then rounds so that the distribution of the rounded margin equals the original one, allowing a cancellation between terms that previous approaches could not exploit.
In their "meet in the middle" bound, the authors even employ a clever multi-scale chaining argument over a countable sequence of grids that removes another residual $\\exp(-k/c)$ factor, thus allowing to select the optimal projection dimension without compromise.
The combination of various different techniques in such a way shows a deep understanding of the problem and a solid technical advancement.

A minor flaw of the paper is that it often omits some central definitions as they are given for granted, which makes it hard to follow the paper for readers not familiar with the topic.
Just to give one example, the authors never define $\\mathbb{B}^d_2$ nor $\\mathcal{S}^{d-1}$, which one may assume to be the unit ball and sphere in $\\mathbb{R}^d$, respectively.
I suggest the authors to clearly define all the adopted notations like these ones in the introduction or in some preliminary section, mainly to avoid any ambiguity.
However, this is not a major issue as the paper is otherwise well-written.
In particular, the proof of the novel generalization error upper bound has a clear structure and can be followed despite its technical nature.

---

> ### Author Rebuttal · Authors · 2025-07-28
>
> Thank you for the thorough review.
>
> We agree that omitting definitions makes the paper hard to follow, we will make sure to include definitions of $\mathbb{B}^d_2$ and $\mathcal{S}^{d-1}$ in the final version. We will also skim through the paper to make sure that there aren't other missing definitions.
>
> As for the question: Do the authors believe some of the techniques adopted can be easily extended to other settings, or are they particularly tailored to the geometric nature of the large-margin halfspaces?
> Yes, we believe the techniques are general enough that they should have applications elsewhere. In particular, random discretizations have previously been used in generalisation bound proofs of e.g. voting classifiers as mentioned in the first line of Section 2.1 [Schapire et al. 1998]. For voting classifiers, a similar gap remains between the upper and lower bounds. We believe our technique is not as such tied to the geometry of half spaces, but instead gives a general way of performing a tighter analysis of random discretisation techniques via the use of Rademacher complexity. Of course, there would be many new problem specific details to work out, but we find the direction promising.

---

> > ### Comment · Reviewer_kXmV · 2025-08-07
> >
> > I thank the authors for their answer. I maintain a positive evaluation.

---

### Decision · Program_Chairs · 2025-09-17

**Decision:**

Accept (spotlight)

**Comment:**

The paper proves tight generalization bounds for large margin classifiers with classification loss bounded by gamma margin loss on the right. The paper is well written and clear and all reviewers agree that the paper is a strong contribution.

Please include the references pointed to by reviewer fuQC with detailed discussion, these seem very relevant to the result at hand, especially the one on generalization on kernel methods.